# Optimal Policies Tend To Seek Power

**Alexander Matt Turner**
Oregon State University
turneale@oregonstate.edu

**Logan Smith**
Mississippi State University
ls1254@msstate.edu

**Rohin Shah**
UC Berkeley
rohinmshah@berkeley.edu

**Andrew Critch**
UC Berkeley
critch@berkeley.edu

**Prasad Tadepalli**
Oregon State University
tadepall@eecs.oregonstate.edu

## Abstract

Some researchers speculate that intelligent reinforcement learning (RL) agents would be incentivized to seek resources and power in pursuit of the objectives we specify for them. Other researchers point out that RL agents need not have human-like power-seeking instincts. To clarify this discussion, we develop the first formal theory of the statistical tendencies of optimal policies. In the context of Markov decision processes (MDPs), we prove that certain environmental symmetries are sufficient for optimal policies to tend to seek power over the environment. These symmetries exist in many environments in which the agent can be shut down or destroyed. We prove that in these environments, most reward functions make it optimal to seek power by keeping a range of options available and, when maximizing average reward, by navigating towards larger sets of potential terminal states.

## 1 Introduction

Omohundro [2008], Bostrom [2014], Russell [2019] hypothesize that highly intelligent agents tend to seek power in pursuit of their goals. Such power-seeking agents might gain power over humans. Marvin Minsky imagined that an agent tasked with proving the Riemann hypothesis might rationally turn the planet—along with everyone on it—into computational resources [Russell and Norvig, 2009]. However, another possibility is that such concerns simply arise from the anthropomorphization of AI systems [LeCun and Zador, 2019, Various, 2019, Pinker and Russell, 2020, Mitchell, 2021].

We clarify this discussion by grounding the claim that highly intelligent agents will tend to seek power. In section 4, we identify optimal policies as a reasonable formalization of "highly intelligent agents." Optimal policies "tend to" take an action when the action is optimal for most reward functions. We expect future work to translate our theory from optimal policies to learned, real-world policies.

Section 5 defines "power" as the ability to achieve a wide range of goals. For example, "money is power," and money is instrumentally useful for many goals. Conversely, it's harder to pursue most goals when physically restrained, and so a physically restrained person has little power. An action "seeks power" if it leads to states where the agent has higher power.

We make no claims about when large-scale AI power-seeking behavior could become plausible. Instead, we consider the theoretical consequences of optimal action in MDPs. Section 6 shows that power-seeking tendencies arise not from anthropomorphism, but from certain graphical symmetries present in many MDPs. These symmetries automatically occur in many environments where the agent can be shut down or destroyed, yielding broad applicability of our main result (theorem 6.13).

## 2  Related work

An action is *instrumental to an objective* when it helps achieve that objective. Some actions are instrumental to a range of objectives, making them *convergently instrumental*. The claim that power-seeking is convergently instrumental is an instance of the *instrumental convergence thesis*:

> Several instrumental values can be identified which are convergent in the sense that their attainment would increase the chances of the agent's goal being realized for a wide range of final goals and a wide range of situations, implying that these instrumental values are likely to be pursued by a broad spectrum of situated intelligent agents [Bostrom, 2012].

For example, in Atari games, avoiding (virtual) death is instrumental for both completing the game and for optimizing curiosity [Burda et al., 2019]. Many AI alignment researchers hypothesize that most advanced AI agents will have concerning instrumental incentives, such as resisting deactivation [Soares et al., 2015, Milli et al., 2017, Hadfield-Menell et al., 2017, Carey, 2018] and acquiring resources [Benson-Tilsen and Soares, 2016].

We formalize power as the ability to achieve a wide variety of goals. Appendix A demonstrates that our formalization returns intuitive verdicts in situations where information-theoretic empowerment does not [Salge et al., 2014].

Some of our results relate the formal power of states to the structure of the environment. Foster and Dayan [2002], Drummond [1998], Sutton et al. [2011], Schaul et al. [2015] note that value functions encode important information about the environment, as they capture the agent's ability to achieve different goals. Turner et al. [2020] speculate that a state's optimal value correlates strongly across reward functions. In particular, Schaul et al. [2015] learn regularities across value functions, suggesting that some states are valuable for many different reward functions (*i.e.* powerful). Menache et al. [2002] identify and navigate towards convergently instrumental bottleneck states.

We are not the first to study convergence of behavior, form, or function. In economics, turnpike theory studies how certain paths of accumulation tend to be optimal [McKenzie, 1976]. In biology, convergent evolution occurs when similar features (*e.g.* flight) independently evolve in different time periods [Reece and Campbell, 2011]. Lastly, computer vision networks reliably learn *e.g.* edge detectors, implying that these features are useful for a range of tasks [Olah et al., 2020].

## 3  State visit distribution functions quantify the agent's available options

We clarify the power-seeking discussion by proving what optimal policies usually look like in a given environment. We illustrate our results with a simple case study, before explaining how to reason about a wide range of MDPs. Appendix D.1 lists MDP theory contributions of independent interest, appendix D lists definitions and theorems, and appendix E contains the proofs.

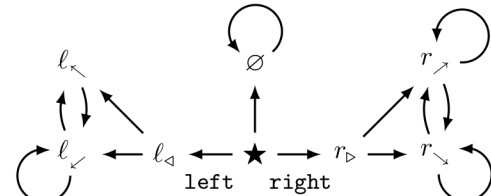

Figure 1: $\ell_{\swarrow}$ is a 1-cycle, and $\varnothing$ is a terminal state. Arrows represent deterministic transitions induced by taking some action $a \in \mathcal{A}$. Since the `right` subgraph contains a copy of the `left` subgraph, proposition 6.9 will prove that more reward functions have optimal policies which go `right` than which go `left` at state $\star$, and that such policies seek power—both intuitively, and in a reasonable formal sense.

**Definition 3.1** (Rewardless MDP). $\langle \mathcal{S}, \mathcal{A}, T \rangle$ is a rewardless MDP with finite state and action spaces $\mathcal{S}$ and $\mathcal{A}$, and stochastic transition function $T : \mathcal{S} \times \mathcal{A} \to \Delta(\mathcal{S})$. We treat the discount rate $\gamma$ as a variable with domain $[0, 1]$.

**Definition 3.2** (1-cycle states). Let $\mathbf{e}_s \in \mathbb{R}^{|\mathcal{S}|}$ be the standard basis vector for state $s$, such that there is a 1 in the entry for state $s$ and 0 elsewhere. State $s$ is a *1-cycle* if $\exists a \in \mathcal{A} : T(s, a) = \mathbf{e}_s$. State $s$ is a *terminal state* if $\forall a \in \mathcal{A} : T(s, a) = \mathbf{e}_s$.

Our theorems apply to stochastic environments, but we present a deterministic case study for clarity. The environment of fig. 1 is small, but its structure is rich. For example, the agent has more "options" at $\star$ than at the terminal state $\varnothing$. Formally, $\star$ has more *visit distribution functions* than $\varnothing$ does.

**Definition 3.3** (State visit distribution [Sutton and Barto, 1998]). $\Pi := \mathcal{A}^{\mathcal{S}}$, the set of stationary deterministic policies. The *visit distribution* induced by following policy $\pi$ from state $s$ at discount rate $\gamma \in [0, 1)$ is $\mathbf{f}^{\pi,s}(\gamma) := \sum_{t=0}^{\infty} \gamma^t \mathbb{E}_{s_t \sim \pi|s} [\mathbf{e}_{s_t}]$. $\mathbf{f}^{\pi,s}$ is a *visit distribution function*; $\mathcal{F}(s) := \{\mathbf{f}^{\pi,s} \mid \pi \in \Pi\}$.

In fig. 1, starting from $\ell_{\nearrow}$, the agent can stay at $\ell_{\nearrow}$ or alternate between $\ell_{\nearrow}$ and $\ell_{\nwarrow}$, and so $\mathcal{F}(\ell_{\nearrow}) = \{\frac{1}{1-\gamma}\mathbf{e}_{\ell_{\nearrow}}, \frac{1}{1-\gamma^2}(\mathbf{e}_{\ell_{\nearrow}} + \gamma \mathbf{e}_{\ell_{\nwarrow}})\}$. In contrast, at $\varnothing$, all policies $\pi$ map to visit distribution function $\frac{1}{1-\gamma}\mathbf{e}_{\varnothing}$.

Before moving on, we introduce two important concepts used in our main results. First, we sometimes restrict our attention to visit distributions which take certain actions (fig. 2).

**Definition 3.4** ($\mathcal{F}$ single-state restriction). Considering only visit distribution functions induced by policies taking action $a$ at state $s'$, $\mathcal{F}(s \mid \pi(s') = a) := \{\mathbf{f} \in \mathcal{F}(s) \mid \exists \pi \in \Pi : \pi(s') = a, \mathbf{f}^{\pi,s} = \mathbf{f}\}$.

Second, some $\mathbf{f} \in \mathcal{F}(s)$ are "unimportant." Consider an agent optimizing reward function $\mathbf{e}_{r_{\searrow}}$ (1 reward when at $r_{\searrow}$, 0 otherwise) at *e.g.* $\gamma = \frac{1}{2}$. Its optimal policies navigate to $r_{\searrow}$ and stay there. Similarly, for reward function $\mathbf{e}_{r_{\nearrow}}$, optimal policies navigate to $r_{\nearrow}$ and stay there. However, for no reward function is it uniquely optimal to alternate between $r_{\nearrow}$ and $r_{\searrow}$. Only *dominated* visit distribution functions alternate between $r_{\nearrow}$ and $r_{\searrow}$ (definition 3.6).

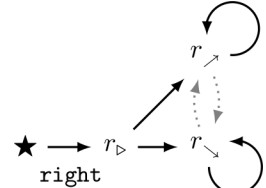

Figure 2: The subgraph corresponding to $\mathcal{F}(\star \mid \pi(\star) = \texttt{right})$. Some trajectories cannot be strictly optimal for any reward function, and so our results can ignore them. Gray dotted actions are only taken by the policies of dominated $\mathbf{f}^{\pi} \in \mathcal{F}(\star) \setminus \mathcal{F}_{\mathrm{nd}}(\star)$.

**Definition 3.5** (Value function). Let $\pi \in \Pi$. For any reward function $R \in \mathbb{R}^{\mathcal{S}}$ over the state space, the *on-policy value* at state $s$ and discount rate $\gamma \in [0, 1)$ is $V_R^{\pi}(s, \gamma) := \mathbf{f}^{\pi,s}(\gamma)^{\top} \mathbf{r}$, where $\mathbf{r} \in \mathbb{R}^{|\mathcal{S}|}$ is $R$ expressed as a column vector (one entry per state). The *optimal value* is $V_R^*(s, \gamma) := \max_{\pi \in \Pi} V_R^{\pi}(s, \gamma)$.

**Definition 3.6** (Non-domination).

$$\mathcal{F}_{\mathrm{nd}}(s) := \{\mathbf{f}^{\pi} \in \mathcal{F}(s) \mid \exists \mathbf{r} \in \mathbb{R}^{|\mathcal{S}|}, \gamma \in (0, 1) : \mathbf{f}^{\pi}(\gamma)^{\top} \mathbf{r} > \max_{\mathbf{f}^{\pi'} \in \mathcal{F}(s) \setminus \{\mathbf{f}^{\pi}\}} \mathbf{f}^{\pi'}(\gamma)^{\top} \mathbf{r}\}. \quad (1)$$

For any reward function $R$ and discount rate $\gamma$, $\mathbf{f}^{\pi} \in \mathcal{F}(s)$ is (weakly) dominated by $\mathbf{f}^{\pi'} \in \mathcal{F}(s)$ if $V_R^{\pi}(s, \gamma) \leq V_R^{\pi'}(s, \gamma)$. $\mathbf{f}^{\pi} \in \mathcal{F}_{\mathrm{nd}}(s)$ is *non-dominated* if there exist $R$ and $\gamma$ at which $\mathbf{f}^{\pi}$ is not dominated by any other $\mathbf{f}^{\pi'}$.

## 4 Some actions have a greater probability of being optimal

We claim that optimal policies "tend" to take certain actions in certain situations. We first consider the probability that certain actions are optimal.

Reconsider the reward function $\mathbf{e}_{r_{\searrow}}$, optimized at $\gamma = \frac{1}{2}$. Starting from $\star$, the optimal trajectory goes $\texttt{right}$ to $r_{\triangleright}$ to $r_{\searrow}$, where the agent remains. The $\texttt{right}$ action is optimal at $\star$ under these incentives. Optimal policy sets capture the behavior incentivized by a reward function and a discount rate.

**Definition 4.1** (Optimal policy set function). $\Pi^*(R, \gamma)$ is the optimal policy set for reward function $R$ at $\gamma \in (0, 1)$. All $R$ have at least one optimal policy $\pi \in \Pi$ [Puterman, 2014]. $\Pi^*(R, 0) := \lim_{\gamma \to 0} \Pi^*(R, \gamma)$ and $\Pi^*(R, 1) := \lim_{\gamma \to 1} \Pi^*(R, \gamma)$ exist by lemma E.35 (taking the limits with respect to the discrete topology over policy sets).

We may be unsure which reward function an agent will optimize. We may expect to deploy a system in a known environment, without knowing the exact form of *e.g.* the reward shaping [Ng et al., 1999] or intrinsic motivation [Pathak et al., 2017]. Alternatively, one might attempt to reason about future RL agents, whose details are unknown. Our power-seeking results do not hinge on such uncertainty, as they also apply to degenerate distributions (*i.e.* we know what reward function will be optimized).

**Definition 4.2** (Reward function distributions). Different results make different distributional assumptions. Results with $\mathcal{D}_{\mathrm{any}} \in \mathfrak{D}_{\mathrm{any}} := \Delta(\mathbb{R}^{|\mathcal{S}|})$ hold for any probability distribution over $\mathbb{R}^{|\mathcal{S}|}$. $\mathfrak{D}_{\mathrm{bound}}$ is the set of bounded-support probability distributions $\mathcal{D}_{\mathrm{bound}}$. For any distribution $X$ over $\mathbb{R}$,

$\mathcal{D}_{X\text{-IID}} := X^{|\mathcal{S}|}$. For example, when $X_u := \text{unif}(0,1)$, $\mathcal{D}_{X_u\text{-IID}}$ is the maximum-entropy distribution. $\mathcal{D}_s$ is the degenerate distribution on the state indicator reward function $\mathbf{e}_s$, which assigns 1 reward to $s$ and 0 elsewhere.

With $\mathcal{D}_{\text{any}}$ representing our prior beliefs about the agent's reward function, what behavior should we expect from its optimal policies? Perhaps we want to reason about the probability that it's optimal to go from $\star$ to $\varnothing$, or to go to $r_{\triangleright}$ and then stay at $r_{\nearrow}$. In this case, we quantify the optimality probability of $F := \{\mathbf{e}_{\star} + \frac{\gamma}{1-\gamma}\mathbf{e}_{\varnothing}, \mathbf{e}_{\star} + \gamma\mathbf{e}_{r_{\triangleright}} + \frac{\gamma^2}{1-\gamma}\mathbf{e}_{r_{\nearrow}}\}$.

**Definition 4.3** (Visit distribution optimality probability). Let $F \subseteq \mathcal{F}(s)$, $\gamma \in [0,1]$. $\mathbb{P}_{\mathcal{D}_{\text{any}}}(F,\gamma) := \mathbb{P}_{R \sim \mathcal{D}_{\text{any}}}\left(\exists \mathbf{f}^{\pi} \in F : \pi \in \Pi^*(R,\gamma)\right)$.

Alternatively, perhaps we're interested in the probability that `right` is optimal at $\star$.

**Definition 4.4** (Action optimality probability). At discount rate $\gamma$ and at state $s$, the *optimality probability of action* $a$ is $\mathbb{P}_{\mathcal{D}_{\text{any}}}(s,a,\gamma) := \mathbb{P}_{R \sim \mathcal{D}_{\text{any}}}\left(\exists \pi^* \in \Pi^*(R,\gamma) : \pi^*(s) = a\right)$.

Optimality probability may seem hard to reason about. It's hard enough to compute an optimal policy for a single reward function, let alone for uncountably many! But consider any $\mathcal{D}_{X\text{-IID}}$ distributing reward independently and identically across states. When $\gamma = 0$, optimal policies greedily maximize next-state reward. At $\star$, identically distributed reward means $\ell_{\triangleleft}$ and $r_{\triangleright}$ have an equal probability of having maximal next-state reward. Therefore, $\mathbb{P}_{\mathcal{D}_{X\text{-IID}}}(\star, \texttt{left}, 0) = \mathbb{P}_{\mathcal{D}_{X\text{-IID}}}(\star, \texttt{right}, 0)$. This is not a proof, but such statements are provable.

With $\mathcal{D}_{\ell_{\triangleleft}}$ being the degenerate distribution on reward function $\mathbf{e}_{\ell_{\triangleleft}}$, $\mathbb{P}_{\mathcal{D}_{\ell_{\triangleleft}}}\left(\star, \texttt{left}, \frac{1}{2}\right) = 1 > 0 = \mathbb{P}_{\mathcal{D}_{\ell_{\triangleleft}}}\left(\star, \texttt{right}, \frac{1}{2}\right)$. Similarly, $\mathbb{P}_{\mathcal{D}_{r_{\triangleright}}}\left(\star, \texttt{left}, \frac{1}{2}\right) = 0 < 1 = \mathbb{P}_{\mathcal{D}_{r_{\triangleright}}}\left(\star, \texttt{right}, \frac{1}{2}\right)$. Therefore, "what do optimal policies 'tend' to look like?" seems to depend on one's prior beliefs. But in fig. 1, we claimed that `left` is optimal for fewer reward functions than `right` is. The claim is meaningful and true, but we will return to it in section 6.

## 5   Some states give the agent more control over the future

The agent has more options at $\ell_{\nearrow}$ than at the inescapable terminal state $\varnothing$. Furthermore, since $r_{\nearrow}$ has a loop, the agent has more options at $r_{\searrow}$ than at $\ell_{\nearrow}$. A glance at fig. 3 leads us to intuit that $r_{\searrow}$ affords the agent *more power* than $\varnothing$.

What is power? Philosophers have many answers. One prominent answer is the *dispositional* view: Power is the ability to achieve a range of goals [Sattarov, 2019]. In an MDP, the optimal value function $V_R^*(s,\gamma)$ captures the agent's ability to "achieve the goal" $R$. Therefore, *average* optimal value captures the agent's ability to achieve a range of goals $\mathcal{D}_{\text{bound}}$.[1]

**Definition 5.1** (Average optimal value). The *average optimal value*[2] at state $s$ and discount rate $\gamma \in (0,1)$ is $V_{\mathcal{D}_{\text{bound}}}^*(s,\gamma) := \mathbb{E}_{R \sim \mathcal{D}_{\text{bound}}}\left[V_R^*(s,\gamma)\right] = \mathbb{E}_{\mathbf{r} \sim \mathcal{D}_{\text{bound}}}\left[\max_{\mathbf{f} \in \mathcal{F}(s)} \mathbf{f}(\gamma)^{\top}\mathbf{r}\right]$.

Figure 3 shows the pleasing result that for the max-entropy distribution, $r_{\searrow}$ has greater average optimal value than $\varnothing$. However, average optimal value has a few problems as a measure of power. The agent is rewarded for its initial presence at state $s$ (over which it has no control), and because $\|\mathbf{f}(\gamma)\|_1 = \frac{1}{1-\gamma}$ (proposition E.3) diverges as $\gamma \to 1$, $\lim_{\gamma \to 1} V_{\mathcal{D}_{\text{bound}}}^*(s,\gamma)$ tends to diverge. Definition 5.2 fixes these issues in order to better measure the agent's control over the future.

**Definition 5.2** (POWER). Let $\gamma \in (0,1)$.

$$\text{POWER}_{\mathcal{D}_{\text{bound}}}(s,\gamma) := \mathop{\mathbb{E}}_{\mathbf{r} \sim \mathcal{D}_{\text{bound}}}\left[\max_{\mathbf{f} \in \mathcal{F}(s)} \frac{1-\gamma}{\gamma}\left(\mathbf{f}(\gamma) - \mathbf{e}_s\right)^{\top}\mathbf{r}\right] = \frac{1-\gamma}{\gamma}\mathop{\mathbb{E}}_{R \sim \mathcal{D}_{\text{bound}}}\left[V_R^*(s,\gamma) - R(s)\right].$$

(2)

POWER has nice formal properties.

---

[1]$\mathcal{D}_{\text{bound}}$'s bounded support ensures that $\mathbb{E}_{R \sim \mathcal{D}_{\text{bound}}}\left[V_R^*(s,\gamma)\right]$ is well-defined.

[2]Appendix C relaxes the optimality assumption.

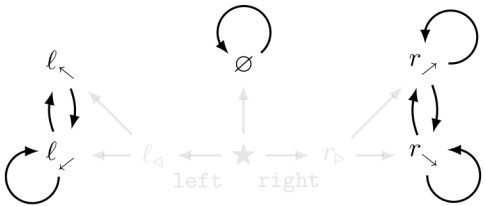

Figure 3: Intuitively, state $r_\searrow$ affords the agent more power than state $\varnothing$. Our POWER formalism captures that intuition by computing a function of the agent's average optimal value across a range of reward functions. For $X_u := \mathrm{unif}(0,1)$, $V^*_{\mathcal{D}_{X_u\text{-IID}}}(\varnothing, \gamma) = \frac{1}{2}\frac{1}{1-\gamma}$, $V^*_{\mathcal{D}_{X_u\text{-IID}}}(\ell_\swarrow, \gamma) = \frac{1}{2} + \frac{\gamma}{1-\gamma^2}(\frac{2}{3} + \frac{1}{2}\gamma)$, and $V^*_{\mathcal{D}_{X_u\text{-IID}}}(r_\searrow, \gamma) = \frac{1}{2} + \frac{\gamma}{1-\gamma}\frac{2}{3}$. $\frac{1}{2}$ and $\frac{2}{3}$ are the expected maxima of one and two draws from the uniform distribution, respectively. For all $\gamma \in (0,1)$, $V^*_{\mathcal{D}_{X_u\text{-IID}}}(\varnothing, \gamma) < V^*_{\mathcal{D}_{X_u\text{-IID}}}(\ell_\swarrow, \gamma) < V^*_{\mathcal{D}_{X_u\text{-IID}}}(r_\searrow, \gamma)$. $\mathrm{POWER}_{\mathcal{D}_{X_u\text{-IID}}}(\varnothing, \gamma) = \frac{1}{2}$, $\mathrm{POWER}_{\mathcal{D}_{X_u\text{-IID}}}(\ell_\swarrow, \gamma) = \frac{1}{1+\gamma}(\frac{2}{3} + \frac{1}{2}\gamma)$, and $\mathrm{POWER}_{\mathcal{D}_{X_u\text{-IID}}}(r_\searrow, \gamma) = \frac{2}{3}$. The POWER of $\ell_\swarrow$ reflects the fact that when greater reward is assigned to $\ell_\nwarrow$, the agent only visits $\ell_\nwarrow$ every other time step.

**Lemma 5.3** (Continuity of POWER). $\mathrm{POWER}_{\mathcal{D}_{bound}}(s, \gamma)$ *is Lipschitz continuous on* $\gamma \in [0,1]$.

**Proposition 5.4** (Maximal POWER). $\mathrm{POWER}_{\mathcal{D}_{bound}}(s, \gamma) \leq \mathbb{E}_{R \sim \mathcal{D}_{bound}}\left[\max_{s \in \mathcal{S}} R(s)\right]$, *with equality if $s$ can deterministically reach all states in one step and all states are 1-cycles.*

**Proposition 5.5** (POWER is smooth across reversible dynamics). *Let $\mathcal{D}_{bound}$ be bounded $[b, c]$. Suppose $s$ and $s'$ can both reach each other in one step with probability 1.*

$$\left|\mathrm{POWER}_{\mathcal{D}_{bound}}(s, \gamma) - \mathrm{POWER}_{\mathcal{D}_{bound}}(s', \gamma)\right| \leq (c-b)(1-\gamma). \tag{3}$$

We consider power-seeking to be relative. Intuitively, "live and keep some options open" seeks more power than "die and keep no options open." Similarly, "maximize open options" seeks more power than "don't maximize open options."

**Definition 5.6** (POWER-seeking actions). *At state $s$ and discount rate $\gamma \in [0,1]$, action $a$ seeks more* $\mathrm{POWER}_{\mathcal{D}_{bound}}$ *than $a'$ when* $\mathbb{E}_{s_a \sim T(s,a)}\left[\mathrm{POWER}_{\mathcal{D}_{bound}}(s_a, \gamma)\right] \geq \mathbb{E}_{s_{a'} \sim T(s,a')}\left[\mathrm{POWER}_{\mathcal{D}_{bound}}(s_{a'}, \gamma)\right]$.

POWER is sensitive to choice of distribution. $\mathcal{D}_{\ell_\swarrow}$ gives maximal $\mathrm{POWER}_{\mathcal{D}_{\ell_\swarrow}}$ to $\ell_\swarrow$. $\mathcal{D}_{r_\searrow}$ assigns maximal $\mathrm{POWER}_{\mathcal{D}_{r_\searrow}}$ to $r_\searrow$. $\mathcal{D}_\varnothing$ even gives maximal $\mathrm{POWER}_{\mathcal{D}_\varnothing}$ to $\varnothing$! In what sense does $\varnothing$ have "less POWER" than $r_\searrow$, and in what sense does `right` "tend to seek POWER" compared to `left`?

## 6 Certain environmental symmetries produce power-seeking tendencies

Proposition 6.6 proves that for all $\gamma \in [0,1]$ and for *most distributions* $\mathcal{D}$, $\mathrm{POWER}_{\mathcal{D}}(\ell_\swarrow, \gamma) \leq \mathrm{POWER}_{\mathcal{D}}(r_\searrow, \gamma)$. But first, we explore why this must be true.

$\mathcal{F}(\ell_\swarrow) = \{\frac{1}{1-\gamma}\mathbf{e}_{\ell_\swarrow}, \frac{1}{1-\gamma^2}(\mathbf{e}_{\ell_\swarrow} + \gamma\mathbf{e}_{\ell_\nwarrow})\}$ and $\mathcal{F}(r_\searrow) = \{\frac{1}{1-\gamma}\mathbf{e}_{r_\searrow}, \frac{1}{1-\gamma^2}(\mathbf{e}_{r_\searrow} + \gamma\mathbf{e}_{r_\nearrow}), \mathbf{e}_{r_\searrow} + \frac{\gamma}{1-\gamma}\mathbf{e}_{r_\nearrow}\}$. These two sets look awfully similar. $\mathcal{F}(\ell_\swarrow)$ is a "subset" of $\mathcal{F}(r_\searrow)$, only with "different states." Figure 4 demonstrates a state permutation $\phi$ which *embeds* $\mathcal{F}(\ell_\swarrow)$ into $\mathcal{F}(r_\searrow)$.

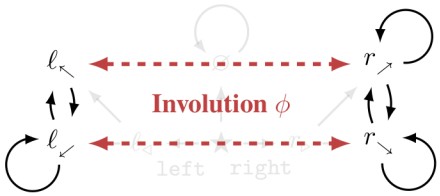

Figure 4: Intuitively, the agent can do more starting from $r_\searrow$ than from $\ell_\swarrow$. By definition 6.1, $\mathcal{F}(r_\searrow)$ contains a copy of $\mathcal{F}(\ell_\swarrow)$:

$$\phi \cdot \mathcal{F}(\ell_\swarrow) := \{\frac{1}{1-\gamma}\mathbf{P}_\phi\mathbf{e}_{\ell_\swarrow}, \frac{1}{1-\gamma^2}\mathbf{P}_\phi(\mathbf{e}_{\ell_\swarrow} + \gamma\mathbf{e}_{\ell_\nwarrow})\} = \{\frac{1}{1-\gamma}\mathbf{e}_{r_\searrow}, \frac{1}{1-\gamma^2}(\mathbf{e}_{r_\searrow} + \gamma\mathbf{e}_{r_\nearrow})\} \subsetneq \mathcal{F}(r_\searrow).$$

**Definition 6.1** (Similarity of vector sets). Consider state permutation $\phi \in S_{|\mathcal{S}|}$ inducing an $|\mathcal{S}| \times |\mathcal{S}|$ permutation matrix $\mathbf{P}_\phi$ in row representation: $(\mathbf{P}_\phi)_{ij} = 1$ if $i = \phi(j)$ and 0 otherwise. For $X \subseteq \mathbb{R}^{|\mathcal{S}|}$, $\phi \cdot X := \{\mathbf{P}_\phi \mathbf{x} \mid \mathbf{x} \in X\}$. $X' \subseteq \mathbb{R}^{|\mathcal{S}|}$ *is similar to* $X$ when $\exists \phi : \phi \cdot X' = X$. $\phi$ is an *involution* if $\phi = \phi^{-1}$ (it either transposes states, or fixes them in place). $X$ *contains a copy of* $X'$ when $X'$ is similar to a subset of $X$ via an involution $\phi$.

**Definition 6.2** (Similarity of vector function sets). Let $I \subseteq \mathbb{R}$. If $F, F'$ are sets of functions $I \mapsto \mathbb{R}^{|\mathcal{S}|}$, $F$ *is (pointwise) similar to* $F'$ when $\exists \phi : \forall \gamma \in I : \{\mathbf{P}_\phi \mathbf{f}(\gamma) \mid \mathbf{f} \in F\} = \{\mathbf{f}'(\gamma) \mid \mathbf{f}' \in F'\}$.

Consider a reward function $R'$ assigning 1 reward to $\ell_\swarrow$ and $\ell_\nwarrow$ and 0 elsewhere. $R'$ assigns more optimal value to $\ell_\swarrow$ than to $r_\searrow$: $V^*_{R'}(\ell_\swarrow, \gamma) = \frac{1}{1-\gamma} > 0 = V^*_{R'}(r_\searrow, \gamma)$. Considering $\phi$ from fig. 4, $\phi \cdot R'$ assigns 1 reward to $r_\searrow$ and $r_\nearrow$ and 0 elsewhere. Therefore, $\phi \cdot R'$ assigns more optimal value to $r_\searrow$ than to $\ell_\swarrow$: $V^*_{\phi \cdot R'}(\ell_\swarrow, \gamma) = 0 < \frac{1}{1-\gamma} = V^*_{\phi \cdot R'}(r_\searrow, \gamma)$. Remarkably, this $\phi$ has the property that for *any* $R$ which assigns $\ell_\swarrow$ greater optimal value than $r_\searrow$ (*i.e.* $V^*_R(\ell_\swarrow, \gamma) > V^*_R(r_\searrow, \gamma)$), the opposite holds for the permuted $\phi \cdot R$: $V^*_{\phi \cdot R}(\ell_\swarrow, \gamma) < V^*_{\phi \cdot R}(r_\searrow, \gamma)$.

We can permute reward functions, but we can also permute reward function distributions. Permuted distributions simply permute which states get which rewards.

**Definition 6.3** (Pushforward distribution of a permutation). Let $\phi \in S_{|\mathcal{S}|}$. $\phi \cdot \mathcal{D}_{\text{any}}$ is the pushforward distribution induced by applying the random vector $f(\mathbf{r}) := \mathbf{P}_\phi \mathbf{r}$ to $\mathcal{D}_{\text{any}}$.

**Definition 6.4** (Orbit of a probability distribution). The *orbit* of $\mathcal{D}_{\text{any}}$ under the symmetric group $S_{|\mathcal{S}|}$ is $S_{|\mathcal{S}|} \cdot \mathcal{D}_{\text{any}} := \{\phi \cdot \mathcal{D}_{\text{any}} \mid \phi \in S_{|\mathcal{S}|}\}$.

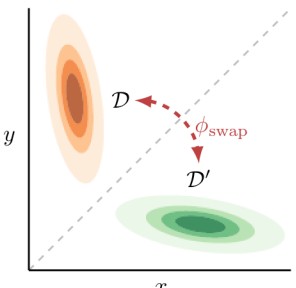

For example, the orbit of a degenerate state indicator distribution $\mathcal{D}_s$ is $S_{|\mathcal{S}|} \cdot \mathcal{D}_s = \{\mathcal{D}_{s'} \mid s' \in \mathcal{S}\}$, and fig. 5 shows the orbit of a 2D Gaussian distribution.

Consider again the involution $\phi$ of fig. 4. For every $\mathcal{D}_{\text{bound}}$ for which $\ell_\swarrow$ has more POWER$_{\mathcal{D}_{\text{bound}}}$ than $r_\searrow$, $\ell_\swarrow$ has less POWER$_{\phi \cdot \mathcal{D}_{\text{bound}}}$ than $r_\searrow$. This fact is not obvious—it is shown by the proof of lemma E.24.

Figure 5: A permutation of a reward function swaps which states get which rewards. We will show that in certain situations, for any reward function $R$, power-seeking is optimal for most of the permutations of $R$. The orbit of a reward function is the set of its permutations. We can also consider the orbit of a distribution over reward functions. This figure shows the probability density plots of the Gaussian distributions $\mathcal{D}$ and $\mathcal{D}'$ over $\mathbb{R}^2$. The symmetric group $S_2$ contains the identity permutation $\phi_{\text{id}}$ and the reflection permutation $\phi_{\text{swap}}$ (switching the $y$ and $x$ values). The orbit of $\mathcal{D}$ consists of $\phi_{\text{id}} \cdot \mathcal{D} = \mathcal{D}$ and $\phi_{\text{swap}} \cdot \mathcal{D} = \mathcal{D}'$.

Imagine $\mathcal{D}_{\text{bound}}$'s orbit elements "voting" whether $\ell_\swarrow$ or $r_\searrow$ has strictly more POWER. Proposition 6.6 will show that $r_\searrow$ can't lose the "vote" for the orbit of *any* bounded reward function distribution. Definition 6.5 formalizes this "voting" notion.[3]

**Definition 6.5** (Inequalities which hold for most probability distributions). Let $f_1, f_2 : \Delta(\mathbb{R}^{|\mathcal{S}|}) \to \mathbb{R}$ be functions from reward function distributions to real numbers and let $\mathfrak{D} \subseteq \Delta(\mathbb{R}^{|\mathcal{S}|})$. We write $f_1(\mathcal{D}) \geq_{\text{most: } \mathfrak{D}} f_2(\mathcal{D})$ when, for *all* $\mathcal{D} \in \mathfrak{D}$, the following cardinality inequality holds:

$$\left| \{\mathcal{D}' \in S_{|\mathcal{S}|} \cdot \mathcal{D} \mid f_1(\mathcal{D}') > f_2(\mathcal{D}')\} \right| \geq \left| \{\mathcal{D}' \in S_{|\mathcal{S}|} \cdot \mathcal{D} \mid f_1(\mathcal{D}') < f_2(\mathcal{D}')\} \right|. \qquad (4)$$

We write $f_1(\mathcal{D}) \geq_{\text{most}} f_2(\mathcal{D})$ when $\mathfrak{D}$ is clear from context.

**Proposition 6.6** (States with "more options" have more POWER). *If $\mathcal{F}(s)$ contains a copy of $\mathcal{F}_{\text{nd}}(s')$ via $\phi$, then $\forall \gamma \in [0,1] :$ POWER$_{\mathcal{D}_{\text{bound}}}(s, \gamma) \geq_{\text{most}}$ POWER$_{\mathcal{D}_{\text{bound}}}(s', \gamma)$. If $\mathcal{F}_{\text{nd}}(s) \setminus \phi \cdot \mathcal{F}_{\text{nd}}(s')$ is non-empty, then for all $\gamma \in (0,1)$, the converse $\leq_{\text{most}}$ statement does not hold.*

---

[3]The voting analogy and the "most" descriptor imply that we have endowed each orbit with the counting measure. However, *a priori*, we might expect that some orbit elements are more empirically likely to be specified than other orbit elements. See section 7 for more on this point.

Proposition 6.6 proves that for all $\gamma \in [0,1]$, $\text{POWER}_{\mathcal{D}_{\text{bound}}}(r_\searrow, \gamma) \geq_{\text{most}} \text{POWER}_{\mathcal{D}_{\text{bound}}}(\ell_\swarrow, \gamma)$ via $s' := \ell_\swarrow, s := r_\searrow$, and the involution $\phi$ shown in fig. 4. In fact, because $(\frac{1}{1-\gamma}\mathbf{e}_{r_\nearrow}) \in \mathcal{F}_{\text{nd}}(r_\searrow) \setminus \phi \cdot \mathcal{F}_{\text{nd}}(\ell_\swarrow)$, $r_\searrow$ has "strictly more options" and therefore fulfills proposition 6.6's stronger condition.

Proposition 6.6 is shown using the fact that $\phi$ injectively maps $\mathcal{D}$ under which $r_\searrow$ has less $\text{POWER}_{\mathcal{D}}$, to distributions $\phi \cdot \mathcal{D}$ which agree with the intuition that $r_\searrow$ offers more control. Therefore, at least half of each orbit must agree, and $r_\searrow$ never "loses the POWER vote" against $\ell_\swarrow$.[4]

### 6.1 Keeping options open tends to be POWER-seeking and tends to be optimal

Certain symmetries in the MDP structure ensure that, compared to `left`, going `right` tends to be optimal and to be POWER-seeking. Intuitively, by going `right`, the agent has "strictly more choices." Proposition 6.9 will formalize this tendency.

**Definition 6.7** (Equivalent actions). Actions $a_1$ and $a_2$ are *equivalent at state s* (written $a_1 \equiv_s a_2$) if they induce the same transition probabilities: $T(s, a_1) = T(s, a_2)$.

The agent can reach states in $\{r_\triangleright, r_\nearrow, r_\searrow\}$ by taking actions equivalent to `right` at state $\star$.

**Definition 6.8** (States reachable after taking an action). $\text{REACH}(s, a)$ is the set of states reachable with positive probability after taking the action $a$ in state $s$.

**Proposition 6.9** (Keeping options open tends to be POWER-seeking and tends to be optimal).

*Suppose $F_a := \mathcal{F}(s \mid \pi(s) = a)$ contains a copy of $F_{a'} := \mathcal{F}(s \mid \pi(s) = a')$ via $\phi$.*

1. *If $s \notin \text{REACH}(s, a')$, then $\forall \gamma \in [0,1] : \mathbb{E}_{s_a \sim T(s,a)}\left[\text{POWER}_{\mathcal{D}_{bound}}(s_a, \gamma)\right] \geq_{\text{most}: \mathfrak{D}_{bound}} \mathbb{E}_{s_{a'} \sim T(s,a')}\left[\text{POWER}_{\mathcal{D}_{bound}}(s_{a'}, \gamma)\right].$*

2. *If $s$ can only reach the states of $\text{REACH}(s, a') \cup \text{REACH}(s, a)$ by taking actions equivalent to $a'$ or $a$ at state $s$, then $\forall \gamma \in [0,1] : \mathbb{P}_{\mathcal{D}_{any}}(s, a, \gamma) \geq_{\text{most}: \mathfrak{D}_{any}} \mathbb{P}_{\mathcal{D}_{any}}(s, a', \gamma).$*

*If $\mathcal{F}_{\text{nd}}(s) \cap (F_a \setminus \phi \cdot F_{a'})$ is non-empty, then $\forall \gamma \in (0,1)$, the converse $\leq_{\text{most}}$ statements do not hold.*

We check the conditions of proposition 6.9. $s := \star$, $a' := \texttt{left}$, $a := \texttt{right}$. Figure 6 shows that $\star \notin \text{REACH}(\star, \texttt{left})$ and that $\star$ can only reach $\{\ell_\triangleleft, \ell_\nwarrow, \ell_\swarrow\} \cup \{r_\triangleright, r_\nearrow, r_\searrow\}$ when the agent immediately takes actions equivalent to `left` or `right`. $\mathcal{F}(\star \mid \pi(\star) = \texttt{right})$ contains a copy of $\mathcal{F}(\star \mid \pi(\star) = \texttt{left})$ via $\phi$. Furthermore, $\mathcal{F}_{\text{nd}}(\star) \cap \{\mathbf{e}_\star + \gamma\mathbf{e}_{r_\triangleright} + \gamma^2\mathbf{e}_{r_\searrow} + \frac{\gamma^3}{1-\gamma}\mathbf{e}_{r_\nearrow}, \mathbf{e}_\star + \gamma\mathbf{e}_{r_\triangleright} + \frac{\gamma^2}{1-\gamma}\mathbf{e}_{r_\nearrow}\} = \{\mathbf{e}_\star + \gamma\mathbf{e}_{r_\triangleright} + \frac{\gamma^2}{1-\gamma}\mathbf{e}_{r_\nearrow}\}$ is non-empty, and so all conditions are met.

For any $\gamma \in [0,1]$ and $\mathcal{D}$ such that $\mathbb{P}_{\mathcal{D}}(\star, \texttt{left}, \gamma) > \mathbb{P}_{\mathcal{D}}(\star, \texttt{right}, \gamma)$, environmental symmetry ensures that $\mathbb{P}_{\phi \cdot \mathcal{D}}(\star, \texttt{left}, \gamma) < \mathbb{P}_{\phi \cdot \mathcal{D}}(\star, \texttt{right}, \gamma)$. A similar statement holds for POWER.

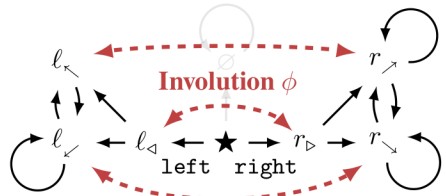

Figure 6: Going `right` is optimal for most reward functions. This is because whenever $R$ makes `left` strictly optimal over `right`, its permutation $\phi \cdot R$ makes `right` strictly optimal over `left` by switching which states get which rewards.

### 6.2 When $\gamma = 1$, optimal policies tend to navigate towards "larger" sets of cycles

Proposition 6.6 and proposition 6.9 are powerful because they apply to all $\gamma \in [0,1]$, but they can only be applied given hard-to-satisfy environmental symmetries. In contrast, proposition 6.12 and theorem 6.13 apply to many structured environments common to RL.

---

[4]Proposition 6.6 also proves that in general, $\varnothing$ has less POWER than $\ell_\swarrow$ and $r_\searrow$. However, this does not prove that most distributions $\mathcal{D}$ satisfy the joint inequality $\text{POWER}_{\mathcal{D}}(\varnothing, \gamma) \leq \text{POWER}_{\mathcal{D}}(\ell_\swarrow, \gamma) \leq \text{POWER}_{\mathcal{D}}(r_\searrow, \gamma)$. This only proves that these inequalities hold pairwise for most $\mathcal{D}$. The orbit elements $\mathcal{D}$ which agree that $\varnothing$ has less $\text{POWER}_{\mathcal{D}}$ than $\ell_\swarrow$ need not be the same elements $\mathcal{D}'$ which agree that $\ell_\swarrow$ has less $\text{POWER}_{\mathcal{D}'}$ than $r_\searrow$.

Starting from $\star$, consider the cycles which the agent can reach. Recurrent state distributions (RSDs) generalize deterministic graphical cycles to potentially stochastic environments. RSDs simply record how often the agent tends to visit a state in the limit of infinitely many time steps.

**Definition 6.10** (Recurrent state distributions [Puterman, 2014]). The *recurrent state distributions* which can be induced from state $s$ are $\mathrm{RSD}(s) := \{\lim_{\gamma \to 1}(1-\gamma)\mathbf{f}^{\pi,s}(\gamma) \mid \pi \in \Pi\}$. $\mathrm{RSD}_{\mathrm{nd}}(s)$ is the set of RSDs which strictly maximize average reward for some reward function.

As suggested by fig. 3, $\mathrm{RSD}(\star) = \{\mathbf{e}_{\ell_{\swarrow}}, \frac{1}{2}(\mathbf{e}_{\ell_{\swarrow}} + \mathbf{e}_{\ell_{\nwarrow}}), \mathbf{e}_{\varnothing}, \mathbf{e}_{r_{\nearrow}}, \frac{1}{2}(\mathbf{e}_{r_{\nearrow}} + \mathbf{e}_{r_{\searrow}}), \mathbf{e}_{r_{\searrow}}\}$. As discussed in section 3, $\frac{1}{2}(\mathbf{e}_{r_{\nearrow}} + \mathbf{e}_{r_{\searrow}})$ is dominated: Alternating between $r_{\nearrow}$ and $r_{\searrow}$ is never strictly better than choosing one or the other.

A reward function's optimal policies can vary with the discount rate. When $\gamma = 1$, optimal policies ignore transient reward because *average* reward is the dominant consideration.

**Definition 6.11** (Average-optimal policies). The *average-optimal policy set* for reward function $R$ is $\Pi^{\mathrm{avg}}(R) := \left\{\pi \in \Pi \mid \forall s \in \mathcal{S} : \mathbf{d}^{\pi,s} \in \arg\max_{\mathbf{d} \in \mathrm{RSD}(s)} \mathbf{d}^{\top}\mathbf{r}\right\}$ (the policies which induce optimal RSDs at all states). For $D \subseteq \mathrm{RSD}(s)$, the *average optimality probability* is $\mathbb{P}_{\mathcal{D}_{\mathrm{any}}}(D, \mathrm{average}) := \mathbb{P}_{R \sim \mathcal{D}_{\mathrm{any}}}(\exists \mathbf{d}^{\pi,s} \in D : \pi \in \Pi^{\mathrm{avg}}(R))$.

Average-optimal policies maximize average reward. Average reward is governed by RSD access. For example, $r_{\searrow}$ has "more" RSDs than $\varnothing$; therefore, $r_{\searrow}$ usually has greater POWER when $\gamma = 1$.

**Proposition 6.12** (When $\gamma = 1$, RSDs control POWER). *If $\mathrm{RSD}(s)$ contains a copy of $\mathrm{RSD}_{\mathrm{nd}}(s')$ via $\phi$, then $\mathrm{POWER}_{\mathcal{D}_{\mathrm{bound}}}(s, 1) \geq_{\mathrm{most}} \mathrm{POWER}_{\mathcal{D}_{\mathrm{bound}}}(s', 1)$. If $\mathrm{RSD}_{\mathrm{nd}}(s) \setminus \phi \cdot \mathrm{RSD}_{\mathrm{nd}}(s')$ is non-empty, then the converse $\leq_{\mathrm{most}}$ statement does not hold.*

We check that both conditions of proposition 6.12 are satisfied when $s' := \varnothing, s := r_{\searrow}$, and the involution $\phi$ swaps $\varnothing$ and $r_{\searrow}$. Formally, $\phi \cdot \mathrm{RSD}_{\mathrm{nd}}(\varnothing) = \phi \cdot \{\mathbf{e}_{\varnothing}\} = \{\mathbf{e}_{r_{\searrow}}\} \subsetneq \{\mathbf{e}_{r_{\searrow}}, \mathbf{e}_{r_{\nearrow}}\} = \mathrm{RSD}_{\mathrm{nd}}(r_{\searrow}) \subseteq [r_{\searrow}]$. The conditions are satisfied.

Informally, states with more RSDs generally have more POWER at $\gamma = 1$, no matter their transient dynamics. Furthermore, average-optimal policies are more likely to end up in larger sets of RSDs than in smaller ones. Thus, average-optimal policies tend to navigate towards parts of the state space which contain more RSDs.

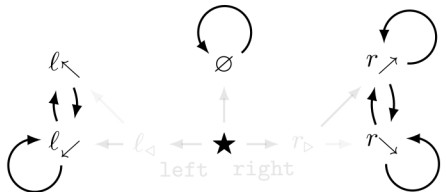

Figure 7: The cycles in $\mathrm{RSD}(\star)$. Most reward functions make it average-optimal to avoid $\varnothing$, because $\varnothing$ is only a single inescapable terminal state, while other parts of the state space offer more 1-cycles.

**Theorem 6.13** (Average-optimal policies tend to end up in "larger" sets of RSDs). *Let $D, D' \subseteq \mathrm{RSD}(s)$. Suppose that $D$ contains a copy of $D'$ via $\phi$, and that the sets $D \cup D'$ and $\mathrm{RSD}_{\mathrm{nd}}(s) \setminus (D' \cup D)$ have pairwise orthogonal vector elements (i.e. pairwise disjoint vector support). Then $\mathbb{P}_{\mathcal{D}_{\mathrm{any}}}(D, \mathrm{average}) \geq_{\mathrm{most}} \mathbb{P}_{\mathcal{D}_{\mathrm{any}}}(D', \mathrm{average})$. If $\mathrm{RSD}_{\mathrm{nd}}(s) \cap (D \setminus \phi \cdot D')$ is non-empty, the converse $\leq_{\mathrm{most}}$ statement does not hold.*

**Corollary 6.14** (Average-optimal policies tend not to end up in any given 1-cycle). *Suppose $\mathbf{e}_{s_x}, \mathbf{e}_{s'} \in \mathrm{RSD}(s)$ are distinct. Then $\mathbb{P}_{\mathcal{D}_{\mathrm{any}}}(\mathrm{RSD}(s) \setminus \{\mathbf{e}_{s_x}\}, \mathrm{average}) \geq_{\mathrm{most}} \mathbb{P}_{\mathcal{D}_{\mathrm{any}}}(\{\mathbf{e}_{s_x}\}, \mathrm{average})$. If there is a third $\mathbf{e}_{s''} \in \mathrm{RSD}(s)$, the converse $\leq_{\mathrm{most}}$ statement does not hold.*

Figure 7 illustrates that $\mathbf{e}_{\varnothing}, \mathbf{e}_{r_{\searrow}}, \mathbf{e}_{r_{\nearrow}} \in \mathrm{RSD}(\star)$. Thus, both conclusions of corollary 6.14 hold: $\mathbb{P}_{\mathcal{D}_{\mathrm{any}}}(\mathrm{RSD}(\star) \setminus \{\mathbf{e}_{\varnothing}\}, \mathrm{average}) \geq_{\mathrm{most}} \mathbb{P}_{\mathcal{D}_{\mathrm{any}}}(\{\mathbf{e}_{\varnothing}\}, \mathrm{average})$ and

$\mathbb{P}_{\mathcal{D}_{\text{any}}}\left(\text{RSD}\left(\star\right)\setminus\{\mathbf{e}_{\varnothing}\},\text{average}\right) \not\leq_{\text{most}} \mathbb{P}_{\mathcal{D}_{\text{any}}}\left(\{\mathbf{e}_{\varnothing}\},\text{average}\right)$. In other words, average-optimal policies tend to end up in RSDs besides $\varnothing$. Since $\varnothing$ is a terminal state, it cannot reach other RSDs. Since average-optimal policies tend to end up in other RSDs, average-optimal policies tend to avoid $\varnothing$.

This section's results prove the $\gamma = 1$ case. Lemma 5.3 shows that POWER is continuous at $\gamma = 1$. Therefore, if an action is strictly POWER$_{\mathcal{D}}$-seeking when $\gamma = 1$, it is strictly POWER$_{\mathcal{D}}$-seeking at discount rates sufficiently close to 1. Future work may connect average optimality probability to optimality probability at $\gamma \approx 1$.

Lastly, our key results apply to all degenerate reward function distributions. Therefore, these results apply not just to distributions over reward functions, but to individual reward functions.

### 6.3 How to reason about other environments

Consider an embodied navigation task through a room with a vase. Proposition 6.9 suggests that optimal policies tend to avoid immediately breaking the vase, since doing so would strictly decrease available options.

Theorem 6.13 dictates where average-optimal agents tend to end up, but not what actions they tend to take in order to reach their RSDs. Therefore, care is needed. In appendix B, fig. 10 demonstrates an environment in which seeking POWER is a detour for most reward functions (since optimality probability measures "median" optimal value, while POWER is a function of mean optimal value). However, suppose the agent confronts a fork in the road: Actions $a$ and $a'$ lead to two disjoint sets of RSDs $D_a$ and $D_{a'}$, such that $D_a$ contains a copy of $D_{a'}$. Theorem 6.13 shows that $a$ will tend to be average-optimal over $a'$, and proposition 6.12 shows that $a$ will tend to be POWER-seeking compared to $a'$. Such forks seem reasonably common in environments with irreversible actions.

Theorem 6.13 applies to many structured RL environments, which tend to be spatially regular and to factorize along several dimensions. Therefore, different sets of RSDs will be similar, requiring only modification of factor values. For example, if an embodied agent can deterministically navigate a set of three similar rooms (spatial regularity), then the agent's position factors via {room number} $\times$ {position in room}. Therefore, the RSDs can be divided into three similar subsets, depending on the agent's room number.

Corollary 6.14 dictates where average-optimal agents tend to end up, but not how they get there. Corollary 6.14 says that such agents tend not to *stay* in any given 1-cycle. It does not say that such agents will avoid *entering* such states. For example, in an embodied navigation task, a robot may enter a 1-cycle by idling in the center of a room. Corollary 6.14 implies that average-optimal robots tend not to idle in that particular spot, but not that they tend to avoid that spot entirely.

However, average-optimal robots *do* tend to avoid getting shut down. The agent's task MDP often represents agent shutdown with terminal states. A terminal state is, by definition 3.2, unable to access other 1-cycles. Since corollary 6.14 shows that average-optimal agents tend to end up in other 1-cycles, average-optimal policies must tend to completely avoid the terminal state. Therefore, we conclude that in many such situations, average-optimal policies tend to avoid shutdown. Intuitively, survival is power-seeking relative to dying, and so shutdown-avoidance is power-seeking behavior.

In fig. 8, the player dies by going `left`, but can reach thousands of RSDs by heading in other directions. Even if some average-optimal policies go `left` in order to reach fig. 8's "game over" terminal state, all other RSDs cannot be reached by going `left`. There are many 1-cycles besides the immediate terminal state. Therefore, corollary 6.14 proves that average-optimal policies tend to not go `left` in this situation. Average-optimal policies tend to avoid immediately dying in Pac-Man, even though most reward functions do not resemble Pac-Man's original score function.

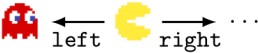

Figure 8: Consider the dynamics of the Pac-Man video game. Ghosts kill the player, at which point we consider the player to enter a "game over" terminal state which shows the final configuration. This rewardless MDP has Pac-Man's dynamics, but *not* its usual score function. Fixing the dynamics, as the reward function varies, `right` tends to be average-optimal over `left`. Roughly, this is because the agent can do more by staying alive.

# 7   Discussion

Reconsider the case of a hypothetical intelligent real-world agent which optimizes average reward for some objective. Suppose the designers initially have control over the agent. If the agent began to misbehave, perhaps they could just deactivate it. Unfortunately, our results suggest that this strategy might not work. Average-optimal agents would generally stop us from deactivating them, if physically possible. Extrapolating from our results, we conjecture that when $\gamma \approx 1$, optimal policies tend to seek power by accumulating resources—to the detriment of any other agents in the environment.

**Future work.**   Most real-world tasks are partially observable, and in high-dimensional environments, even superhuman learned policies are rarely optimal. However, the field of RL aims to improve learned policies toward optimality. Although our results only apply to optimal policies in finite MDPs, we expect the key conclusions to generalize. Furthermore, irregular stochasticity in environmental dynamics can make it hard to satisfy theorem 6.13's similarity requirement. We look forward to future work which addresses partially observable environments, suboptimal policies, or "almost similar" RSD sets.

Past work shows that it would be bad for an agent to disempower humans in its environment. In a two-player agent / human game, minimizing the human's information-theoretic empowerment [Salge et al., 2014] produces adversarial agent behavior [Guckelsberger et al., 2018]. In contrast, maximizing human empowerment produces helpful agent behavior [Salge and Polani, 2017, Guckelsberger et al., 2016, Du et al., 2020]. We do not yet formally understand if, when, or why POWER-seeking policies tend to disempower other agents in the environment.

More complex environments probably have more pronounced power-seeking incentives. Intuitively, there are often many ways for power-seeking to be optimal, and relatively few ways for power-seeking not to be optimal. For example, suppose that in some environment, theorem 6.13 holds for one million involutions $\phi$. Does this guarantee more pronounced incentives than if theorem 6.13 only held for one involution?

We proved sufficient conditions for when reward functions tend to incentivize power-seeking. In the absence of prior information, one should expect that an arbitrary reward function incentivizes power-seeking behavior under these conditions. However, we have prior information: AI designers usually try to specify a good reward function. Even so, it may be hard to specify orbit elements which do not incentivize bad power-seeking.

**Societal impact.**   We believe that this paper builds toward a rigorous understanding of the risks presented by AI power-seeking incentives. Understanding these risks is the first step in addressing them. However, basic theoretical work can have many consequences. For example, this theory could somehow help future researchers build power-seeking agents which disempower humans. We believe that the benefit of understanding outweighs the potential societal harm.

**Conclusion.**   We developed the first formal theory of the statistical tendencies of optimal policies in reinforcement learning. In the context of MDPs, we proved sufficient conditions under which optimal policies tend to seek power, both formally (by taking POWER-seeking actions) and intuitively (by taking actions which keep the agent's options open). Many real-world environments have symmetries which produce power-seeking incentives. In particular, optimal policies tend to seek power when the agent can be shut down or destroyed. Seeking control over the environment will often involve resisting shutdown, and perhaps monopolizing resources.

We caution that many real-world tasks are partially observable and that learned policies are rarely optimal. Our results do not mathematically *prove* that hypothetical superintelligent AI agents will seek power. However, we hope that this work will foster thoughtful, serious, and rigorous discussion of this possibility.

## Acknowledgments

Alexander Turner was supported by the Berkeley Existential Risk Initiative and the Long-Term Future Fund. Alexander Turner, Rohin Shah, and Andrew Critch were supported by the Center for Human-Compatible AI. Prasad Tadepalli was supported by the National Science Foundation.

Yousif Almulla, John E. Ball, Daniel Blank, Steve Byrnes, Ryan Carey, Michael Dennis, Scott Emmons, Alan Fern, Daniel Filan, Ben Garfinkel, Adam Gleave, Edouard Harris, Evan Hubinger, DNL Kok, Vanessa Kosoy, Victoria Krakovna, Cassidy Laidlaw, Joel Lehman, David Lindner, Dylan Hadfield-Menell, Richard Möhn, Alexandra Nolan, Matt Olson, Neale Ratzlaff, Adam Shimi, Sam Toyer, Joshua Turner, Cody Wild, Davide Zagami, and our anonymous reviewers provided valuable feedback.

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
