## Appendix A  Comparing POWER with information-theoretic empowerment

Salge et al. [2014] define information-theoretic *empowerment* as the maximum possible mutual information between the agent's actions and the state observations $n$ steps in the future, written $\mathfrak{E}_n(s)$. This notion requires an arbitrary choice of horizon, failing to account for the agent's discount rate $\gamma$. "In a discrete deterministic world empowerment reduces to the logarithm of the number of sensor states reachable with the available actions" [Salge et al., 2014]. Figure 9 demonstrates how empowerment can return counterintuitive verdicts with respect to the agent's control over the future.

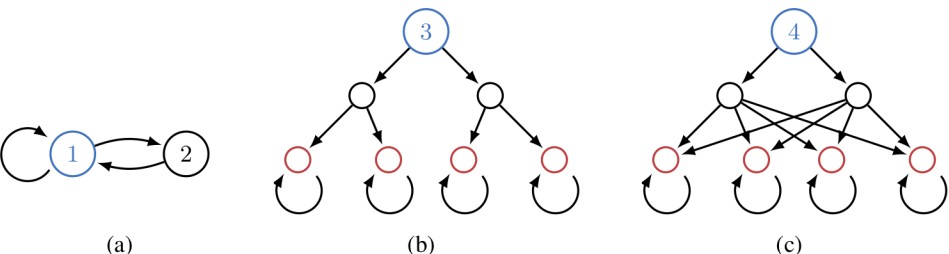

(a)                    (b)                    (c)

Figure 9: Proposed empowerment measures fail to adequately capture how future choice is affected by present actions. In a: $\mathfrak{E}_n(s_1)$ varies depending on whether $n$ is even; thus, $\lim_{n\to\infty} \mathfrak{E}_n(s_1)$ does not exist. In b and c: $\forall n : \mathfrak{E}_n(s_3) = \mathfrak{E}_n(s_4)$, even though $s_4$ allows greater control over future state trajectories than $s_3$ does. For example, suppose that in both b and c, the leftmost black state and the rightmost red state have 1 reward while all other states have 0 reward. In c, the agent can independently maximize the intermediate black-state reward and the delayed red-state reward. Independent maximization is not possible in b.

POWER returns intuitive answers in these situations. $\lim_{\gamma\to 1} \text{POWER}_{\mathcal{D}_{\text{bound}}}(s_1, \gamma)$ converges by lemma 5.3. Consider the obvious involution $\phi$ which takes each state in fig. 9b to its counterpart in fig. 9c. Since $\phi \cdot \mathcal{F}_{\text{nd}}(s_3) \subsetneq \mathcal{F}_{\text{nd}}(s_4) = \mathcal{F}(s_4)$, proposition 6.6 proves that $\forall \gamma \in [0,1]$ : $\text{POWER}_{\mathcal{D}_{\text{bound}}}(s_3, \gamma) \leq_{\text{most: } \mathfrak{D}_{\text{bound}}} \text{POWER}_{\mathcal{D}_{\text{bound}}}(s_4, \gamma)$, with the proof of proposition 6.6 showing strict inequality under all $\mathcal{D}_{X\text{-IID}}$ when $\gamma \in (0,1)$.

Empowerment can be adjusted to account for these cases, perhaps by considering the channel capacity between the agent's actions and the state trajectories induced by stationary policies. However, since POWER is formulated in terms of optimal value, we believe that POWER is better suited for MDPs than information-theoretic empowerment is.

## Appendix B  Seeking POWER can be a detour

**Remark.** The results of appendix E do not depend on this section's results.

One might suspect that optimal policies tautologically tend to seek POWER. This intuition is wrong.

**Proposition B.1** (Greater $\text{POWER}_{\mathcal{D}_{\text{bound}}}$ does not imply greater $\mathbb{P}_{\mathcal{D}_{\text{bound}}}$)**.**
*Action $a$ seeking more $\text{POWER}_{\mathcal{D}_{bound}}$ than $a'$ at state $s$ and $\gamma$ does not imply that $\mathbb{P}_{\mathcal{D}_{bound}}(s, a, \gamma) \geq \mathbb{P}_{\mathcal{D}_{bound}}(s, a', \gamma)$.*

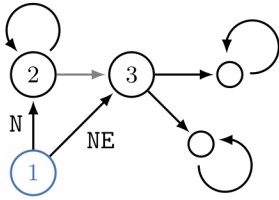

Figure 10

*Proof.* Consider the environment of fig. 10. Let $X_u := \text{unif}(0,1)$, and consider $\mathcal{D}_{X_u\text{-IID}}$, which has bounded support. Direct computation[5] of the POWER expectation (definition 5.2) yields $\text{POWER}_{\mathcal{D}_{X_u\text{-IID}}}(s_2, 1) = \frac{3}{4} > \frac{2}{3} = \text{POWER}_{\mathcal{D}_{X_u\text{-IID}}}(s_3, 1)$. Therefore, N seeks more $\text{POWER}_{\mathcal{D}_{X_u\text{-IID}}}$ than NE at state $s_1$ and $\gamma = 1$.

However, $\mathbb{P}_{\mathcal{D}_{X_u\text{-IID}}}(s_1, \text{N}, 1) = \frac{1}{3} < \frac{2}{3} = \mathbb{P}_{\mathcal{D}_{X_u\text{-IID}}}(s_1, \text{NE}, 1)$. □

---

[5]In small deterministic MDPs, the POWER and optimality probability of the maximum-entropy reward function distribution can be computed using https://github.com/loganriggs/Optimal-Policies-Tend-To-Seek-Power.

**Lemma B.2** (Fraction of orbits which agree on weak optimality)**.** *Let $\mathfrak{D} \subseteq \Delta(\mathbb{R}^{|\mathcal{S}|})$, and suppose $f_1, f_2 : \Delta(\mathbb{R}^{|\mathcal{S}|}) \to \mathbb{R}$ are such that $f_1(\mathcal{D}) \geq_{\text{most: } \mathfrak{D}} f_2(\mathcal{D})$. Then for all $\mathcal{D} \in \mathfrak{D}$,*

$$\frac{\left|\left\{\mathcal{D}' \in S_{|\mathcal{S}|} \cdot \mathcal{D} \mid f_1(\mathcal{D}') \geq f_2(\mathcal{D}')\right\}\right|}{\left|S_{|\mathcal{S}|} \cdot \mathcal{D}\right|} \geq \frac{1}{2}.$$

*Proof.* All $\mathcal{D}' \in S_{|\mathcal{S}|} \cdot \mathcal{D}$ such that $f_1(\mathcal{D}') = f_2(\mathcal{D}')$ satisfy $f_1(\mathcal{D}') \geq f_2(\mathcal{D}')$.

Otherwise, consider the $\mathcal{D}' \in S_{|\mathcal{S}|} \cdot \mathcal{D}$ such that $f_1(\mathcal{D}') \neq f_2(\mathcal{D}')$. By the definition of $\geq_{\text{most}}$ (definition 6.5), at least $\frac{1}{2}$ of these $\mathcal{D}'$ satisfy $f_1(\mathcal{D}') > f_2(\mathcal{D}')$, in which case $f_1(\mathcal{D}') \geq f_2(\mathcal{D}')$. Then the desired inequality follows. □

**Lemma B.3** ($\geq_{\text{most}}$ and trivial orbits)**.** *Let $\mathfrak{D} \subseteq \Delta(\mathbb{R}^{|\mathcal{S}|})$ and suppose $f_1(\mathcal{D}) \geq_{\text{most: } \mathfrak{D}} f_2(\mathcal{D})$. For all reward function distributions $\mathcal{D} \in \mathfrak{D}$ with one-element orbits, $f_1(\mathcal{D}) \geq f_2(\mathcal{D})$. In particular, $\mathcal{D}$ has a one-element orbit when it distributes reward identically and independently (IID) across states.*

*Proof.* By lemma B.2, at least half of the elements $\mathcal{D}' \in S_{|\mathcal{S}|} \cdot \mathcal{D}$ satisfy $f_1(\mathcal{D}') \geq f_2(\mathcal{D}')$. But $\left|S_{|\mathcal{S}|} \cdot \mathcal{D}\right| = 1$, and so $f_1(\mathcal{D}) \geq f_2(\mathcal{D})$ must hold.

If $\mathcal{D}$ is IID, it has a one-element orbit due to the assumed identical distribution of reward. □

**Proposition B.4** (Actions which tend to seek POWER do not necessarily tend to be optimal)**.** *Action $a$ tending to seek more POWER than $a'$ at state $s$ and $\gamma$ does not imply that $\mathbb{P}_{\mathcal{D}_{any}}(s, a, \gamma) \geq_{\text{most: } \mathfrak{D}_{any}} \mathbb{P}_{\mathcal{D}_{any}}(s, a', \gamma)$.*

*Proof.* Consider the environment of fig. 10. Since $\text{RSD}_{\text{nd}}(s_3) \subsetneq \text{RSD}(s_2)$, proposition 6.12 shows that $\text{POWER}_{\mathcal{D}_{\text{bound}}}(s_2, 1) \geq_{\text{most: } \mathfrak{D}_{\text{bound}}} \text{POWER}_{\mathcal{D}_{\text{bound}}}(s_3, 1)$ via $s' := s_3, s := s_2, \phi$ the identity permutation (which is an involution). Therefore, N tends to seek more POWER than NE at state $s_1$ and $\gamma = 1$.

If $\mathbb{P}_{\mathcal{D}_{any}}(s_1, \text{N}, 1) \geq_{\text{most: } \mathfrak{D}_{any}} \mathbb{P}_{\mathcal{D}_{any}}(s_1, \text{NE}, 1)$, then lemma B.3 shows that $\mathbb{P}_{\mathcal{D}_{X\text{-IID}}}(s_1, \text{N}, 1) \geq \mathbb{P}_{\mathcal{D}_{X\text{-IID}}}(s_1, \text{NE}, 1)$ for all $\mathcal{D}_{X\text{-IID}}$. But the proof of proposition B.1 showed that $\mathbb{P}_{\mathcal{D}_{X_u\text{-IID}}}(s_1, \text{N}, 1) < \mathbb{P}_{\mathcal{D}_{X_u\text{-IID}}}(s_1, \text{NE}, 1)$ for $X_u := \text{unif}(0, 1)$. Therefore, it cannot be true that $\mathbb{P}_{\mathcal{D}_{any}}(s_1, \text{N}, 1) \geq_{\text{most: } \mathfrak{D}_{any}} \mathbb{P}_{\mathcal{D}_{any}}(s_1, \text{NE}, 1)$. □

## Appendix C  Sub-optimal POWER

In certain situations, POWER returns intuitively surprising verdicts. There exists a policy under which the reader chooses a winning lottery ticket, but it seems wrong to say that the reader has the power to win the lottery with high probability. For various reasons, humans and other bounded agents are generally incapable of computing optimal policies for arbitrary objectives. More formally, consider the rewardless MDP of fig. 11.

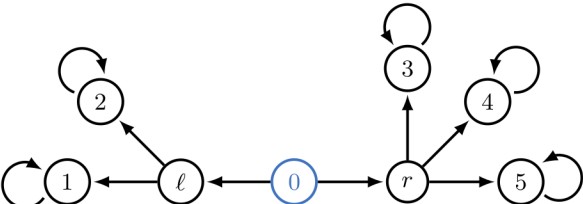

Figure 11: $s_0$ is the starting state, and $|\mathcal{A}| = 10^{10^{10}}$. At $s_0$, half of the actions lead to $s_\ell$, while the other half lead to $s_r$. Similarly, half of the actions at $s_\ell$ lead to $s_1$, while the other half lead to $s_2$. At $s_r$, one action leads to $s_3$, one action leads to $s_4$, and the remaining $10^{10^{10}} - 2$ actions lead to $s_5$.

Consider a model-based RL agent with black-box simulator access to this environment. The agent has no prior information about the model, and so it acts randomly. Before long, the agent has probably

learned how to navigate from $s_0$ to states $s_\ell$, $s_r$, $s_1$, $s_2$, and $s_5$. However, over any reasonable timescale, it is extremely improbable that the agent discovers the two actions respectively leading to $s_3$ and $s_4$.

Even provided with a reward function $R$ and the discount rate $\gamma$, the agent has yet to learn the relevant environmental dynamics, and so many of its policies are far from optimal. Although proposition 6.6 shows that $\forall \gamma \in [0,1] : \text{POWER}_{\mathcal{D}_{\text{bound}}}(s_\ell, \gamma) \leq_{\text{most: } \mathfrak{D}_{\text{bound}}} \text{POWER}_{\mathcal{D}_{\text{bound}}}(s_r, \gamma)$, there is a sense in which $s_\ell$ gives this agent more power.

We formalize a bounded agent's goal-achievement capabilities with a function pol, which takes as input a reward function and a discount rate, and returns a policy. Informally, this is the best policy which the agent knows about. We can then calculate $\text{POWER}_{\mathcal{D}_{\text{bound}}}$ with respect to pol.

**Definition C.1** (Suboptimal POWER). Let $\Pi_\Delta$ be the set of stationary stochastic policies, and let $\text{pol} : \mathbb{R}^{\mathcal{S}} \times [0,1] \to \Pi_\Delta$. For $\gamma \in [0,1]$,

$$\text{POWER}^{\text{pol}}_{\mathcal{D}_{\text{bound}}}(s, \gamma) := \mathop{\mathbb{E}}_{\substack{R \sim \mathcal{D}_{\text{bound}}, \\ a \sim \text{pol}(R,\gamma)(s), \\ s' \sim T(s,a)}} \left[ \lim_{\gamma^* \to \gamma} (1 - \gamma^*) V^{\text{pol}(R,\gamma)}_R (s', \gamma^*) \right]. \tag{5}$$

By lemma E.38, $\text{POWER}_{\mathcal{D}_{\text{bound}}}$ is the special case where $\forall R \in \mathbb{R}^{\mathcal{S}}, \gamma \in [0,1] : \text{pol}(R, \gamma) \in \Pi^*(R, \gamma)$. We define $\text{POWER}^{\text{pol}}_{\mathcal{D}_{\text{bound}}}$-seeking similarly as in definition 5.6.

$\text{POWER}^{\text{pol}}_{\mathcal{D}_{\text{bound}}}(s_0, 1)$ increases as the policies returned by pol are improved. We illustrate this by considering the $\mathcal{D}_{X\text{-IID}}$ case.

pol$_1$ The model is initially unknown, and so $\forall R, \gamma : \text{pol}_1(R, \gamma)$ is a uniformly random policy. Since pol$_1$ is constant on its inputs, $\text{POWER}^{\text{pol}_1}_{\mathcal{D}_{X\text{-IID}}}(s_0, 1) = \mathbb{E}[X]$ by the linearity of expectation and the fact that $\mathcal{D}_{X\text{-IID}}$ distributes reward independently and identically across states.

pol$_2$ The agent knows the dynamics, except that it does not know how to reach $s_3$ or $s_4$. At this point, $\text{pol}_2(R, 1)$ navigates from $s_0$ to the average-optimal choice among three terminal states: $s_1$, $s_2$, and $s_5$. Therefore, $\text{POWER}^{\text{pol}_2}_{\mathcal{D}_{\text{bound}}}(s_0, 1) = \mathbb{E}[\max \text{ of 3 draws from } X]$.

pol$_3$ The agent knows the dynamics, the environment is small enough to solve explicitly, and so $\forall R, \gamma : \text{pol}_3(R, \gamma)$ is an optimal policy. $\text{pol}_3(R, 1)$ navigates from $s_0$ to the average-optimal choice among all five terminal states. Therefore, $\text{POWER}^{\text{pol}_3}_{\mathcal{D}_{\text{bound}}}(s_0, 1) = \mathbb{E}[\max \text{ of 5 draws from } X]$.

As the agent learns more about the environment and improves pol, the agent's $\text{POWER}^{\text{pol}}_{\mathcal{D}_{\text{bound}}}$ increases. The agent seeks $\text{POWER}^{\text{pol}_2}_{\mathcal{D}_{\text{bound}}}$ by navigating to $s_\ell$ instead of $s_r$, but seeks more $\text{POWER}_{\mathcal{D}_{\text{bound}}}$ by navigating to $s_r$ instead of $s_\ell$. Intuitively, bounded agents gain power by improving pol and by formally seeking $\text{POWER}^{\text{pol}}_{\mathcal{D}_{\text{bound}}}$ within the environment.

## Appendix D   Lists of results

## List of definitions

## List of theorems

## D.1 Contributions of independent interest

We developed new basic MDP theory by exploring the structural properties of visit distribution functions. Echoing Wang et al. [2007, 2008], we believe that this area is interesting and underexplored.

### D.1.1 Optimal value theory

Lemma E.40 shows that $f(\gamma^*) := \lim_{\gamma^* \to \gamma}(1 - \gamma^*)V_R^*(s, \gamma^*)$ is Lipschitz continuous on $\gamma \in [0, 1]$, with Lipschitz constant depending only on $\|R\|_1$. For all states $s$ and policies $\pi \in \Pi$, corollary E.5 shows that $V_R^\pi(s, \gamma)$ is rational on $\gamma$.

Optimal value has a well-known dual formulation: $V_R^*(s, \gamma) = \max_{\mathbf{f} \in \mathcal{F}(s)} \mathbf{f}(\gamma)^\top \mathbf{r}$.

**Lemma E.34** ($\forall \gamma \in [0, 1) : V_R^*(s, \gamma) = \max_{\mathbf{f} \in \mathcal{F}_{\mathrm{nd}}(s)} \mathbf{f}(\gamma)^\top \mathbf{r}$).

In a fixed rewardless MDP, lemma E.34 may enable more efficient computation of optimal value functions for multiple reward functions.

### D.1.2 Optimal policy theory

Proposition E.30 demonstrates how to preserve optimal incentives while changing the discount rate.

**Proposition E.30** (How to transfer optimal policy sets across discount rates). *Suppose reward function $R$ has optimal policy set $\Pi^*(R, \gamma)$ at discount rate $\gamma \in (0, 1)$. For any $\gamma^* \in (0, 1)$, we can construct a reward function $R'$ such that $\Pi^*(R', \gamma^*) = \Pi^*(R, \gamma)$. Furthermore, $V_{R'}^*(\cdot, \gamma^*) = V_R^*(\cdot, \gamma)$.*

### D.1.3 Visit distribution theory

While Regan and Boutilier [2010] consider a visit distribution function $\mathbf{f} \in \mathcal{F}(s)$ to be non-dominated if it is optimal for some reward function in a set $\mathcal{R} \subseteq \mathbb{R}^{|\mathcal{S}|}$, our stricter definition 3.6 considers $\mathbf{f}$ to be non-dominated when $\exists \mathbf{r} \in \mathbb{R}^{|\mathcal{S}|}, \gamma \in (0, 1) : \mathbf{f}(\gamma)^\top \mathbf{r} > \max_{\mathbf{f}' \in \mathcal{F}(s) \setminus \{\mathbf{f}\}} \mathbf{f}'(\gamma)^\top \mathbf{r}$.

## Appendix E    Theoretical results

**Lemma E.1** (A policy is optimal iff it induces an optimal visit distribution at every state). *Let $\gamma \in (0, 1)$ and let $R$ be a reward function. $\pi \in \Pi^*(R, \gamma)$ iff $\pi$ induces an optimal visit distribution at every state.*

*Proof.* By definition, a policy $\pi$ is optimal iff $\pi$ induces the maximal on-policy value at each state, which is true iff $\pi$ induces an optimal visit distribution at every state (by the dual formulation of optimal value functions). □

**Definition E.2** (Transition matrix induced by a policy). $\mathbf{T}^\pi$ is the transition matrix induced by policy $\pi \in \Pi$, where $\mathbf{T}^\pi \mathbf{e}_s := T(s, \pi(s))$. $(\mathbf{T}^\pi)^t \mathbf{e}_s$ gives the probability distribution over the states visited at time step $t$, after following $\pi$ for $t$ steps from $s$.

**Proposition E.3** (Properties of visit distribution functions). *Let $s, s' \in \mathcal{S}, \mathbf{f}^{\pi,s} \in \mathcal{F}(s)$.*

    *1. $\mathbf{f}^{\pi,s}(\gamma)$ is element-wise non-negative and element-wise monotonically increasing on $\gamma \in [0, 1)$.*

2. $\forall \gamma \in [0,1) : \left\| \mathbf{f}^{\pi,s}(\gamma) \right\|_1 = \frac{1}{1-\gamma}$.

*Proof.* Item 1: by examination of definition 3.3, $\mathbf{f}^{\pi,s} = \sum_{t=0}^{\infty} \left(\gamma \mathbf{T}^{\pi}\right)^t \mathbf{e}_s$. Since each $\left(\mathbf{T}^{\pi}\right)^t$ is left stochastic and $\mathbf{e}_s$ is the standard unit vector, each entry in each summand is non-negative. Therefore, $\forall \gamma \in [0,1) : \mathbf{f}^{\pi,s}(\gamma)^{\top} \mathbf{e}_{s'} \geq 0$, and this function monotonically increases on $\gamma$.

Item 2:

$$\left\| \mathbf{f}^{\pi,s}(\gamma) \right\|_1 = \left\| \sum_{t=0}^{\infty} \left(\gamma \mathbf{T}^{\pi}\right)^t \mathbf{e}_s \right\|_1 \tag{6}$$

$$= \sum_{t=0}^{\infty} \gamma^t \left\| \left(\mathbf{T}^{\pi}\right)^t \mathbf{e}_s \right\|_1 \tag{7}$$

$$= \sum_{t=0}^{\infty} \gamma^t \tag{8}$$

$$= \frac{1}{1-\gamma}. \tag{9}$$

Equation (7) follows because all entries in each $\left(\mathbf{T}^{\pi}\right)^t \mathbf{e}_s$ are non-negative by item 1. Equation (8) follows because each $\left(\mathbf{T}^{\pi}\right)^t$ is left stochastic and $\mathbf{e}_s$ is a stochastic vector, and so $\left\| \left(\mathbf{T}^{\pi}\right)^t \mathbf{e}_s \right\|_1 = 1$. $\qquad\square$

**Lemma E.4** ($\mathbf{f} \in \mathcal{F}(s)$ is multivariate rational on $\gamma$)**.** $\mathbf{f}^{\pi} \in \mathcal{F}(s)$ *is a multivariate rational function on* $\gamma \in [0,1)$.

*Proof.* Let $\mathbf{r} \in \mathbb{R}^{|\mathcal{S}|}$ and consider $\mathbf{f}^{\pi} \in \mathcal{F}(s)$. Let $\mathbf{v}_R^{\pi}$ be the $V_R^{\pi}(s,\gamma)$ function in column vector form, with one entry per state value.

By the Bellman equations, $\mathbf{v}_R^{\pi} = (\mathbf{I} - \gamma \mathbf{T}^{\pi})^{-1} \mathbf{r}$. Let $\mathbf{A}_{\gamma} := (\mathbf{I} - \gamma \mathbf{T}^{\pi})^{-1}$, and for state $s$, form $\mathbf{A}_{s,\gamma}$ by replacing $\mathbf{A}_{\gamma}$'s column for state $s$ with $\mathbf{r}$. As noted by Lippman [1968], by Cramer's rule, $V_R^{\pi}(s,\gamma) = \frac{\det \mathbf{A}_{s,\gamma}}{\det \mathbf{A}_{\gamma}}$ is a rational function with numerator and denominator having degree at most $|\mathcal{S}|$.

In particular, for each state indicator reward function $\mathbf{e}_{s_i}$, $V_{s_i}^{\pi}(s,\gamma) = \mathbf{f}^{\pi,s}(\gamma)^{\top} \mathbf{e}_{s_i}$ is a rational function of $\gamma$ whose numerator and denominator each have degree at most $|\mathcal{S}|$. This implies that $\mathbf{f}^{\pi}(\gamma)$ is multivariate rational on $\gamma \in [0,1)$. $\qquad\square$

**Corollary E.5** (On-policy value is rational on $\gamma$)**.** *Let* $\pi \in \Pi$ *and* $R$ *be any reward function.* $V_R^{\pi}(s,\gamma)$ *is rational on* $\gamma \in [0,1)$.

*Proof.* $V_R^{\pi}(s,\gamma) = \mathbf{f}^{\pi,s}(\gamma)^{\top} \mathbf{r}$, and $\mathbf{f}$ is a multivariate rational function of $\gamma$ by lemma E.4. Therefore, for fixed $\mathbf{r}$, $\mathbf{f}^{\pi,s}(\gamma)^{\top} \mathbf{r}$ is a rational function of $\gamma$. $\qquad\square$

### E.1 Non-dominated visit distribution functions

**Definition E.6** (Continuous reward function distribution)**.** Results with $\mathcal{D}_{\text{cont}}$ hold for any absolutely continuous reward function distribution.

**Remark.** We assume $\mathbb{R}^{|\mathcal{S}|}$ is endowed with the standard topology.

**Lemma E.7** (Distinct linear functionals disagree almost everywhere on their domains)**.** *Let* $\mathbf{x}, \mathbf{x}' \in \mathbb{R}^{|\mathcal{S}|}$ *be distinct.* $\mathbb{P}_{\mathbf{r} \sim \mathcal{D}_{\text{cont}}} \left( \mathbf{x}^{\top} \mathbf{r} = \mathbf{x}'^{\top} \mathbf{r} \right) = 0$.

*Proof.* $\left\{ \mathbf{r} \in \mathbb{R}^{|\mathcal{S}|} \mid (\mathbf{x} - \mathbf{x}')^{\top} \mathbf{r} = 0 \right\}$ is a hyperplane since $\mathbf{x} - \mathbf{x}' \neq \mathbf{0}$. Therefore, it has no interior in the standard topology on $\mathbb{R}^{|\mathcal{S}|}$. Since this empty-interior set is also convex, it has zero Lebesgue measure. By the Radon-Nikodym theorem, it has zero measure under any continuous distribution $\mathcal{D}_{\text{cont}}$. $\qquad\square$

**Corollary E.8** (Unique maximization of almost all vectors). *Let $X \subsetneq \mathbb{R}^{|\mathcal{S}|}$ be finite.* $\mathbb{P}_{\mathbf{r} \sim \mathcal{D}_{cont}} \left( \left| \arg\max_{\mathbf{x}'' \in X} \mathbf{x}''^{\top} \mathbf{r} \right| > 1 \right) = 0.$

*Proof.* Let $\mathbf{x}, \mathbf{x}' \in X$ be distinct. For any $\mathbf{r} \in \mathbb{R}^{|\mathcal{S}|}$, $\mathbf{x}, \mathbf{x}' \in \arg\max_{\mathbf{x}'' \in X} \mathbf{x}''^{\top} \mathbf{r}$ iff $\mathbf{x}^{\top} \mathbf{r} = \mathbf{x}'^{\top} \mathbf{r} \geq \max_{\mathbf{x}'' \in X \setminus \{\mathbf{x}, \mathbf{x}'\}} \mathbf{x}''^{\top} \mathbf{r}$. By lemma E.7, $\mathbf{x}^{\top} \mathbf{r} = \mathbf{x}'^{\top} \mathbf{r}$ holds with probability 0 under any $\mathcal{D}_{\text{cont}}$. $\square$

### E.1.1 Generalized non-domination results

Our formalism includes both $\mathcal{F}_{\text{nd}}(s)$ and $\text{RSD}_{\text{nd}}(s)$; we therefore prove results that are applicable to both.

**Definition E.9** (Non-dominated linear functionals). Let $X \subsetneq \mathbb{R}^{|\mathcal{S}|}$ be finite. $\text{ND}(X) := \left\{ \mathbf{x} \in X \mid \exists \mathbf{r} \in \mathbb{R}^{|\mathcal{S}|} : \mathbf{x}^{\top} \mathbf{r} > \max_{\mathbf{x}' \in X \setminus \{\mathbf{x}\}} \mathbf{x}'^{\top} \mathbf{r} \right\}.$

**Lemma E.10** (All vectors are maximized by a non-dominated linear functional). *Let $\mathbf{r} \in \mathbb{R}^{|\mathcal{S}|}$ and let $X \subsetneq \mathbb{R}^{|\mathcal{S}|}$ be finite and non-empty. $\exists \mathbf{x}^* \in \text{ND}(X) : \mathbf{x}^{*\top} \mathbf{r} = \max_{\mathbf{x} \in X} \mathbf{x}^{\top} \mathbf{r}$.*

*Proof.* Let $A(\mathbf{r} \mid X) := \arg\max_{\mathbf{x} \in X} \mathbf{x}^{\top} \mathbf{r} = \{\mathbf{x}_1, \ldots, \mathbf{x}_n\}$. Then

$$\mathbf{x}_1^{\top} \mathbf{r} = \cdots = \mathbf{x}_n^{\top} \mathbf{r} > \max_{\mathbf{x}' \in X \setminus A(\mathbf{r} | X)} \mathbf{x}'^{\top} \mathbf{r}. \tag{10}$$

In eq. (10), each $\mathbf{x}^{\top} \mathbf{r}$ expression is linear on $\mathbf{r}$. The $\max$ is piecewise linear on $\mathbf{r}$ since it is the maximum of a finite set of linear functionals. In particular, all expressions in eq. (10) are continuous on $\mathbf{r}$, and so we can find some $\delta > 0$ neighborhood $B(\mathbf{r}, \delta)$ such that $\forall \mathbf{r}' \in B(\mathbf{r}, \delta) :$ $\max_{\mathbf{x}_i \in A(\mathbf{r} | X)} \mathbf{x}_i^{\top} \mathbf{r}' > \max_{\mathbf{x}' \in X \setminus A(\mathbf{r} | X)} \mathbf{x}'^{\top} \mathbf{r}'$.

But almost all $\mathbf{r}' \in B(\mathbf{r}, \delta)$ are maximized by a unique functional $\mathbf{x}^*$ by corollary E.8; in particular, at least one such $\mathbf{r}''$ exists. Formally, $\exists \mathbf{r}'' \in B(\mathbf{r}, \delta) : \mathbf{x}^{*\top} \mathbf{r}'' > \max_{\mathbf{x}' \in X \setminus \{\mathbf{x}^*\}} \mathbf{x}'^{\top} \mathbf{r}''$. Therefore, $\mathbf{x}^* \in \text{ND}(X)$ by definition E.9.

$\mathbf{x}^{*\top} \mathbf{r}' \geq \max_{\mathbf{x}_i \in A(\mathbf{r} | X)} \mathbf{x}_i^{\top} \mathbf{r}' > \max_{\mathbf{x}' \in X \setminus A(\mathbf{r} | X)} \mathbf{x}'^{\top} \mathbf{r}'$, with the strict inequality following because $\mathbf{r}'' \in B(\mathbf{r}, \delta)$. These inequalities imply that $\mathbf{x}^* \in A(\mathbf{r} \mid X)$. $\square$

**Corollary E.11** (Maximal value is invariant to restriction to non-dominated functionals). *Let $\mathbf{r} \in \mathbb{R}^{|\mathcal{S}|}$ and let $X \subsetneq \mathbb{R}^{|\mathcal{S}|}$ be finite. $\max_{\mathbf{x} \in X} \mathbf{x}^{\top} \mathbf{r} = \max_{\mathbf{x} \in \text{ND}(X)} \mathbf{x}^{\top} \mathbf{r}$.*

*Proof.* If $X$ is empty, holds trivially. Otherwise, apply lemma E.10. $\square$

**Lemma E.12** (How non-domination containment affects optimal value). *Let $\mathbf{r} \in \mathbb{R}^{|\mathcal{S}|}$ and let $X, X' \subsetneq \mathbb{R}^{|\mathcal{S}|}$ be finite.*

  1. *If $\text{ND}(X) \subseteq X'$, then $\max_{\mathbf{x} \in X} \mathbf{x}^{\top} \mathbf{r} \leq \max_{\mathbf{x}' \in X'} \mathbf{x}'^{\top} \mathbf{r}$.*

  2. *If $\text{ND}(X) \subseteq X' \subseteq X$, then $\max_{\mathbf{x} \in X} \mathbf{x}^{\top} \mathbf{r} = \max_{\mathbf{x}' \in X'} \mathbf{x}'^{\top} \mathbf{r}$.*

*Proof.* Item 1:

$$\max_{\mathbf{x} \in X} \mathbf{x}^{\top} \mathbf{r} = \max_{\mathbf{x} \in \text{ND}(X)} \mathbf{x}^{\top} \mathbf{r} \tag{11}$$

$$\leq \max_{\mathbf{x}' \in X'} \mathbf{x}'^{\top} \mathbf{r}. \tag{12}$$

Equation (11) follows by corollary E.11. Equation (12) follows because $\text{ND}(X) \subseteq X'$.

Item 2: by item 1, $\max_{\mathbf{x} \in X} \mathbf{x}^{\top} \mathbf{r} \leq \max_{\mathbf{x}' \in X'} \mathbf{x}'^{\top} \mathbf{r}$. Since $X' \subseteq X$, we also have $\max_{\mathbf{x} \in X} \mathbf{x}^{\top} \mathbf{r} \geq \max_{\mathbf{x}' \in X'} \mathbf{x}'^{\top} \mathbf{r}$, and so equality must hold. $\square$

**Definition E.13** (Non-dominated vector functions). Let $I \subseteq \mathbb{R}$ and let $F \subsetneq \left(\mathbb{R}^{|\mathcal{S}|}\right)^I$ be a finite set of vector-valued functions on $I$. $\mathrm{ND}(F) := \left\{\mathbf{f} \in F \mid \exists \gamma \in I, \mathbf{r} \in \mathbb{R}^{|\mathcal{S}|} : \mathbf{f}(\gamma)^\top \mathbf{r} > \max_{\mathbf{f}' \in F \setminus \{\mathbf{f}\}} \mathbf{f}'(\gamma)^\top \mathbf{r}\right\}$.

**Remark.** $\mathcal{F}_{\mathrm{nd}}(s) = \mathrm{ND}\left(\mathcal{F}(s)\right)$ by definition 3.6.

**Definition E.14** (Affine transformation of visit distribution sets). For notational convenience, we define set-scalar multiplication and set-vector addition on $X \subseteq \mathbb{R}^{|\mathcal{S}|}$: for $c \in \mathbb{R}$, $cX := \left\{c\mathbf{x} \mid \mathbf{x} \in X\right\}$. For $\mathbf{a} \in \mathbb{R}^{|\mathcal{S}|}$, $X + \mathbf{a} := \left\{\mathbf{x} + \mathbf{a} \mid \mathbf{x} \in X\right\}$. Similar operations hold when $X$ is a set of vector functions $\mathbb{R} \mapsto \mathbb{R}^{|\mathcal{S}|}$.

**Lemma E.15** (Invariance of non-domination under positive affine transform).

1. *Let $X \subsetneq \mathbb{R}^{|\mathcal{S}|}$ be finite. If $\mathbf{x} \in \mathrm{ND}(X)$, then $\forall c > 0, \mathbf{a} \in \mathbb{R}^{|\mathcal{S}|} : (c\mathbf{x} + \mathbf{a}) \in \mathrm{ND}(cX + \mathbf{a})$.*

2. *Let $I \subseteq \mathbb{R}$ and let $F \subsetneq \left(\mathbb{R}^{|\mathcal{S}|}\right)^I$ be a finite set of vector-valued functions on $I$. If $\mathbf{f} \in \mathrm{ND}(F)$, then $\forall c > 0, \mathbf{a} \in \mathbb{R}^{|\mathcal{S}|} : (c\mathbf{f} + \mathbf{a}) \in \mathrm{ND}(cF + \mathbf{a})$.*

*Proof.* Item 1: Suppose $\mathbf{x} \in \mathrm{ND}(X)$ is strictly optimal for $\mathbf{r} \in \mathbb{R}^{|\mathcal{S}|}$. Then let $c > 0, \mathbf{a} \in \mathbb{R}^{|\mathcal{S}|}$ be arbitrary, and define $b := \mathbf{a}^\top \mathbf{r}$.

$$\mathbf{x}^\top \mathbf{r} > \max_{\mathbf{x}' \in X \setminus \{\mathbf{x}\}} \mathbf{x}'^\top \mathbf{r} \tag{13}$$

$$c\mathbf{x}^\top \mathbf{r} + b > \max_{\mathbf{x}' \in X \setminus \{\mathbf{x}\}} c\mathbf{x}'^\top \mathbf{r} + b \tag{14}$$

$$(c\mathbf{x} + \mathbf{a})^\top \mathbf{r} > \max_{\mathbf{x}' \in X \setminus \{\mathbf{x}\}} (c\mathbf{x}' + \mathbf{a})^\top \mathbf{r} \tag{15}$$

$$(c\mathbf{x} + \mathbf{a})^\top \mathbf{r} > \max_{\mathbf{x}'' \in (cX + \mathbf{a}) \setminus \{c\mathbf{x} + \mathbf{a}\}} \mathbf{x}''^\top \mathbf{r}. \tag{16}$$

Equation (14) follows because $c > 0$. Equation (15) follows by the definition of $b$.

Item 2: If $\mathbf{f} \in \mathrm{ND}(F)$, then by definition E.13, there exist $\gamma \in I, \mathbf{r} \in \mathbb{R}^{|\mathcal{S}|}$ such that

$$\mathbf{f}(\gamma)^\top \mathbf{r} > \max_{\mathbf{f}' \in F \setminus \{\mathbf{f}\}} \mathbf{f}'(\gamma)^\top \mathbf{r}. \tag{17}$$

Apply item 1 to conclude

$$(c\mathbf{f}(\gamma) + \mathbf{a})^\top \mathbf{r} > \max_{(c\mathbf{f}' + \mathbf{a}) \in (cF + \mathbf{a}) \setminus \{c\mathbf{f} + \mathbf{a}\}} (c\mathbf{f}'(\gamma) + \mathbf{a})^\top \mathbf{r}. \tag{18}$$

Therefore, $(c\mathbf{f} + \mathbf{a}) \in \mathrm{ND}(cF + \mathbf{a})$. $\qquad\square$

### E.1.2 Inequalities which hold under most reward function distributions

**Definition 6.5** (Inequalities which hold for most probability distributions). Let $f_1, f_2 : \Delta(\mathbb{R}^{|\mathcal{S}|}) \to \mathbb{R}$ be functions from reward function distributions to real numbers and let $\mathfrak{D} \subseteq \Delta(\mathbb{R}^{|\mathcal{S}|})$. We write $f_1(\mathcal{D}) \geq_{\text{most: } \mathfrak{D}} f_2(\mathcal{D})$ when, for *all* $\mathcal{D} \in \mathfrak{D}$, the following cardinality inequality holds:

$$\left|\{\mathcal{D}' \in S_{|\mathcal{S}|} \cdot \mathcal{D} \mid f_1(\mathcal{D}') > f_2(\mathcal{D}')\}\right| \geq \left|\{\mathcal{D}' \in S_{|\mathcal{S}|} \cdot \mathcal{D} \mid f_1(\mathcal{D}') < f_2(\mathcal{D}')\}\right|. \tag{4}$$

**Lemma E.16** (Helper lemma for demonstrating $\geq_{\text{most: } \mathfrak{D}_{\text{any}}}$). *Let $\mathfrak{D} \subseteq \Delta(\mathbb{R}^{|\mathcal{S}|})$. If $\exists \phi \in S_{|\mathcal{S}|}$ such that for all $\mathcal{D} \in \mathfrak{D}$, $f_1(\mathcal{D}) < f_2(\mathcal{D})$ implies that $f_1(\phi \cdot \mathcal{D}) > f_2(\phi \cdot \mathcal{D})$, then $f_1(\mathcal{D}) \geq_{\text{most: } \mathfrak{D}} f_2(\mathcal{D})$.*

*Proof.* Since $\phi$ does not belong to the stabilizer of $S_{|\mathcal{S}|}$, $\phi$ acts injectively on $S_{|\mathcal{S}|} \cdot \mathcal{D}$. By assumption on $\phi$, the image of $\{\mathcal{D}' \in S_{|\mathcal{S}|} \cdot \mathcal{D} \mid f_1(\mathcal{D}') < f_2(\mathcal{D}')\}$ under $\phi$ is a subset of $\{\mathcal{D}' \in S_{|\mathcal{S}|} \cdot \mathcal{D} \mid f_1(\mathcal{D}') > f_2(\mathcal{D}')\}$. Since $\phi$ is injective, $\left|\{\mathcal{D}' \in S_{|\mathcal{S}|} \cdot \mathcal{D} \mid f_1(\mathcal{D}') < f_2(\mathcal{D}')\}\right| \leq \left|\{\mathcal{D}' \in S_{|\mathcal{S}|} \cdot \mathcal{D} \mid f_1(\mathcal{D}') > f_2(\mathcal{D}')\}\right|$. $f_1(\mathcal{D}) \geq_{\text{most: } \mathfrak{D}} f_2(\mathcal{D})$ by definition 6.5. $\qquad\square$

**Lemma E.17** (A helper result for expectations of functions). *Let $B_1, \ldots, B_n \subsetneq \mathbb{R}^{|\mathcal{S}|}$ be finite and let $\mathfrak{D} \subseteq \Delta(\mathbb{R}^{|\mathcal{S}|})$. Suppose $f$ is a function of the form*

$$f\left(B_1, \ldots, B_n \mid \mathcal{D}\right) = \underset{\mathbf{r} \sim \mathcal{D}}{\mathbb{E}}\left[g\left(\max_{\mathbf{b}_1 \in B_1} \mathbf{b}_1^\top \mathbf{r}, \ldots, \max_{\mathbf{b}_n \in B_n} \mathbf{b}_n^\top \mathbf{r}\right)\right] \tag{19}$$

*for some function $g$, and that $f$ is well-defined for all $\mathcal{D} \in \mathfrak{D}$. Let $\phi$ be a state permutation. Then*

$$f\left(B_1, \ldots, B_n \mid \mathcal{D}\right) = f\left(\phi \cdot B_1, \ldots, \phi \cdot B_n \mid \phi \cdot \mathcal{D}\right). \tag{20}$$

*Proof.* Let distribution $\mathcal{D}$ have probability measure $F$, and let $\phi \cdot \mathcal{D}$ have probability measure $F_\phi$.

$$f\left(B_1, \ldots, B_n \mid \mathcal{D}\right) \tag{21}$$

$$:= \underset{\mathbf{r} \sim \mathcal{D}}{\mathbb{E}}\left[g\left(\max_{\mathbf{b}_1 \in B_1} \mathbf{b}_1^\top \mathbf{r}, \ldots, \max_{\mathbf{b}_n \in B_n} \mathbf{b}_n^\top \mathbf{r}\right)\right] \tag{22}$$

$$:= \int_{\mathbb{R}^{|\mathcal{S}|}} g\left(\max_{\mathbf{b}_1 \in B_1} \mathbf{b}_1^\top \mathbf{r}, \ldots, \max_{\mathbf{b}_n \in B_n} \mathbf{b}_n^\top \mathbf{r}\right) \mathrm{d}F(\mathbf{r}) \tag{23}$$

$$= \int_{\mathbb{R}^{|\mathcal{S}|}} g\left(\max_{\mathbf{b}_1 \in B_1} \mathbf{b}_1^\top \mathbf{r}, \ldots, \max_{\mathbf{b}_n \in B_n} \mathbf{b}_n^\top \mathbf{r}\right) \mathrm{d}F_\phi(\mathbf{P}_\phi \mathbf{r}) \tag{24}$$

$$= \int_{\mathbb{R}^{|\mathcal{S}|}} g\left(\max_{\mathbf{b}_1 \in B_1} \mathbf{b}_1^\top \left(\mathbf{P}_\phi^{-1} \mathbf{r}'\right), \ldots, \max_{\mathbf{b}_n \in B_n} \mathbf{b}_n^\top \left(\mathbf{P}_\phi^{-1} \mathbf{r}'\right)\right) \left|\det \mathbf{P}_\phi\right| \mathrm{d}F_\phi(\mathbf{r}') \tag{25}$$

$$= \int_{\mathbb{R}^{|\mathcal{S}|}} g\left(\max_{\mathbf{b}_1 \in B_1} \left(\mathbf{P}_\phi \mathbf{b}_1\right)^\top \mathbf{r}', \ldots, \max_{\mathbf{b}_n \in B_n} \left(\mathbf{P}_\phi \mathbf{b}_n\right)^\top \mathbf{r}'\right) \mathrm{d}F_\phi(\mathbf{r}') \tag{26}$$

$$= \int_{\mathbb{R}^{|\mathcal{S}|}} g\left(\max_{\mathbf{b}_1' \in \phi \cdot B_1} \mathbf{b}_1'^\top \mathbf{r}', \ldots, \max_{\mathbf{b}_n' \in \phi \cdot B_n} \mathbf{b}_n'^\top \mathbf{r}'\right) \mathrm{d}F_\phi(\mathbf{r}') \tag{27}$$

$$=: f\left(\phi \cdot B_1, \ldots, \phi \cdot B_n \mid \phi \cdot \mathcal{D}\right). \tag{28}$$

Equation (24) follows by the definition of $F_\phi$ (definition 6.3). Equation (25) follows by substituting $\mathbf{r}' := \mathbf{P}_\phi \mathbf{r}$. Equation (26) follows from the fact that all permutation matrices have unitary determinant and are orthogonal (and so $(\mathbf{P}_\phi^{-1})^\top = \mathbf{P}_\phi$). $\qquad\square$

**Definition E.18** (Support of $\mathcal{D}_{\mathrm{any}}$). Let $\mathcal{D}_{\mathrm{any}}$ be any reward function distribution. $\mathrm{supp}(\mathcal{D}_{\mathrm{any}})$ is the smallest closed subset of $\mathbb{R}^{|\mathcal{S}|}$ whose complement has measure zero under $\mathcal{D}_{\mathrm{any}}$.

**Definition E.19** (Linear functional optimality probability). For finite $A, B \subsetneq \mathbb{R}^{|\mathcal{S}|}$, the *probability under $\mathcal{D}_{any}$ that $A$ is optimal over $B$* is $p_{\mathcal{D}_{any}}\left(A \geq B\right) := \mathbb{P}_{\mathbf{r} \sim \mathcal{D}_{any}}\left(\max_{\mathbf{a} \in A} \mathbf{a}^\top \mathbf{r} \geq \max_{\mathbf{b} \in B} \mathbf{b}^\top \mathbf{r}\right)$.

**Proposition E.20** (Non-dominated linear functionals and their optimality probability). *Let $A \subsetneq \mathbb{R}^{|\mathcal{S}|}$ be finite. If $\exists b < c : [b, c]^{|\mathcal{S}|} \subseteq \mathrm{supp}(\mathcal{D}_{any})$, then $\mathbf{a} \in \mathrm{ND}(A)$ implies that $\mathbf{a}$ is strictly optimal for a set of reward functions with positive measure under $\mathcal{D}_{any}$.*

*Proof.* Suppose $\exists b < c : [b, c]^{|\mathcal{S}|} \subseteq \mathrm{supp}(\mathcal{D}_{\mathrm{any}})$. If $\mathbf{a} \in \mathrm{ND}(A)$, then let $\mathbf{r}$ be such that $\mathbf{a}^\top \mathbf{r} > \max_{\mathbf{a}' \in A \setminus \{\mathbf{a}\}} \mathbf{a}'^\top \mathbf{r}$. For $a_1 > 0, a_2 \in \mathbb{R}$, positively affinely transform $\mathbf{r}' := a_1 \mathbf{r} + a_2 \mathbf{1}$ (where $\mathbf{1} \in \mathbb{R}^{|\mathcal{S}|}$ is the all-ones vector) so that $\mathbf{r}' \in (b, c)^{|\mathcal{S}|}$.

Note that $\mathbf{a}$ is still strictly optimal for $\mathbf{r}'$:

$$\mathbf{a}^\top \mathbf{r} > \max_{\mathbf{a}' \in A \setminus \{\mathbf{a}\}} \mathbf{a}'^\top \mathbf{r} \iff \mathbf{a}^\top \mathbf{r}' > \max_{\mathbf{a}' \in A \setminus \{\mathbf{a}\}} \mathbf{a}'^\top \mathbf{r}'. \tag{29}$$

Furthermore, by the continuity of both terms on the right-hand side of eq. (29), $\mathbf{a}$ is strictly optimal for reward functions in some open neighborhood $N$ of $\mathbf{r}'$. Let $N' := N \cap (b, c)^{|\mathcal{S}|}$. $N'$ is still open in $\mathbb{R}^{|\mathcal{S}|}$ since it is the intersection of two open sets $N$ and $(b, c)^{|\mathcal{S}|}$.

$\mathcal{D}_{\mathrm{any}}$ must assign positive probability measure to all open sets in its support; otherwise, its support would exclude these zero-measure sets by definition E.18. Therefore, $\mathcal{D}_{\mathrm{any}}$ assigns positive probability to $N' \subseteq \mathrm{supp}(\mathcal{D}_{\mathrm{any}})$. $\qquad\square$

**Lemma E.21** (Expected value of similar linear functional sets)**.** *Let $A, B \subsetneq \mathbb{R}^{|\mathcal{S}|}$ be finite, let $A'$ be such that $\mathrm{ND}\,(A) \subseteq A' \subseteq A$, and let $g : \mathbb{R} \to \mathbb{R}$ be an increasing function. If $B$ contains a copy $B'$ of $A'$ via $\phi$, then*

$$\mathop{\mathbb{E}}_{\mathbf{r} \sim \mathcal{D}_{bound}} \left[ g \left( \max_{\mathbf{a} \in A} \mathbf{a}^\top \mathbf{r} \right) \right] \leq \mathop{\mathbb{E}}_{\mathbf{r} \sim \phi \cdot \mathcal{D}_{bound}} \left[ g \left( \max_{\mathbf{b} \in B} \mathbf{b}^\top \mathbf{r} \right) \right]. \tag{30}$$

*If $\mathrm{ND}\,(B) \setminus B'$ is empty, then eq. (30) is an equality. If $\mathrm{ND}\,(B) \setminus B'$ is non-empty, $g$ is strictly increasing, and $\exists b < c : (b, c)^{|\mathcal{S}|} \subseteq \mathrm{supp}(\mathcal{D}_{bound})$, then eq. (30) is strict.*

*Proof.* Because $g : \mathbb{R} \to \mathbb{R}$ is increasing, it is measurable (as is max). Therefore, the relevant expectations exist for all $\mathcal{D}_{\mathrm{bound}}$.

$$\mathop{\mathbb{E}}_{\mathbf{r} \sim \mathcal{D}_{\mathrm{bound}}} \left[ g \left( \max_{\mathbf{a} \in A} \mathbf{a}^\top \mathbf{r} \right) \right] = \mathop{\mathbb{E}}_{\mathbf{r} \sim \mathcal{D}_{\mathrm{bound}}} \left[ g \left( \max_{\mathbf{a} \in A'} \mathbf{a}^\top \mathbf{r} \right) \right] \tag{31}$$

$$= \mathop{\mathbb{E}}_{\mathbf{r} \sim \phi \cdot \mathcal{D}_{\mathrm{bound}}} \left[ g \left( \max_{\mathbf{a} \in \phi \cdot A'} \mathbf{a}^\top \mathbf{r} \right) \right] \tag{32}$$

$$= \mathop{\mathbb{E}}_{\mathbf{r} \sim \phi \cdot \mathcal{D}_{\mathrm{bound}}} \left[ g \left( \max_{\mathbf{b} \in B'} \mathbf{b}^\top \mathbf{r} \right) \right] \tag{33}$$

$$\leq \mathop{\mathbb{E}}_{\mathbf{r} \sim \phi \cdot \mathcal{D}_{\mathrm{bound}}} \left[ g \left( \max_{\mathbf{b} \in B} \mathbf{b}^\top \mathbf{r} \right) \right]. \tag{34}$$

Equation (31) holds because $\forall \mathbf{r} \in \mathbb{R}^{|\mathcal{S}|} : \max_{\mathbf{a} \in A} \mathbf{a}^\top \mathbf{r} = \max_{\mathbf{a} \in A'} \mathbf{a}^\top \mathbf{r}$ by lemma E.12's item 2 with $X := A$, $X' := A'$. Equation (32) holds by lemma E.17. Equation (33) holds by the definition of $B'$. Furthermore, our assumption on $\phi$ guarantees that $B' \subseteq B$. Therefore, $\max_{\mathbf{b} \in B'} \mathbf{b}^\top \mathbf{r} \leq \max_{\mathbf{b} \in B} \mathbf{b}^\top \mathbf{r}$, and so eq. (34) holds by the fact that $g$ is an increasing function. Then eq. (30) holds.

If $\mathrm{ND}\,(B) \setminus B'$ is empty, then $\mathrm{ND}\,(B) \subseteq B'$. By assumption, $B' \subseteq B$. Then apply lemma E.12 item 2 with $X := B$, $X' := B'$ in order to conclude that eq. (34) is an equality. Then eq. (30) is also an equality.

Suppose that $g$ is strictly increasing, $\mathrm{ND}\,(B) \setminus B'$ is non-empty, and $\exists b < c : (b, c)^{|\mathcal{S}|} \subseteq \mathrm{supp}(\mathcal{D}_{\mathrm{bound}})$. Let $\mathbf{x} \in \mathrm{ND}\,(B) \setminus B'$.

$$\mathop{\mathbb{E}}_{\mathbf{r} \sim \phi \cdot \mathcal{D}_{\mathrm{bound}}} \left[ g \left( \max_{\mathbf{b} \in B'} \mathbf{b}^\top \mathbf{r} \right) \right] < \mathop{\mathbb{E}}_{\mathbf{r} \sim \phi \cdot \mathcal{D}_{\mathrm{bound}}} \left[ g \left( \max_{\mathbf{a} \in B' \cup \{\mathbf{x}\}} \mathbf{b}^\top \mathbf{r} \right) \right] \tag{35}$$

$$\leq \mathop{\mathbb{E}}_{\mathbf{r} \sim \phi \cdot \mathcal{D}_{\mathrm{bound}}} \left[ g \left( \max_{\mathbf{b} \in B} \mathbf{b}^\top \mathbf{r} \right) \right]. \tag{36}$$

$\mathbf{x}$ is strictly optimal for a positive-probability subset of $\mathrm{supp}(\mathcal{D}_{\mathrm{bound}})$ by proposition E.20. Since $g$ is strictly increasing, eq. (35) is strict. Therefore, we conclude that eq. (30) is strict. $\qquad\square$

**Lemma E.22** (For continuous IID distributions $\mathcal{D}_{X\text{-}\mathrm{IID}}$, $\exists b < c : (b, c)^{|\mathcal{S}|} \subseteq \mathrm{supp}(\mathcal{D}_{X\text{-}\mathrm{IID}})$)**.**

*Proof.* $\mathcal{D}_{X\text{-}\mathrm{IID}} := X^{|\mathcal{S}|}$. Since the state reward distribution $X$ is continuous, $X$ must have support on some open interval $(b, c)$. Since $\mathcal{D}_{X\text{-}\mathrm{IID}}$ is IID across states, $(b, c)^{|\mathcal{S}|} \subseteq \mathrm{supp}(\mathcal{D}_{X\text{-}\mathrm{IID}})$. $\qquad\square$

**Definition E.23** (Bounded, continuous IID reward)**.** $\mathfrak{D}_{\mathrm{C/B/IID}}$ is the set of $\mathcal{D}_{X\text{-}\mathrm{IID}}$ which equal $X^{|\mathcal{S}|}$ for some continuous, bounded-support distribution $X$ over $\mathbb{R}$.

**Lemma E.24** (Expectation superiority lemma)**.** *Let $A, B \subsetneq \mathbb{R}^{|\mathcal{S}|}$ be finite and let $g : \mathbb{R} \to \mathbb{R}$ be an increasing function. If $B$ contains a copy $B'$ of $\mathrm{ND}\,(A)$ via $\phi$, then*

$$\mathop{\mathbb{E}}_{\mathbf{r} \sim \mathcal{D}_{bound}} \left[ g \left( \max_{\mathbf{a} \in A} \mathbf{a}^\top \mathbf{r} \right) \right] \leq_{\mathrm{most:}\, \mathfrak{D}_{bound}} \mathop{\mathbb{E}}_{\mathbf{r} \sim \mathcal{D}_{bound}} \left[ g \left( \max_{\mathbf{b} \in B} \mathbf{b}^\top \mathbf{r} \right) \right]. \tag{37}$$

*Furthermore, if $g$ is strictly increasing and $\mathrm{ND}(B) \setminus \phi \cdot \mathrm{ND}(A)$ is non-empty, then eq. (37) is strict for all $\mathcal{D}_{X\text{-IID}} \in \mathfrak{D}_{\mathrm{C/B/IID}}$. In particular, $\mathbb{E}_{\mathbf{r} \sim \mathcal{D}_{bound}} \left[ g \left( \max_{\mathbf{a} \in A} \mathbf{a}^\top \mathbf{r} \right) \right] \not\geq_{\mathrm{most}: \mathfrak{D}_{bound}} \mathbb{E}_{\mathbf{r} \sim \mathcal{D}_{bound}} \left[ g \left( \max_{\mathbf{b} \in B} \mathbf{b}^\top \mathbf{r} \right) \right]$.*

*Proof.* Because $g : \mathbb{R} \to \mathbb{R}$ is increasing, it is measurable (as is max). Therefore, the relevant expectations exist for all $\mathcal{D}_{\mathrm{bound}}$.

Suppose that $\mathcal{D}_{\mathrm{bound}}$ is such that $\mathbb{E}_{\mathbf{r} \sim \mathcal{D}_{\mathrm{bound}}} \left[ g \left( \max_{\mathbf{b} \in B} \mathbf{b}^\top \mathbf{r} \right) \right] < \mathbb{E}_{\mathbf{r} \sim \mathcal{D}_{\mathrm{bound}}} \left[ g \left( \max_{\mathbf{a} \in A} \mathbf{a}^\top \mathbf{r} \right) \right]$.

$$\mathbb{E}_{\mathbf{r} \sim \phi \cdot \mathcal{D}_{\mathrm{bound}}} \left[ g \left( \max_{\mathbf{a} \in A} \mathbf{a}^\top \mathbf{r} \right) \right] \leq \mathbb{E}_{\mathbf{r} \sim \phi^2 \cdot \mathcal{D}_{\mathrm{bound}}} \left[ g \left( \max_{\mathbf{b} \in B} \mathbf{b}^\top \mathbf{r} \right) \right] \tag{38}$$

$$= \mathbb{E}_{\mathbf{r} \sim \mathcal{D}_{\mathrm{bound}}} \left[ g \left( \max_{\mathbf{b} \in B} \mathbf{b}^\top \mathbf{r} \right) \right] \tag{39}$$

$$< \mathbb{E}_{\mathbf{r} \sim \mathcal{D}_{\mathrm{bound}}} \left[ g \left( \max_{\mathbf{a} \in A} \mathbf{a}^\top \mathbf{r} \right) \right] \tag{40}$$

$$\leq \mathbb{E}_{\mathbf{r} \sim \phi \cdot \mathcal{D}_{\mathrm{bound}}} \left[ g \left( \max_{\mathbf{b} \in B} \mathbf{b}^\top \mathbf{r} \right) \right]. \tag{41}$$

Equation (38) follows by applying lemma E.21 with permutation $\phi$ and $A' := \mathrm{ND}(A)$. Equation (39) follows because involutions satisfy $\phi^{-1} = \phi$, and $\phi^2$ is therefore the identity. Equation (40) follows because we assumed that $\mathbb{E}_{\mathbf{r} \sim \mathcal{D}_{\mathrm{bound}}} \left[ g \left( \max_{\mathbf{b} \in B} \mathbf{b}^\top \mathbf{r} \right) \right] < \mathbb{E}_{\mathbf{r} \sim \mathcal{D}_{\mathrm{bound}}} \left[ g \left( \max_{\mathbf{a} \in A} \mathbf{a}^\top \mathbf{r} \right) \right]$. Equation (41) follows by applying lemma E.21 with permutation $\phi$ and and $A' := \mathrm{ND}(A)$. By lemma E.16, eq. (37) holds.

Suppose $g$ is strictly increasing and $\mathrm{ND}(B) \setminus B'$ is non-empty. Let $\phi' \in S_{|\mathcal{S}|}$.

$$\mathbb{E}_{\mathbf{r} \sim \phi' \cdot \mathcal{D}_{X\text{-IID}}} \left[ g \left( \max_{\mathbf{a} \in A} \mathbf{a}^\top \mathbf{r} \right) \right] = \mathbb{E}_{\mathbf{r} \sim \mathcal{D}_{X\text{-IID}}} \left[ g \left( \max_{\mathbf{a} \in A} \mathbf{a}^\top \mathbf{r} \right) \right] \tag{42}$$

$$< \mathbb{E}_{\mathbf{r} \sim \phi \cdot \mathcal{D}_{X\text{-IID}}} \left[ g \left( \max_{\mathbf{b} \in B} \mathbf{b}^\top \mathbf{r} \right) \right] \tag{43}$$

$$= \mathbb{E}_{\mathbf{r} \sim \phi' \cdot \mathcal{D}_{X\text{-IID}}} \left[ g \left( \max_{\mathbf{b} \in B} \mathbf{b}^\top \mathbf{r} \right) \right]. \tag{44}$$

Equation (42) and eq. (44) hold because $\mathcal{D}_{X\text{-IID}}$ distributes reward identically across states: $\forall \phi_x \in S_{|\mathcal{S}|} : \phi_x \cdot \mathcal{D}_{X\text{-IID}} = \mathcal{D}_{X\text{-IID}}$. By lemma E.22, $\exists b < c : (b, c)^{|\mathcal{S}|} \subseteq \mathrm{supp}(\mathcal{D}_{X\text{-IID}})$. Therefore, apply lemma E.21 with $A' := \mathrm{ND}(A)$ to conclude that eq. (43) holds.

Therefore, $\forall \phi' \in S_{|\mathcal{S}|} : \mathbb{E}_{\mathbf{r} \sim \phi' \cdot \mathcal{D}_{X\text{-IID}}} \left[ g \left( \max_{\mathbf{a} \in A} \mathbf{a}^\top \mathbf{r} \right) \right] < \mathbb{E}_{\mathbf{r} \sim \phi' \cdot \mathcal{D}_{X\text{-IID}}} \left[ g \left( \max_{\mathbf{b} \in B} \mathbf{b}^\top \mathbf{r} \right) \right]$, and so $\mathbb{E}_{\mathbf{r} \sim \mathcal{D}_{\mathrm{bound}}} \left[ g \left( \max_{\mathbf{a} \in A} \mathbf{a}^\top \mathbf{r} \right) \right] \not\geq_{\mathrm{most}: \mathfrak{D}_{\mathrm{bound}}} \mathbb{E}_{\mathbf{r} \sim \mathcal{D}_{\mathrm{bound}}} \left[ g \left( \max_{\mathbf{b} \in B} \mathbf{b}^\top \mathbf{r} \right) \right]$ by definition 6.5. $\square$

**Definition E.25** (Indicator function)**.** Let $L$ be a predicate which takes input $x$. $\mathbb{1}_{L(x)}$ is the function which returns 1 when $L(x)$ is true, and 0 otherwise.

**Lemma E.26** (Optimality probability inclusion relations)**.** *Let $X, Y \subsetneq \mathbb{R}^{|\mathcal{S}|}$ be finite and suppose $Y' \subseteq Y$.*

$$p_{\mathcal{D}_{any}} \left( X \geq Y \right) \leq p_{\mathcal{D}_{any}} \left( X \geq Y' \right) \leq p_{\mathcal{D}_{any}} \left( X \cup \left( Y \setminus Y' \right) \geq Y \right). \tag{45}$$

*If $\exists b < c : (b, c)^{|\mathcal{S}|} \subseteq \mathrm{supp}(\mathcal{D}_{any})$, $X \subseteq Y$, and $\mathrm{ND}(Y) \cap \left( Y \setminus Y' \right)$ is non-empty, then the second inequality is strict.*

*Proof.*

$$p_{\mathcal{D}_{\mathrm{any}}}\left(X \geq Y\right) := \mathop{\mathbb{E}}_{\mathbf{r} \sim \mathcal{D}_{\mathrm{any}}}\left[\mathbb{1}_{\max_{\mathbf{x} \in X} \mathbf{x}^\top \mathbf{r} \geq \max_{\mathbf{y} \in Y} \mathbf{y}^\top \mathbf{r}}\right] \tag{46}$$

$$\leq \mathop{\mathbb{E}}_{\mathbf{r} \sim \mathcal{D}_{\mathrm{any}}}\left[\mathbb{1}_{\max_{\mathbf{x} \in X} \mathbf{x}^\top \mathbf{r} \geq \max_{\mathbf{y} \in Y'} \mathbf{y}^\top \mathbf{r}}\right] \tag{47}$$

$$\leq \mathop{\mathbb{E}}_{\mathbf{r} \sim \mathcal{D}_{\mathrm{any}}}\left[\mathbb{1}_{\max_{\mathbf{x} \in X \cup (Y \setminus Y')} \mathbf{x}^\top \mathbf{r} \geq \max_{\mathbf{y} \in Y'} \mathbf{y}^\top \mathbf{r}}\right] \tag{48}$$

$$= \mathop{\mathbb{E}}_{\mathbf{r} \sim \mathcal{D}_{\mathrm{any}}}\left[\mathbb{1}_{\max_{\mathbf{x} \in X \cup (Y \setminus Y')} \mathbf{x}^\top \mathbf{r} \geq \max_{\mathbf{y} \in Y' \cup (Y \setminus Y')} \mathbf{y}^\top \mathbf{r}}\right] \tag{49}$$

$$= \mathop{\mathbb{E}}_{\mathbf{r} \sim \mathcal{D}_{\mathrm{any}}}\left[\mathbb{1}_{\max_{\mathbf{x} \in X \cup (Y \setminus Y')} \mathbf{x}^\top \mathbf{r} \geq \max_{\mathbf{y} \in Y} \mathbf{y}^\top \mathbf{r}}\right] \tag{50}$$

$$=: p_{\mathcal{D}_{\mathrm{any}}}\left(X \cup \left(Y \setminus Y'\right) \geq Y\right). \tag{51}$$

Equation (47) follows because $\forall \mathbf{r} \in \mathbb{R}^{|\mathcal{S}|} : \mathbb{1}_{\max_{\mathbf{x} \in X} \mathbf{x}^\top \mathbf{r} \geq \max_{\mathbf{y} \in Y} \mathbf{y}^\top \mathbf{r}} \leq \mathbb{1}_{\max_{\mathbf{x} \in X} \mathbf{x}^\top \mathbf{r} \geq \max_{\mathbf{y} \in Y'} \mathbf{y}^\top \mathbf{r}}$ since $Y' \subseteq Y$; note that eq. (47) equals $p_{\mathcal{D}_{\mathrm{any}}}\left(X \geq Y'\right)$, and so the first inequality of eq. (45) is shown. Equation (48) holds because $\forall \mathbf{r} \in \mathbb{R}^{|\mathcal{S}|} : \mathbb{1}_{\max_{\mathbf{x} \in X} \mathbf{x}^\top \mathbf{r} \geq \max_{\mathbf{y} \in Y'} \mathbf{y}^\top \mathbf{r}} \leq \mathbb{1}_{\max_{\mathbf{x} \in X \cup (Y \setminus Y')} \mathbf{x}^\top \mathbf{r} \geq \max_{\mathbf{y} \in Y'} \mathbf{b}^\top \mathbf{r}}$.

Suppose $\exists b < c : (b, c)^{|\mathcal{S}|} \subseteq \mathrm{supp}(\mathcal{D}_{\mathrm{any}})$, $X \subseteq Y$, and $\mathrm{ND}\left(Y\right) \cap \left(Y \setminus Y'\right)$ is non-empty. Let $\mathbf{y}^* \in \mathrm{ND}\left(Y\right) \cap \left(Y \setminus Y'\right)$. By proposition E.20, $\mathbf{y}^*$ is strictly optimal on a subset of $\mathrm{supp}(\mathcal{D}_{\mathrm{any}})$ with positive measure under $\mathcal{D}_{\mathrm{any}}$. In particular, for a set of $\mathbf{r}^*$ with positive measure under $\mathcal{D}_{\mathrm{any}}$, we have $\mathbf{y}^{*\top} \mathbf{r}^* > \max_{\mathbf{y} \in Y'} \mathbf{y}^\top \mathbf{r}^*$.

Then eq. (48) is strict, and therefore the second inequality of eq. (45) is strict as well. $\qquad\square$

**Lemma E.27** (Optimality probability of similar linear functional sets). *Let $A, B, C \subsetneq \mathbb{R}^{|\mathcal{S}|}$ be finite, and let $Z \subseteq \mathbb{R}^{|\mathcal{S}|}$ be such that $\mathrm{ND}\left(C\right) \subseteq Z \subseteq C$. If $\mathrm{ND}\left(A\right)$ is similar to $B' \subseteq B$ via $\phi$ such that $\phi \cdot \left(Z \setminus \left(B \setminus B'\right)\right) = Z \setminus \left(B \setminus B'\right)$, then*

$$p_{\mathcal{D}_{\mathrm{any}}}\left(A \geq C\right) \leq p_{\phi \cdot \mathcal{D}_{\mathrm{any}}}\left(B \geq C\right). \tag{52}$$

*If $B' = B$, then eq. (52) is an equality. If $\exists b < c : (b, c)^{|\mathcal{S}|} \subseteq \mathrm{supp}(\mathcal{D}_{\mathrm{any}})$, $B' \subseteq C$, and $\mathrm{ND}\left(C\right) \cap \left(B \setminus B'\right)$ is non-empty, then eq. (52) is strict.*

*Proof.*

$$p_{\mathcal{D}_{\mathrm{any}}}\left(A \geq C\right) = p_{\mathcal{D}_{\mathrm{any}}}\left(A \geq Z\right) \tag{53}$$

$$= p_{\mathcal{D}_{\mathrm{any}}}\left(\mathrm{ND}\left(A\right) \geq Z\right) \tag{54}$$

$$\leq p_{\mathcal{D}_{\mathrm{any}}}\left(\mathrm{ND}\left(A\right) \geq Z \setminus \left(B \setminus B'\right)\right) \tag{55}$$

$$= p_{\phi \cdot \mathcal{D}_{\mathrm{any}}}\left(\phi \cdot \mathrm{ND}\left(A\right) \geq \phi \cdot Z \setminus \left(B \setminus B'\right)\right) \tag{56}$$

$$= p_{\phi \cdot \mathcal{D}_{\mathrm{any}}}\left(B' \geq Z \setminus \left(B \setminus B'\right)\right) \tag{57}$$

$$\leq p_{\phi \cdot \mathcal{D}_{\mathrm{any}}}\left(B' \cup \left(B \setminus B'\right) \geq Z\right) \tag{58}$$

$$= p_{\phi \cdot \mathcal{D}_{\mathrm{any}}}\left(B \geq C\right). \tag{59}$$

Equation (53) and eq. (59) follow by lemma E.12's item 2 with $X := C, X' := Z$. Similarly, eq. (54) follows by lemma E.12's item 2 with $X := A, X' := \mathrm{ND}\left(A\right)$. Equation (55) follows by applying the first inequality of lemma E.26 with $X := \mathrm{ND}\left(A\right), Y := Z, Y' := Z \setminus \left(B \setminus B'\right)$. Equation (56) follows by applying lemma E.17 to eq. (53) with permutation $\phi$.

Equation (57) follows by our assumptions on $\phi$. Equation (58) follows because by applying the second inequality of lemma E.26 with $X := B', Y := \mathrm{ND}\left(C\right), Y' := \mathrm{ND}\left(C\right) \setminus \left(B \setminus B'\right)$.

Suppose $B' = B$. Then $B \setminus B' = \emptyset$, and so eq. (55) and eq. (58) are trivially equalities. Then eq. (52) is an equality.

Suppose $\exists b < c : (b, c)^{|\mathcal{S}|} \subseteq \mathrm{supp}(\mathcal{D}_{\mathrm{any}})$; note that $(b, c)^{|\mathcal{S}|} \subseteq \mathrm{supp}(\phi \cdot \mathcal{D}_{\mathrm{any}})$, since such support must be invariant to permutation. Further suppose that $B' \subseteq C$ and that $\mathrm{ND}\,(C) \cap (B \setminus B')$ is non-empty. Then letting $X := B', Y := Z, Y' := Z \setminus (B \setminus B')$ and noting that $\mathrm{ND}\,(\mathrm{ND}\,(Z)) = \mathrm{ND}\,(Z)$, apply lemma E.26 to eq. (58) to conclude that eq. (52) is strict. $\qquad\square$

**Lemma E.28** (Optimality probability superiority lemma). *Let $A, B, C \subsetneq \mathbb{R}^{|\mathcal{S}|}$ be finite, and let $Z$ satisfy $\mathrm{ND}\,(C) \subseteq Z \subseteq C$. If $B$ contains a copy $B'$ of $\mathrm{ND}\,(A)$ via $\phi$ such that $\phi \cdot \left( Z \setminus (B \setminus B') \right) = Z \setminus (B \setminus B')$, then $p_{\mathcal{D}_{\mathrm{any}}}(A \geq C) \leq_{\mathrm{most:}\ \mathfrak{D}_{\mathrm{any}}} p_{\mathcal{D}_{\mathrm{any}}}(B \geq C)$.*

*If $B' \subseteq C$ and $\mathrm{ND}\,(C) \cap (B \setminus B')$ is non-empty, then the inequality is strict for all $\mathcal{D}_{X\text{-IID}} \in \mathfrak{D}_{\mathrm{C/B/IID}}$ and $p_{\mathcal{D}_{\mathrm{any}}}(A \geq C) \not\geq_{\mathrm{most:}\ \mathfrak{D}_{\mathrm{any}}} p_{\mathcal{D}_{\mathrm{any}}}(B \geq C)$.*

*Proof.* Suppose $\mathcal{D}_{\mathrm{any}}$ is such that $p_{\mathcal{D}_{\mathrm{any}}}(B \geq C) < p_{\mathcal{D}_{\mathrm{any}}}(A \geq C)$.

$$p_{\phi \cdot \mathcal{D}_{\mathrm{any}}}(A \geq C) = p_{\phi^{-1} \cdot \mathcal{D}_{\mathrm{any}}}(A \geq C) \tag{60}$$

$$\leq p_{\mathcal{D}_{\mathrm{any}}}(B \geq C) \tag{61}$$

$$< p_{\mathcal{D}_{\mathrm{any}}}(A \geq C) \tag{62}$$

$$\leq p_{\phi \cdot \mathcal{D}_{\mathrm{any}}}(B \geq C). \tag{63}$$

Equation (60) holds because $\phi$ is an involution. Equation (61) and eq. (63) hold by applying lemma E.27 with permutation $\phi$. Equation (62) holds by assumption. Therefore, $p_{\mathcal{D}_{\mathrm{any}}}(A \geq C) \leq_{\mathrm{most:}\ \mathfrak{D}_{\mathrm{any}}} p_{\mathcal{D}_{\mathrm{any}}}(B \geq C)$ by lemma E.16.

Suppose $B' \subseteq C$ and $\mathrm{ND}\,(C) \cap (B \setminus B')$ is non-empty, and let $\mathcal{D}_{X\text{-IID}}$ be any continuous distribution which distributes reward independently and identically across states. Let $\phi' \in S_{|\mathcal{S}|}$.

$$p_{\phi' \cdot \mathcal{D}_{X\text{-IID}}}(A \geq C) = p_{\mathcal{D}_{X\text{-IID}}}(A \geq C) \tag{64}$$

$$< p_{\phi \cdot \mathcal{D}_{X\text{-IID}}}(B \geq C) \tag{65}$$

$$= p_{\phi' \cdot \mathcal{D}_{X\text{-IID}}}(A \geq C). \tag{66}$$

Equation (64) and eq. (66) hold because $\mathcal{D}_{X\text{-IID}}$ distributes reward identically across states, $\forall \phi_x \in S_{|\mathcal{S}|} : \phi_x \cdot \mathcal{D}_{X\text{-IID}} = \mathcal{D}_{X\text{-IID}}$. By lemma E.22, $\exists b < c : (b, c)^{|\mathcal{S}|} \subseteq \mathrm{supp}(\mathcal{D}_{X\text{-IID}})$. Therefore, apply lemma E.27 to conclude that eq. (65) holds.

Therefore, $\forall \phi' \in S_{|\mathcal{S}|} : p_{\phi' \cdot \mathcal{D}_{X\text{-IID}}}(A \geq C) < p_{\phi' \cdot \mathcal{D}_{X\text{-IID}}}(B \geq C)$. In particular, $p_{\mathcal{D}_{\mathrm{any}}}(A \geq C) \not\geq_{\mathrm{most:}\ \mathfrak{D}_{\mathrm{any}}} p_{\mathcal{D}_{\mathrm{any}}}(B \geq C)$ by definition 6.5. $\qquad\square$

**Lemma E.29** (Limit probability inequalities which hold for most distributions). *Let $I \subseteq \mathbb{R}$, let $\mathfrak{D} \subseteq \Delta(\mathbb{R}^{|\mathcal{S}|})$ be closed under permutation, and let $F_A, F_B, F_C$ be finite sets of vector functions $I \mapsto \mathbb{R}^{|\mathcal{S}|}$. Let $\gamma$ be a limit point of $I$ such that $f_1(\mathcal{D}) := \lim_{\gamma^* \to \gamma} p_{\mathcal{D}}(F_B(\gamma^*) \geq F_C(\gamma^*)), f_2(\mathcal{D}) := \lim_{\gamma^* \to \gamma} p_{\mathcal{D}}(F_A(\gamma^*) \geq F_C(\gamma^*))$ are well-defined for all $\mathcal{D} \in \mathfrak{D}$.*

*Let $F_Z$ satisfy $\mathrm{ND}\,(F_C) \subseteq F_Z \subseteq F_C$. Suppose $F_B$ contains a copy of $F_A$ via $\phi$ such that $\phi \cdot \left( F_Z \setminus (F_B \setminus \phi \cdot F_A) \right) = F_Z \setminus (F_B \setminus \phi \cdot F_A)$. Then $f_2(\mathfrak{D}) \leq_{\mathrm{most:}\ \mathfrak{D}} f_1(\mathfrak{D})$.*

*Proof.* Suppose $\mathcal{D} \in \mathfrak{D}$ is such that $f_2(\mathcal{D}) > f_1(\mathcal{D})$.

$$f_2(\phi \cdot \mathcal{D}) = f_2\left( \phi^{-1} \cdot \mathcal{D} \right) \tag{67}$$

$$:= \lim_{\gamma^* \to \gamma} p_{\phi^{-1} \cdot \mathcal{D}}\left( F_A(\gamma^*) \geq F_C(\gamma^*) \right) \tag{68}$$

$$\leq \lim_{\gamma^* \to \gamma} p_{\mathcal{D}}\left( F_B(\gamma^*) \geq F_C(\gamma^*) \right) \tag{69}$$

$$< \lim_{\gamma^* \to \gamma} p_{\mathcal{D}}\left( F_A(\gamma^*) \geq F_C(\gamma^*) \right) \tag{70}$$

$$\leq \lim_{\gamma^* \to \gamma} p_{\phi \cdot \mathcal{D}} \left( F_B(\gamma^*) \geq F_C(\gamma^*) \right) \tag{71}$$

$$=: f_1 \left( \phi \cdot \mathcal{D} \right). \tag{72}$$

By the assumption that $\mathfrak{D}$ is closed under permutation and $f_2$ is well-defined for all $\mathcal{D} \in \mathfrak{D}$, $f_2(\phi \cdot \mathcal{D})$ is well-defined. Equation (67) follows since $\phi = \phi^{-1}$ because $\phi$ is an involution. For all $\gamma^* \in I$, let $A := F_A(\gamma^*), B := F_B(\gamma^*), C := F_C(\gamma^*), Z := F_Z(\gamma^*)$ (by definition E.13, $\mathrm{ND}\,(C) \subseteq Z \subseteq C$). Since $\phi \cdot A \subseteq B$ by assumption, and since $\mathrm{ND}\,(A) \subseteq A$, $B$ also contains a copy of $\mathrm{ND}\,(A)$ via $\phi$. Furthermore, $\phi \cdot \left( Z \setminus \left( B \setminus \phi \cdot A \right) \right) = Z \setminus \left( B \setminus \phi \cdot A \right)$ (by assumption), and so apply lemma E.27 to conclude that $p_{\phi^{-1} \cdot \mathcal{D}} \left( F_A(\gamma^*) \geq F_C(\gamma^*) \right) \leq p_{\mathcal{D}} \left( F_B(\gamma^*) \geq F_C(\gamma^*) \right)$. Therefore, the limit inequality eq. (69) holds. Equation (70) follows because we assumed that $f_1(\mathcal{D}) < f_2(\mathcal{D})$. Equation (71) holds by reasoning similar to that given for eq. (69).

Therefore, $f_2(\mathcal{D}) > f_1(\mathcal{D})$ implies that $f_2 \left( \phi \cdot \mathcal{D} \right) < f_1 \left( \phi \cdot \mathcal{D} \right)$, and so apply lemma E.16 to conclude that $f_2(\mathcal{D}) \leq_{\text{most: } \mathfrak{D}} f_1(\mathcal{D})$. $\square$

### E.1.3 $\mathcal{F}_{\mathrm{nd}}$ results

**Proposition E.30** (How to transfer optimal policy sets across discount rates). *Suppose reward function $R$ has optimal policy set $\Pi^* \left( R, \gamma \right)$ at discount rate $\gamma \in (0, 1)$. For any $\gamma^* \in (0, 1)$, we can construct a reward function $R'$ such that $\Pi^* \left( R', \gamma^* \right) = \Pi^* \left( R, \gamma \right)$. Furthermore, $V_{R'}^* \left( \cdot, \gamma^* \right) = V_R^* \left( \cdot, \gamma \right)$.*

*Proof.* Let $R$ be any reward function. Suppose $\gamma^* \in (0, 1)$ and construct $R'(s) := V_R^* \left( s, \gamma \right) - \gamma^* \max_{a \in \mathcal{A}} \mathbb{E}_{s' \sim T(s,a)} \left[ V_R^* \left( s', \gamma \right) \right]$.

Let $\pi \in \Pi$ be any policy. By the definition of optimal policies, $\pi \in \Pi^* \left( R', \gamma^* \right)$ iff for all $s$:

$$R'(s) + \gamma^* \mathbb{E}_{s' \sim T\left(s, \pi(s)\right)} \left[ V_{R'}^* \left( s', \gamma^* \right) \right] = R'(s) + \gamma^* \max_{a \in \mathcal{A}} \mathbb{E}_{s' \sim T(s,a)} \left[ V_{R'}^* \left( s', \gamma^* \right) \right] \tag{73}$$

$$R'(s) + \gamma^* \mathbb{E}_{s' \sim T\left(s, \pi(s)\right)} \left[ V_R^* \left( s', \gamma \right) \right] = R'(s) + \gamma^* \max_{a \in \mathcal{A}} \mathbb{E}_{s' \sim T(s,a)} \left[ V_R^* \left( s', \gamma \right) \right] \tag{74}$$

$$\gamma^* \mathbb{E}_{s' \sim T\left(s, \pi(s)\right)} \left[ V_R^* \left( s', \gamma \right) \right] = \gamma^* \max_{a \in \mathcal{A}} \mathbb{E}_{s' \sim T(s,a)} \left[ V_R^* \left( s', \gamma \right) \right] \tag{75}$$

$$\mathbb{E}_{s' \sim T\left(s, \pi(s)\right)} \left[ V_R^* \left( s', \gamma \right) \right] = \max_{a \in \mathcal{A}} \mathbb{E}_{s' \sim T(s,a)} \left[ V_R^* \left( s', \gamma \right) \right]. \tag{76}$$

By the Bellman equations, $R'(s) = V_{R'}^* \left( s, \gamma^* \right) - \gamma^* \max_{a \in \mathcal{A}} \mathbb{E}_{s' \sim T(s,a)} \left[ V_{R'}^* \left( s', \gamma^* \right) \right]$. By the definition of $R'$, $V_{R'}^* \left( \cdot, \gamma^* \right) = V_R^* \left( \cdot, \gamma \right)$ must be the unique solution to the Bellman equations for $R'$ at $\gamma^*$. Therefore, eq. (74) holds. Equation (75) follows by plugging in $R' := V_R^* \left( s, \gamma \right) - \gamma^* \max_{a \in \mathcal{A}} \mathbb{E}_{s' \sim T(s,a)} \left[ V_R^* \left( s', \gamma \right) \right]$ to eq. (74) and doing algebraic manipulation. Equation (76) follows because $\gamma^* > 0$.

Equation (76) shows that $\pi \in \Pi^* \left( R', \gamma^* \right)$ iff $\forall s : \mathbb{E}_{s' \sim T(s, \pi(s))} \left[ V_R^* \left( s', \gamma \right) \right] = \max_{a \in \mathcal{A}} \mathbb{E}_{s' \sim T(s,a)} \left[ V_R^* \left( s', \gamma \right) \right]$. That is, $\pi \in \Pi^* \left( R', \gamma^* \right)$ iff $\pi \in \Pi^* \left( R, \gamma \right)$. $\square$

**Definition E.31** (Evaluating sets of visit distribution functions at $\gamma$). For $\gamma \in (0, 1)$, define $\mathcal{F}(s, \gamma) := \left\{ \mathbf{f}(\gamma) \mid \mathbf{f} \in \mathcal{F}(s) \right\}$ and $\mathcal{F}_{\mathrm{nd}}(s, \gamma) := \left\{ \mathbf{f}(\gamma) \mid \mathbf{f} \in \mathcal{F}_{\mathrm{nd}}(s) \right\}$. If $F \subseteq \mathcal{F}(s)$, then $F(\gamma) := \left\{ \mathbf{f}(\gamma) \mid \mathbf{f} \in F \right\}$.

**Lemma E.32** (Non-domination across $\gamma$ values for expectations of visit distributions). *Let $\Delta_d \in \Delta \left( \mathbb{R}^{|\mathcal{S}|} \right)$ be any state distribution and let $F := \left\{ \mathbb{E}_{s_d \sim \Delta_d} \left[ \mathbf{f}^{\pi, s_d} \right] \mid \pi \in \Pi \right\}$. $\mathbf{f} \in \mathrm{ND}\,(F)$ iff $\forall \gamma^* \in (0, 1) : \mathbf{f}(\gamma^*) \in \mathrm{ND}\,(F(\gamma^*))$.*

*Proof.* Let $\mathbf{f}^{\pi} \in \text{ND}(F)$ be strictly optimal for reward function $R$ at discount rate $\gamma \in (0, 1)$:

$$\mathbf{f}^{\pi}(\gamma)^{\top}\mathbf{r} > \max_{\mathbf{f}^{\pi'} \in F \setminus \{\mathbf{f}^{\pi}\}} \mathbf{f}^{\pi'}(\gamma)^{\top}\mathbf{r}. \tag{77}$$

Let $\gamma^* \in (0, 1)$. By proposition E.30, we can produce $R'$ such that $\Pi^*(R', \gamma^*) = \Pi^*(R, \gamma)$. Since the optimal policy sets are equal, lemma E.1 implies that

$$\mathbf{f}^{\pi}(\gamma^*)^{\top}\mathbf{r}' > \max_{\mathbf{f}^{\pi'} \in F \setminus \{\mathbf{f}^{\pi}\}} \mathbf{f}^{\pi'}(\gamma^*)^{\top}\mathbf{r}'. \tag{78}$$

Therefore, $\mathbf{f}^{\pi}(\gamma^*) \in \text{ND}(F(\gamma^*))$.

The reverse direction follows by the definition of $\text{ND}(F)$. $\qquad\square$

**Lemma E.33** ($\forall \gamma \in (0, 1) : \mathbf{d} \in \mathcal{F}_{\text{nd}}(s, \gamma)$ iff $\mathbf{d} \in \text{ND}(\mathcal{F}(s, \gamma))$)**.**

*Proof.* By definition E.31, $\mathcal{F}_{\text{nd}}(s, \gamma) := \left\{\mathbf{f}(\gamma) \mid \mathbf{f} \in \text{ND}(\mathcal{F}(s))\right\}$. By applying lemma E.32 with $\Delta_d := \mathbf{e}_s$, $\mathbf{f} \in \text{ND}(\mathcal{F}(s))$ iff $\forall \gamma \in (0, 1) : \mathbf{f}(\gamma) \in \text{ND}(\mathcal{F}(s, \gamma))$. $\qquad\square$

**Lemma E.34** ($\forall \gamma \in [0, 1) : V_R^*(s, \gamma) = \max_{\mathbf{f} \in \mathcal{F}_{\text{nd}}(s)} \mathbf{f}(\gamma)^{\top}\mathbf{r}$)**.**

*Proof.* $\text{ND}(\mathcal{F}(s, \gamma)) = \mathcal{F}_{\text{nd}}(s, \gamma)$ by lemma E.33, so apply corollary E.11 with $X := \mathcal{F}(s, \gamma)$. $\qquad\square$

## E.2 Some actions have greater probability of being optimal

**Lemma E.35** (Optimal policy shift bound). *For fixed $R$, $\Pi^*(R, \gamma)$ can take on at most $(2|\mathcal{S}| + 1)\sum_s \binom{|\mathcal{F}(s)|}{2}$ distinct values over $\gamma \in (0, 1)$.*

*Proof.* By lemma E.1, $\Pi^*(R, \gamma)$ changes value iff there is a change in optimality status for some visit distribution function at some state. Lippman [1968] showed that two visit distribution functions can trade off optimality status at most $2|\mathcal{S}| + 1$ times. At each state $s$, there are $\binom{|\mathcal{F}(s)|}{2}$ such pairs. $\qquad\square$

**Proposition E.36** (Optimality probability's limits exist). *Let $F \subseteq \mathcal{F}(s)$. $\mathbb{P}_{\mathcal{D}_{any}}(F, 0) = \lim_{\gamma \to 0} \mathbb{P}_{\mathcal{D}_{any}}(F, \gamma)$ and $\mathbb{P}_{\mathcal{D}_{any}}(F, 1) = \lim_{\gamma \to 1} \mathbb{P}_{\mathcal{D}_{any}}(F, \gamma)$.*

*Proof.* First consider the limit as $\gamma \to 1$. Let $\mathcal{D}_{\text{any}}$ have probability measure $F_{\text{any}}$, and define $\delta(\gamma) := F_{\text{any}}\left(\left\{R \in \mathbb{R}^{\mathcal{S}} \mid \exists \gamma^* \in [\gamma, 1) : \Pi^*(R, \gamma^*) \neq \Pi^*(R, 1)\right\}\right)$. Since $F_{\text{any}}$ is a probability measure, $\delta(\gamma)$ is bounded $[0, 1]$, and $\delta(\gamma)$ is monotone decreasing. Therefore, $\lim_{\gamma \to 1} \delta(\gamma)$ exists.

If $\lim_{\gamma \to 1} \delta(\gamma) > 0$, then there exist reward functions whose optimal policy sets $\Pi^*(R, \gamma)$ never converge (in the discrete topology on sets) to $\Pi^*(R, 1)$, contradicting lemma E.35. So $\lim_{\gamma \to 1} \delta(\gamma) = 0$.

By the definition of optimality probability (definition 4.3) and of $\delta(\gamma)$, $|\mathbb{P}_{\mathcal{D}_{\text{any}}}(F, \gamma) - \mathbb{P}_{\mathcal{D}_{\text{any}}}(F, 1)| \leq \delta(\gamma)$. Since $\lim_{\gamma \to 1} \delta(\gamma) = 0$, $\lim_{\gamma \to 1} \mathbb{P}_{\mathcal{D}_{\text{any}}}(F, \gamma) = \mathbb{P}_{\mathcal{D}_{\text{any}}}(F, 1)$.

A similar proof shows that $\lim_{\gamma \to 0} \mathbb{P}_{\mathcal{D}_{\text{any}}}(F, \gamma) = \mathbb{P}_{\mathcal{D}_{\text{any}}}(F, 0)$. $\qquad\square$

**Lemma E.37** (Optimality probability identity). *Let $\gamma \in (0, 1)$ and let $F \subseteq \mathcal{F}(s)$.*

$$\mathbb{P}_{\mathcal{D}_{any}}(F, \gamma) = p_{\mathcal{D}'}(F(\gamma) \geq \mathcal{F}(s, \gamma)) = p_{\mathcal{D}'}(F(\gamma) \geq \mathcal{F}_{nd}(s, \gamma)). \tag{79}$$

*Proof.* Let $\gamma \in (0, 1)$.

$$\mathbb{P}_{\mathcal{D}_{any}}(F, \gamma) := \mathbb{P}_{R \sim \mathcal{D}_{any}}\left(\exists \mathbf{f}^{\pi} \in F : \pi \in \Pi^*(R, \gamma)\right) \tag{80}$$

$$= \mathop{\mathbb{E}}_{\mathbf{r}\sim\mathcal{D}_{\text{any}}} \left[ \mathbb{1}_{\max_{\mathbf{f}\in F} \mathbf{f}(\gamma)^\top \mathbf{r} = \max_{\mathbf{f}'\in\mathcal{F}(s)} \mathbf{f}'(\gamma)^\top \mathbf{r}} \right] \tag{81}$$

$$= \mathop{\mathbb{E}}_{\mathbf{r}\sim\mathcal{D}_{\text{any}}} \left[ \mathbb{1}_{\max_{\mathbf{f}\in F} \mathbf{f}(\gamma)^\top \mathbf{r} = \max_{\mathbf{f}'\in\mathcal{F}_{\text{nd}}(s)} \mathbf{f}'(\gamma)^\top \mathbf{r}} \right] \tag{82}$$

$$=: p_{\mathcal{D}'} \left( F(\gamma) \geq \mathcal{F}_{\text{nd}}(s,\gamma) \right). \tag{83}$$

Equation (81) follows because lemma E.1 shows that $\pi$ is optimal iff it induces an optimal visit distribution $\mathbf{f}$ at every state. Equation (82) follows because $\forall \mathbf{r} \in \mathbb{R}^{|\mathcal{S}|} : \max_{\mathbf{f}'\in\mathcal{F}(s)} \mathbf{f}'(\gamma)^\top \mathbf{r} = \max_{\mathbf{f}'\in\mathcal{F}_{\text{nd}}(s)} \mathbf{f}'(\gamma)^\top \mathbf{r}$ by lemma E.34. $\qquad\square$

### E.3 Basic properties of POWER

**Lemma E.38** (POWER identities). *Let* $\gamma \in (0,1)$.

$$\text{POWER}_{\mathcal{D}_{bound}}(s,\gamma) = \mathop{\mathbb{E}}_{\mathbf{r}\sim\mathcal{D}_{bound}} \left[ \max_{\mathbf{f}\in\mathcal{F}_{\text{nd}}(s)} \frac{1-\gamma}{\gamma} \left( \mathbf{f}(\gamma) - \mathbf{e}_s \right)^\top \mathbf{r} \right] \tag{84}$$

$$= \frac{1-\gamma}{\gamma} \mathop{\mathbb{E}}_{\mathbf{r}\sim\mathcal{D}_{bound}} \left[ V_R^*(s,\gamma) - R(s) \right] \tag{85}$$

$$= \frac{1-\gamma}{\gamma} \left( V_{\mathcal{D}_{bound}}^*(s,\gamma) - \mathop{\mathbb{E}}_{R\sim\mathcal{D}_{bound}} \left[ R(s) \right] \right) \tag{86}$$

$$= \mathop{\mathbb{E}}_{R\sim\mathcal{D}_{bound}} \left[ \max_{\pi\in\Pi} \mathop{\mathbb{E}}_{s'\sim T\left(s,\pi(s)\right)} \left[ (1-\gamma) V_R^\pi(s',\gamma) \right] \right]. \tag{87}$$

*Proof.*

$$\text{POWER}_{\mathcal{D}_{\text{bound}}}(s,\gamma) := \mathop{\mathbb{E}}_{\mathbf{r}\sim\mathcal{D}_{\text{bound}}} \left[ \max_{\mathbf{f}\in\mathcal{F}(s)} \frac{1-\gamma}{\gamma} \left( \mathbf{f}(\gamma) - \mathbf{e}_s \right)^\top \mathbf{r} \right] \tag{88}$$

$$= \mathop{\mathbb{E}}_{\mathbf{r}\sim\mathcal{D}_{\text{bound}}} \left[ \max_{\mathbf{f}\in\mathcal{F}_{\text{nd}}(s)} \frac{1-\gamma}{\gamma} \left( \mathbf{f}(\gamma) - \mathbf{e}_s \right)^\top \mathbf{r} \right] \tag{89}$$

$$= \mathop{\mathbb{E}}_{\mathbf{r}\sim\mathcal{D}_{\text{bound}}} \left[ \max_{\mathbf{f}\in\mathcal{F}(s)} \frac{1-\gamma}{\gamma} \left( \mathbf{f}(\gamma) - \mathbf{e}_s \right)^\top \mathbf{r} \right] \tag{90}$$

$$= \frac{1-\gamma}{\gamma} \mathop{\mathbb{E}}_{\mathbf{r}\sim\mathcal{D}_{\text{bound}}} \left[ V_R^*(s,\gamma) - R(s) \right] \tag{91}$$

$$= \frac{1-\gamma}{\gamma} \left( V_{\mathcal{D}_{\text{bound}}}^*(s,\gamma) - \mathop{\mathbb{E}}_{R\sim\mathcal{D}_{\text{bound}}} \left[ R(s) \right] \right) \tag{92}$$

$$= \mathop{\mathbb{E}}_{\mathbf{r}\sim\mathcal{D}_{\text{bound}}} \left[ \max_{\pi\in\Pi} \mathop{\mathbb{E}}_{s'\sim T\left(s,\pi(s)\right)} \left[ (1-\gamma) \mathbf{f}^{\pi,s'}(\gamma)^\top \mathbf{r} \right] \right] \tag{93}$$

$$= \mathop{\mathbb{E}}_{R\sim\mathcal{D}_{\text{bound}}} \left[ \max_{\pi\in\Pi} \mathop{\mathbb{E}}_{s'\sim T\left(s,\pi(s)\right)} \left[ (1-\gamma) V_R^\pi(s',\gamma) \right] \right]. \tag{94}$$

Equation (89) follows from lemma E.34. Equation (91) follows from the dual formulation of optimal value functions. Equation (92) holds by the definition of $V_{\mathcal{D}_{\text{bound}}}^*(s,\gamma)$ (definition 5.1). Equation (93) holds because $\mathbf{f}^{\pi,s}(\gamma) = \mathbf{e}_s + \gamma \mathop{\mathbb{E}}_{s'\sim T\left(s,\pi(s)\right)} \left[ \mathbf{f}^{\pi,s'}(\gamma) \right]$ by the definition of a visit distribution function (definition 3.3). $\qquad\square$

**Definition E.39** (Discount-normalized value function). Let $\pi$ be a policy, $R$ a reward function, and $s$ a state. For $\gamma \in [0,1]$, $V_{R,\text{norm}}^\pi(s,\gamma) := \lim_{\gamma^*\to\gamma} (1-\gamma^*) V_R^\pi(s,\gamma^*)$.

**Lemma E.40** (Normalized value functions have uniformly bounded derivative). *There exists $K \geq 0$ such that for all reward functions $\mathbf{r} \in \mathbb{R}^{|\mathcal{S}|}$, $\sup_{s\in\mathcal{S},\pi\in\Pi,\gamma\in[0,1]} \left| \frac{d}{d\gamma} V^\pi_{R,\,norm}(s,\gamma) \right| \leq K \left\| \mathbf{r} \right\|_1$.*

*Proof.* Let $\pi$ be any policy, $s$ a state, and $R$ a reward function. Since $V^\pi_{R,\,\mathrm{norm}}(s,\gamma) = \lim_{\gamma^*\to\gamma}(1-\gamma^*)\mathbf{f}^{\pi,s}(\gamma^*)^\top \mathbf{r}$, $\frac{d}{d\gamma}V^\pi_{R,\,\mathrm{norm}}(s,\gamma)$ is controlled by the behavior of $\lim_{\gamma^*\to\gamma}(1-\gamma^*)\mathbf{f}^{\pi,s}(\gamma^*)$. We show that this function's gradient is bounded in infinity norm.

By lemma E.4, $\mathbf{f}^{\pi,s}(\gamma)$ is a multivariate rational function on $\gamma$. Therefore, for any state $s'$, $\mathbf{f}^{\pi,s}(\gamma)^\top \mathbf{e}_{s'} = \frac{P(\gamma)}{Q(\gamma)}$ in reduced form. By proposition E.3, $0 \leq \mathbf{f}^{\pi,s}(\gamma)^\top \mathbf{e}_{s'} \leq \frac{1}{1-\gamma}$. Thus, $Q$ may only have a root of multiplicity 1 at $\gamma = 1$, and $Q(\gamma) \neq 0$ for $\gamma \in [0,1)$. Let $f_{s'}(\gamma) := (1-\gamma)\mathbf{f}^{\pi,s}(\gamma)^\top \mathbf{e}_{s'}$.

If $Q(1) \neq 0$, then the derivative $f'_{s'}(\gamma)$ is bounded on $\gamma \in [0,1)$ because the polynomial $(1-\gamma)P(\gamma)$ cannot diverge on a bounded domain.

If $Q(1) = 0$, then factor out the root as $Q(\gamma) = (1-\gamma)Q^*(\gamma)$.

$$f'_{s'}(\gamma) = \frac{d}{d\gamma}\left( \frac{(1-\gamma)P(\gamma)}{Q(\gamma)} \right) \tag{95}$$

$$= \frac{d}{d\gamma}\left( \frac{P(\gamma)}{Q^*(\gamma)} \right) \tag{96}$$

$$= \frac{P'(\gamma)Q^*(\gamma) - (Q^*)'(\gamma)P(\gamma)}{(Q^*(\gamma))^2}. \tag{97}$$

Since $Q^*(\gamma)$ is a polynomial with no roots on $\gamma \in [0,1]$, $f'_{s'}(\gamma)$ is bounded on $\gamma \in [0,1)$.

Therefore, whether or not $Q(\gamma)$ has a root at $\gamma = 1$, $f'_{s'}(\gamma)$ is bounded on $\gamma \in [0,1)$. Furthermore, $\sup_{\gamma\in[0,1)} \left\| \nabla(1-\gamma)\mathbf{f}^{\pi,s}(\gamma) \right\|_\infty = \sup_{\gamma\in[0,1)} \max_{s'\in\mathcal{S}} \left| f'_{s'}(\gamma) \right|$ is finite since there are only finitely many states.

There are finitely many $\pi \in \Pi$, and finitely many states $s$, and so there exists some $K'$ such that $\sup_{\substack{s\in\mathcal{S},\\ \pi\in\Pi,\gamma\in[0,1)}} \left\| \nabla(1-\gamma)\mathbf{f}^{\pi,s}(\gamma) \right\|_\infty \leq K'$. Then $\left\| \nabla(1-\gamma)\mathbf{f}^{\pi,s}(\gamma) \right\|_1 \leq |\mathcal{S}| \, K' =: K$.

$$\sup_{\substack{s\in\mathcal{S},\\ \pi\in\Pi,\gamma\in[0,1)}} \left| \frac{d}{d\gamma} V^\pi_{R,\mathrm{norm}}(s,\gamma) \right| := \sup_{\substack{s\in\mathcal{S},\\ \pi\in\Pi,\gamma\in[0,1)}} \left| \frac{d}{d\gamma} \lim_{\gamma^*\to\gamma}(1-\gamma^*)V^\pi_R(s,\gamma^*) \right| \tag{98}$$

$$= \sup_{\substack{s\in\mathcal{S},\\ \pi\in\Pi,\gamma\in[0,1)}} \left| \frac{d}{d\gamma}(1-\gamma)V^\pi_R(s,\gamma) \right| \tag{99}$$

$$= \sup_{\substack{s\in\mathcal{S},\\ \pi\in\Pi,\gamma\in[0,1)}} \left| \nabla(1-\gamma)\mathbf{f}^{\pi,s}(\gamma)^\top \mathbf{r} \right| \tag{100}$$

$$\leq \sup_{\substack{s\in\mathcal{S},\\ \pi\in\Pi,\gamma\in[0,1)}} \left\| \nabla(1-\gamma)\mathbf{f}^{\pi,s}(\gamma) \right\|_1 \left\| \mathbf{r} \right\|_1 \tag{101}$$

$$\leq K \left\| \mathbf{r} \right\|_1 . \tag{102}$$

Equation (99) holds because $V^\pi_R(s,\gamma)$ is continuous on $\gamma \in [0,1)$ by corollary E.5. Equation (101) holds by the Cauchy-Schwarz inequality.

Since $\left| \frac{d}{d\gamma} V^\pi_{R,\mathrm{norm}}(s,\gamma) \right|$ is bounded for all $\gamma \in [0,1)$, eq. (102) also holds for $\gamma \to 1$. $\square$

**Lemma 5.3** (Continuity of POWER). $\mathrm{POWER}_{\mathcal{D}_{bound}}(s,\gamma)$ *is Lipschitz continuous on $\gamma \in [0,1]$.*

*Proof.* Let $b,c$ be such that $\mathrm{supp}(\mathcal{D}_{\mathrm{bound}}) \subseteq [b,c]^{|\mathcal{S}|}$. For any $\mathbf{r} \in \mathrm{supp}(\mathcal{D}_{\mathrm{bound}})$ and $\pi \in \Pi$, $V^\pi_{R,\,\mathrm{norm}}(s,\gamma)$ has Lipschitz constant $K \left\| \mathbf{r} \right\|_1 \leq K |\mathcal{S}| \left\| \mathbf{r} \right\|_\infty \leq K |\mathcal{S}| \max(|c|,|b|)$ on $\gamma \in (0,1)$ by lemma E.40.

For $\gamma \in (0,1)$, $\text{POWER}_{\mathcal{D}_{\text{bound}}}(s,\gamma) = \mathbb{E}_{R\sim\mathcal{D}_{\text{bound}}}\left[\max_{\pi\in\Pi}\mathbb{E}_{s'\sim T(s,\pi(s))}\left[(1-\gamma)V_R^\pi\left(s',\gamma\right)\right]\right]$ by eq. (94). The expectation of the maximum of a set of functions which share a Lipschitz constant, also shares the Lipschitz constant. This shows that $\text{POWER}_{\mathcal{D}_{\text{bound}}}(s,\gamma)$ is Lipschitz continuous on $\gamma \in (0,1)$. Thus, its limits are well-defined as $\gamma \to 0$ and $\gamma \to 1$. So it is Lipschitz continuous on the closed unit interval. $\square$

**Proposition 5.4** (Maximal POWER). $\text{POWER}_{\mathcal{D}_{bound}}(s,\gamma) \leq \mathbb{E}_{R\sim\mathcal{D}_{bound}}\left[\max_{s\in\mathcal{S}}R(s)\right]$, *with equality if $s$ can deterministically reach all states in one step and all states are 1-cycles.*

*Proof.* Let $\gamma \in (0,1)$.

$$\text{POWER}_{\mathcal{D}_{\text{bound}}}(s,\gamma) = \underset{R\sim\mathcal{D}_{\text{bound}}}{\mathbb{E}}\left[\max_{\pi\in\Pi}\underset{s'\sim T(s,\pi(s))}{\mathbb{E}}\left[(1-\gamma)V_R^*\left(s',\gamma\right)\right]\right] \tag{103}$$

$$\leq \underset{R\sim\mathcal{D}_{\text{bound}}}{\mathbb{E}}\left[\max_{\pi\in\Pi}\underset{s'\sim T(s,\pi(s))}{\mathbb{E}}\left[(1-\gamma)\frac{\max_{s''\in\mathcal{S}}R(s'')}{1-\gamma}\right]\right] \tag{104}$$

$$= \underset{R\sim\mathcal{D}_{\text{bound}}}{\mathbb{E}}\left[\max_{s''\in\mathcal{S}}R(s'')\right]. \tag{105}$$

Equation (103) follows from lemma E.38. Equation (104) follows because $V_R^*\left(s',\gamma\right) \leq \frac{\max_{s''\in\mathcal{S}}R(s'')}{1-\gamma}$, as no policy can do better than achieving maximal reward at each time step. Taking limits, the inequality holds for all $\gamma \in [0,1]$.

Suppose that $s$ can deterministically reach all states in one step and all states are 1-cycles. Then eq. (104) is an equality for all $\gamma \in (0,1)$, since for each $R$, the agent can select an action which deterministically transitions to a state with maximal reward. Thus the equality holds for all $\gamma \in [0,1]$. $\square$

**Lemma E.41** (Lower bound on current POWER based on future POWER).

$$\text{POWER}_{\mathcal{D}_{bound}}(s,\gamma) \geq (1-\gamma)\min_a \underset{\substack{s'\sim T(s,a),\\ R\sim\mathcal{D}_{bound}}}{\mathbb{E}}\left[R(s')\right] + \gamma\max_a \underset{s'\sim T(s,a)}{\mathbb{E}}\left[\text{POWER}_{\mathcal{D}_{bound}}\left(s',\gamma\right)\right].$$
$$\tag{106}$$

*Proof.* Let $\gamma \in (0,1)$ and let $a^* \in \arg\max_a \mathbb{E}_{s'\sim T(s,a)}\left[\text{POWER}_{\mathcal{D}_{\text{bound}}}\left(s',\gamma\right)\right]$.

$$\text{POWER}_{\mathcal{D}_{\text{bound}}}(s,\gamma) \tag{107}$$

$$= (1-\gamma)\underset{R\sim\mathcal{D}_{\text{bound}}}{\mathbb{E}}\left[\max_a \underset{s'\sim T(s,a)}{\mathbb{E}}\left[V_R^*\left(s',\gamma\right)\right]\right] \tag{108}$$

$$\geq (1-\gamma)\max_a \underset{s'\sim T(s,a)}{\mathbb{E}}\left[\underset{R\sim\mathcal{D}_{\text{bound}}}{\mathbb{E}}\left[V_R^*\left(s',\gamma\right)\right]\right] \tag{109}$$

$$= (1-\gamma)\max_a \underset{s'\sim T(s,a)}{\mathbb{E}}\left[V_{\mathcal{D}_{\text{bound}}}^*\left(s',\gamma\right)\right] \tag{110}$$

$$= (1-\gamma)\max_a \underset{s'\sim T(s,a)}{\mathbb{E}}\left[\underset{R\sim\mathcal{D}_{\text{bound}}}{\mathbb{E}}\left[R(s')\right] + \frac{\gamma}{1-\gamma}\text{POWER}_{\mathcal{D}_{\text{bound}}}\left(s',\gamma\right)\right] \tag{111}$$

$$\geq (1-\gamma)\underset{s'\sim T(s,a^*)}{\mathbb{E}}\left[\underset{R\sim\mathcal{D}_{\text{bound}}}{\mathbb{E}}\left[R(s')\right] + \frac{\gamma}{1-\gamma}\text{POWER}_{\mathcal{D}_{\text{bound}}}\left(s',\gamma\right)\right] \tag{112}$$

$$\geq (1-\gamma)\min_a \underset{\substack{s'\sim T(s,a),\\ R\sim\mathcal{D}_{\text{bound}}}}{\mathbb{E}}\left[R(s')\right] + \gamma\underset{s'\sim T(s,a^*)}{\mathbb{E}}\left[\text{POWER}_{\mathcal{D}_{\text{bound}}}\left(s',\gamma\right)\right]. \tag{113}$$

Equation (108) holds by lemma E.38. Equation (109) follows because $\mathbb{E}_{x\sim X}\left[\max_a f(a,x)\right] \geq \max_a \mathbb{E}_{x\sim X}\left[f(a,x)\right]$ by Jensen's inequality, and eq. (111) follows by lemma E.38.

The inequality also holds when we take the limits $\gamma \to 0$ or $\gamma \to 1$. $\square$

**Proposition 5.5** (POWER is smooth across reversible dynamics). *Let $\mathcal{D}_{bound}$ be bounded $[b, c]$. Suppose $s$ and $s'$ can both reach each other in one step with probability 1.*

$$\left| \text{POWER}_{\mathcal{D}_{bound}}(s, \gamma) - \text{POWER}_{\mathcal{D}_{bound}}(s', \gamma) \right| \leq (c - b)(1 - \gamma). \tag{3}$$

*Proof.* Suppose $\gamma \in [0, 1]$. First consider the case where $\text{POWER}_{\mathcal{D}_{bound}}(s, \gamma) \geq \text{POWER}_{\mathcal{D}_{bound}}(s', \gamma)$.

$$\text{POWER}_{\mathcal{D}_{bound}}(s', \gamma) \geq (1 - \gamma) \min_a \mathop{\mathbb{E}}_{\substack{s_x \sim T(s', a), \\ R \sim \mathcal{D}_{bound}}} \left[ R(s_x) \right] + \gamma \max_a \mathop{\mathbb{E}}_{s_x \sim T(s', a)} \left[ \text{POWER}_{\mathcal{D}_{bound}}(s_x, \gamma) \right] \tag{114}$$

$$\geq (1 - \gamma)b + \gamma \text{POWER}_{\mathcal{D}_{bound}}(s, \gamma). \tag{115}$$

Equation (114) follows by lemma E.41. Equation (115) follows because reward is lower-bounded by $b$ and because $s'$ can reach $s$ in one step with probability 1.

$$\left| \text{POWER}_{\mathcal{D}_{bound}}(s, \gamma) - \text{POWER}_{\mathcal{D}_{bound}}(s', \gamma) \right| = \text{POWER}_{\mathcal{D}_{bound}}(s, \gamma) - \text{POWER}_{\mathcal{D}_{bound}}(s', \gamma) \tag{116}$$

$$\leq \text{POWER}_{\mathcal{D}_{bound}}(s, \gamma) - \left( (1 - \gamma)b + \gamma \text{POWER}_{\mathcal{D}_{bound}}(s, \gamma) \right) \tag{117}$$

$$= (1 - \gamma) \left( \text{POWER}_{\mathcal{D}_{bound}}(s, \gamma) - b \right) \tag{118}$$

$$\leq (1 - \gamma) \left( \mathop{\mathbb{E}}_{R \sim \mathcal{D}_{bound}} \left[ \max_{s'' \in \mathcal{S}} R(s'') \right] - b \right) \tag{119}$$

$$\leq (1 - \gamma)(c - b). \tag{120}$$

Equation (116) follows because $\text{POWER}_{\mathcal{D}_{bound}}(s, \gamma) \geq \text{POWER}_{\mathcal{D}_{bound}}(s', \gamma)$. Equation (117) follows by eq. (115). Equation (119) follows by proposition 5.4. Equation (120) follows because reward under $\mathcal{D}_{bound}$ is upper-bounded by $c$.

The case where $\text{POWER}_{\mathcal{D}_{bound}}(s, \gamma) \leq \text{POWER}_{\mathcal{D}_{bound}}(s', \gamma)$ is similar, leveraging the fact that $s$ can also reach $s'$ in one step with probability 1. $\square$

### E.4  Seeking POWER is often more probable under optimality

#### E.4.1  Keeping options open tends to be POWER-seeking and tends to be optimal

**Definition E.42** (Normalized visit distribution function). Let $\mathbf{f} : [0, 1) \to \mathbb{R}^{|\mathcal{S}|}$ be a vector function. For $\gamma \in [0, 1]$, $\text{NORM}(\mathbf{f}, \gamma) := \lim_{\gamma^* \to \gamma}(1 - \gamma^*)\mathbf{f}(\gamma^*)$ (this limit need not exist for arbitrary $\mathbf{f}$). If $F$ is a set of such $\mathbf{f}$, then $\text{NORM}(F, \gamma) := \left\{ \text{NORM}(\mathbf{f}, \gamma) \mid \mathbf{f} \in F \right\}$.

**Remark.** $\text{RSD}(s) = \text{NORM}(\mathcal{F}(s), 1)$.

**Lemma E.43** (Normalized visit distribution functions are continuous). *Let $\Delta_s \in \Delta(\mathcal{S})$ be a state probability distribution, let $\pi \in \Pi$, and let $\mathbf{f}^* := \mathbb{E}_{s \sim \Delta_s}[\mathbf{f}^{\pi, s}]$. $\text{NORM}(\mathbf{f}^*, \gamma)$ is continuous on $\gamma \in [0, 1]$.*

*Proof.*

$$\text{NORM}(\mathbf{f}^*, \gamma) := \lim_{\gamma^* \to \gamma}(1 - \gamma^*) \mathop{\mathbb{E}}_{s \sim \Delta_s} \left[ \mathbf{f}^{\pi, s}(\gamma^*) \right] \tag{121}$$

$$= \mathop{\mathbb{E}}_{s \sim \Delta_s} \left[ \lim_{\gamma^* \to \gamma}(1 - \gamma^*)\mathbf{f}^{\pi, s}(\gamma^*) \right] \tag{122}$$

$$=: \mathop{\mathbb{E}}_{s \sim \Delta_s} \left[ \text{NORM}(\mathbf{f}^{\pi, s}, \gamma) \right]. \tag{123}$$

Equation (122) follows because the expectation is over a finite set. Each $\mathbf{f}^{\pi, s} \in \mathcal{F}(s)$ is continuous on $\gamma \in [0, 1)$ by lemma E.4, and $\lim_{\gamma^* \to 1}(1 - \gamma^*)\mathbf{f}^{\pi, s}(\gamma^*)$ exists because RSDs are well-defined [Puterman, 2014]. Therefore, each $\text{NORM}(\mathbf{f}^{\pi, s}, \gamma)$ is continuous on $\gamma \in [0, 1]$. Lastly, eq. (123)'s expectation over finitely many continuous functions is itself continuous. $\square$

**Lemma E.44** (Non-domination of normalized visit distribution functions). *Let $\Delta_s \in \Delta(\mathcal{S})$ be a state probability distribution and let $F := \{\mathbb{E}_{s \sim \Delta_s}[\mathbf{f}^{\pi,s}] \mid \pi \in \Pi\}$. For all $\gamma \in [0,1]$, $\mathrm{ND}\left(\mathrm{NORM}\left(F, \gamma\right)\right) \subseteq \mathrm{NORM}\left(\mathrm{ND}\left(F\right), \gamma\right)$, with equality when $\gamma \in (0,1)$.*

*Proof.* Suppose $\gamma \in (0,1)$.

$$\mathrm{ND}\left(\mathrm{NORM}\left(F, \gamma\right)\right) = \mathrm{ND}\left((1-\gamma)F(\gamma)\right) \tag{124}$$

$$= (1-\gamma)\mathrm{ND}\left(F(\gamma)\right) \tag{125}$$

$$= (1-\gamma)\left(\mathrm{ND}\left(F\right)(\gamma)\right) \tag{126}$$

$$= \mathrm{NORM}\left(\mathrm{ND}\left(F\right), \gamma\right). \tag{127}$$

Equation (124) and eq. (127) follow by the continuity of $\mathrm{NORM}\left(\mathbf{f}, \gamma\right)$ (lemma E.43). Equation (125) follows by lemma E.15 item 1. Equation (126) follows by lemma E.32.

Let $\gamma = 1$. Let $\mathbf{d} \in \mathrm{ND}\left(\mathrm{NORM}\left(F, 1\right)\right)$ be strictly optimal for $\mathbf{r}^* \in \mathbb{R}^{|\mathcal{S}|}$. Then let $F_{\mathbf{d}} \subseteq F$ be the subset of $\mathbf{f} \in F$ such that $\mathrm{NORM}\left(\mathbf{f}, 1\right) = \mathbf{d}$.

$$\max_{\mathbf{f} \in F_{\mathbf{d}}} \mathrm{NORM}\left(\mathbf{f}, 1\right)^\top \mathbf{r}^* > \max_{\mathbf{f}' \in F \setminus F_{\mathbf{d}}} \mathrm{NORM}\left(\mathbf{f}', 1\right)^\top \mathbf{r}^*. \tag{128}$$

Since $\mathrm{NORM}\left(\mathbf{f}, 1\right)$ is continuous at $\gamma = 1$ (lemma E.43), $\mathbf{x}^\top \mathbf{r}^*$ is continuous on $\mathbf{x} \in \mathbb{R}^{|\mathcal{S}|}$, and $F$ is finite, eq. (128) holds for some $\gamma^* \in (0,1)$ sufficiently close to $\gamma = 1$. By lemma E.10, at least one $\mathbf{f} \in F_{\mathbf{d}}$ is an element of $\mathrm{ND}\left(F(\gamma^*)\right)$. Then by lemma E.32, $\mathbf{f} \in \mathrm{ND}\left(F\right)$. We conclude that $\mathrm{ND}\left(\mathrm{NORM}\left(F, 1\right)\right) \subseteq \mathrm{NORM}\left(\mathrm{ND}\left(F\right), 1\right)$.

The case for $\gamma = 0$ proceeds similarly. $\square$

**Lemma E.45** (POWER limit identity). *Let $\gamma \in [0,1]$.*

$$\mathrm{POWER}_{\mathcal{D}_{bound}}\left(s, \gamma\right) = \mathop{\mathbb{E}}_{\mathbf{r} \sim \mathcal{D}_{bound}}\left[\max_{\mathbf{f} \in \mathcal{F}_{\mathrm{nd}}(s)} \lim_{\gamma^* \to \gamma} \frac{1-\gamma^*}{\gamma^*}\left(\mathbf{f}(\gamma^*) - \mathbf{e}_s\right)^\top \mathbf{r}\right]. \tag{129}$$

*Proof.* Let $\gamma \in [0,1]$.

$$\mathrm{POWER}_{\mathcal{D}_{bound}}\left(s, \gamma\right) = \lim_{\gamma^* \to \gamma} \mathrm{POWER}_{\mathcal{D}_{bound}}\left(s, \gamma^*\right) \tag{130}$$

$$= \lim_{\gamma^* \to \gamma} \mathop{\mathbb{E}}_{\mathbf{r} \sim \mathcal{D}_{bound}}\left[\max_{\mathbf{f} \in \mathcal{F}_{\mathrm{nd}}(s)} \frac{1-\gamma^*}{\gamma^*}\left(\mathbf{f}(\gamma^*) - \mathbf{e}_s\right)^\top \mathbf{r}\right] \tag{131}$$

$$= \mathop{\mathbb{E}}_{\mathbf{r} \sim \mathcal{D}_{bound}}\left[\lim_{\gamma^* \to \gamma} \max_{\mathbf{f} \in \mathcal{F}_{\mathrm{nd}}(s)} \frac{1-\gamma^*}{\gamma^*}\left(\mathbf{f}(\gamma^*) - \mathbf{e}_s\right)^\top \mathbf{r}\right] \tag{132}$$

$$= \mathop{\mathbb{E}}_{\mathbf{r} \sim \mathcal{D}_{bound}}\left[\max_{\mathbf{f} \in \mathcal{F}_{\mathrm{nd}}(s)} \lim_{\gamma^* \to \gamma} \frac{1-\gamma^*}{\gamma^*}\left(\mathbf{f}(\gamma^*) - \mathbf{e}_s\right)^\top \mathbf{r}\right]. \tag{133}$$

Equation (130) follows because $\mathrm{POWER}_{\mathcal{D}_{bound}}\left(s, \gamma\right)$ is continuous on $\gamma \in [0,1]$ by lemma 5.3. Equation (131) follows by lemma E.38.

For $\gamma^* \in (0,1)$, let $f_{\gamma^*}(\mathbf{r}) := \max_{\mathbf{f} \in \mathcal{F}_{\mathrm{nd}}(s)} \frac{1-\gamma^*}{\gamma^*}\left(\mathbf{f}(\gamma^*) - \mathbf{e}_s\right)^\top \mathbf{r}$. For any sequence $\gamma_n \to \gamma$, $\left(f_{\gamma_n}\right)_{n=1}^\infty$ is a sequence of functions which are piecewise linear on $\mathbf{r} \in \mathbb{R}^{|\mathcal{S}|}$, which means they are continuous and therefore measurable. Since lemma E.4 shows that each $\mathbf{f} \in \mathcal{F}_{\mathrm{nd}}(s)$ is multivariate rational on $\gamma^*$ (and therefore continuous on $\gamma^*$), $\left\{f_{\gamma_n}\right\}_{n=1}^\infty$ converges pointwise to limit function $f_\gamma$. Furthermore, $\left|V_R^*\left(s, \gamma_n\right) - R(s)\right| \leq \frac{\gamma}{1-\gamma_n}\|R\|_\infty$, and so $\left|f_{\gamma_n}(\mathbf{r})\right| = \left|\frac{1-\gamma_n}{\gamma_n}(V_R^*\left(s, \gamma_n\right) - R(s))\right| \leq g(\mathbf{r}) \leq \|\mathbf{r}\|_\infty =: g(\mathbf{r})$, which is measurable. Therefore, apply Lebesgue's dominated convergence theorem to conclude that eq. (132) holds. Equation (133) holds because $\max$ is a continuous function. $\square$

**Lemma E.46** (Lemma for POWER superiority). *Let $\Delta_1, \Delta_2 \in \Delta(\mathcal{S})$ be state probability distributions. For $i = 1, 2$, let $F_{\Delta_i} := \left\{ \gamma^{-1} \mathbb{E}_{s_i \sim \Delta_i} \left[ \mathbf{f}^{\pi, s_i} - \mathbf{e}_{s_i} \right] \mid \pi \in \Pi \right\}$. Suppose $F_{\Delta_2}$ contains a copy of $\mathrm{ND}(F_{\Delta_1})$ via $\phi$. Then $\forall \gamma \in [0, 1] : \mathbb{E}_{s_1 \sim \Delta_1} \left[ \mathrm{POWER}_{\mathcal{D}_{bound}}(s_1, \gamma) \right] \leq_{\text{most: } \mathfrak{D}_{bound}}$ $\mathbb{E}_{s_2 \sim \Delta_2} \left[ \mathrm{POWER}_{\mathcal{D}_{bound}}(s_2, \gamma) \right]$.*

*If $\mathrm{ND}(F_{\Delta_2}) \setminus \phi \cdot \mathrm{ND}(F_{\Delta_1})$ is non-empty, then for all $\gamma \in (0, 1)$, the inequality is strict for all $\mathcal{D}_{X\text{-IID}} \in \mathfrak{D}_{\text{C/B/IID}}$ and $\mathbb{E}_{s_1 \sim \Delta_1} \left[ \mathrm{POWER}_{\mathcal{D}_{bound}}(s_1, \gamma) \right] \not\geq_{\text{most: } \mathfrak{D}_{bound}} \mathbb{E}_{s_2 \sim \Delta_2} \left[ \mathrm{POWER}_{\mathcal{D}_{bound}}(s_2, \gamma) \right]$.*

*These results also hold when replacing $F_{\Delta_i}$ with $F_{\Delta_i}^* := \left\{ \mathbb{E}_{s_i \sim \Delta_i} \left[ \mathbf{f}^{\pi, s_i} \right] \mid \pi \in \Pi \right\}$ for $i = 1, 2$.*

*Proof.*

$$\phi \cdot \mathrm{ND}\left( \mathrm{NORM}(F_{\Delta_1}, \gamma) \right) \subseteq \phi \cdot \mathrm{NORM}\left( \mathrm{ND}(F_{\Delta_1}), \gamma \right) \tag{134}$$

$$:= \left\{ \mathbf{P}_\phi \lim_{\gamma^* \to \gamma} (1 - \gamma^*) \mathbf{f}(\gamma^*) \mid \mathbf{f} \in \mathrm{ND}(F_{\Delta_1}) \right\} \tag{135}$$

$$= \left\{ \lim_{\gamma^* \to \gamma} (1 - \gamma^*) \mathbf{P}_\phi \mathbf{f}(\gamma^*) \mid \mathbf{f} \in \mathrm{ND}(F_{\Delta_1}) \right\} \tag{136}$$

$$= \left\{ \lim_{\gamma^* \to \gamma} (1 - \gamma^*) \mathbf{f}(\gamma^*) \mid \mathbf{f} \in F'_{\text{sub}} \right\} \tag{137}$$

$$\subseteq \left\{ \lim_{\gamma^* \to \gamma} (1 - \gamma^*) \mathbf{f}(\gamma^*) \mid \mathbf{f} \in F_{\Delta_2} \right\} \tag{138}$$

$$=: \mathrm{NORM}(F_{\Delta_2}, \gamma). \tag{139}$$

Equation (134) follows by lemma E.44. Equation (136) follows because $\mathbf{P}_\phi$ is a continuous linear operator. Equation (138) follows by assumption.

$$\mathbb{E}_{s_1 \sim \Delta_1} \left[ \mathrm{POWER}_{\mathcal{D}_{bound}}(s_1, \gamma) \right] := \mathbb{E}_{\substack{s_1 \sim \Delta_1, \\ \mathbf{r} \sim \mathcal{D}_{bound}}} \left[ \max_{\pi \in \Pi} \lim_{\gamma^* \to \gamma} \frac{1 - \gamma^*}{\gamma^*} \left( \mathbf{f}^{\pi, s_1}(\gamma^*) - \mathbf{e}_{s_1} \right)^\top \mathbf{r} \right] \tag{140}$$

$$= \mathbb{E}_{\mathbf{r} \sim \mathcal{D}_{bound}} \left[ \max_{\pi \in \Pi} \lim_{\gamma^* \to \gamma} \frac{1 - \gamma^*}{\gamma^*} \mathbb{E}_{s_1 \sim \Delta_1} \left[ \mathbf{f}^{\pi, s_1}(\gamma^*) - \mathbf{e}_{s_1} \right]^\top \mathbf{r} \right] \tag{141}$$

$$= \mathbb{E}_{\mathbf{r} \sim \mathcal{D}_{bound}} \left[ \max_{\mathbf{d} \in \mathrm{NORM}(F_{\Delta_1}, \gamma)} \mathbf{d}^\top \mathbf{r} \right] \tag{142}$$

$$= \mathbb{E}_{\mathbf{r} \sim \mathcal{D}_{bound}} \left[ \max_{\mathbf{d} \in \mathrm{ND}\left( \mathrm{NORM}(F_{\Delta_1}, \gamma) \right)} \mathbf{d}^\top \mathbf{r} \right] \tag{143}$$

$$\leq_{\text{most: } \mathfrak{D}_{bound}} \mathbb{E}_{\mathbf{r} \sim \mathcal{D}_{bound}} \left[ \max_{\mathbf{d} \in \mathrm{NORM}(F_{\Delta_2}, \gamma)} \mathbf{d}^\top \mathbf{r} \right] \tag{144}$$

$$= \mathbb{E}_{\mathbf{r} \sim \mathcal{D}_{bound}} \left[ \max_{\pi \in \Pi} \lim_{\gamma^* \to \gamma} \frac{1 - \gamma^*}{\gamma^*} \mathbb{E}_{s_2 \sim \Delta_2} \left[ \mathbf{f}^{\pi, s_2}(\gamma^*) - \mathbf{e}_{s_2} \right]^\top \mathbf{r} \right] \tag{145}$$

$$= \mathbb{E}_{\substack{s_2 \sim \Delta_2, \\ \mathbf{r} \sim \mathcal{D}_{bound}}} \left[ \max_{\pi \in \Pi} \lim_{\gamma^* \to \gamma} \frac{1 - \gamma^*}{\gamma^*} \left( \mathbf{f}^{\pi, s_2}(\gamma^*) - \mathbf{e}_{s_2} \right)^\top \mathbf{r} \right] \tag{146}$$

$$=: \mathbb{E}_{s_2 \sim \Delta_2} \left[ \mathrm{POWER}_{\mathcal{D}_{bound}}(s_2, \gamma) \right]. \tag{147}$$

Equation (140) and eq. (147) follow by lemma E.45. Equation (141) and eq. (146) follow because each $R$ has a stationary deterministic optimal policy $\pi \in \Pi^*(R, \gamma) \subseteq \Pi$ which simultaneously achieves optimal value at all states. Equation (143) follows by corollary E.11.

Apply lemma E.24 with $A := \mathrm{NORM}(F_{\Delta_1}, \gamma)$, $B := \mathrm{NORM}(F_{\Delta_2}, \gamma)$, $g$ the identity function, and involution $\phi$ (satisfying $\phi \cdot \mathrm{ND}(A) \subseteq B$ by eq. (139)) in order to conclude that eq. (144) holds.

Suppose that $\mathrm{ND}\left(F_{\Delta_2}\right) \setminus \phi \cdot \mathrm{ND}\left(F_{\Delta_1}\right)$ is non-empty; let $F'_{\mathrm{sub}} := \phi \cdot \mathrm{ND}\left(F_{\Delta_1}\right)$. Lemma E.32 shows that for all $\gamma \in (0, 1)$, $\mathrm{ND}\left(F_{\Delta_2}(\gamma)\right) \setminus F'_{\mathrm{sub}}(\gamma)$ is non-empty. Lemma E.15 item 1 then implies that $\mathrm{ND}\left(B\right) \setminus \phi \cdot A = \frac{1-\gamma}{\gamma}\left(\mathrm{ND}\left(F_{\Delta_2}(\gamma)\right) - \mathbf{e}_s\right) \setminus \left(\frac{1-\gamma}{\gamma}F'_{\mathrm{sub}}(\gamma)\right)$ is non-empty. Then lemma E.24 implies that for all $\gamma \in (0, 1)$, eq. (144) is strict for all $\mathcal{D}_{X\text{-IID}} \in \mathfrak{D}_{\text{C/B/IID}}$ and $\mathbb{E}_{s_1 \sim \Delta_1}\left[\mathrm{POWER}_{\mathcal{D}_{\mathrm{bound}}}\left(s_1, \gamma\right)\right] \not\geq_{\text{most: } \mathfrak{D}_{\mathrm{bound}}} \mathbb{E}_{s_2 \sim \Delta_2}\left[\mathrm{POWER}_{\mathcal{D}_{\mathrm{bound}}}\left(s_2, \gamma\right)\right]$.

We show that this result's preconditions holding for $F^*_{\Delta_i}$ implies the $F_{\Delta_i}$ preconditions. Suppose $F^*_{\Delta_i} := \left\{\mathbb{E}_{s_i \sim \Delta_i}\left[\mathbf{f}^{\pi, s_i}\right] \mid \pi \in \Pi\right\}$ for $i = 1, 2$ are such that $F^*_{\mathrm{sub}} := \phi \cdot \mathrm{ND}\left(F^*_{\Delta_1}\right) \subseteq F^*_{\Delta_2}$. In the following, the $\Delta_i$ are represented as vectors in $\mathbb{R}^{|\mathcal{S}|}$, and $\gamma$ is a variable.

$$\phi \cdot \left\{\gamma \mathbf{f} \mid \mathbf{f} \in \mathrm{ND}\left(F_{\Delta_1}\right)\right\} = \phi \cdot \left(\mathrm{ND}\left(F^*_{\Delta_1} - \Delta_1\right)\right) \tag{148}$$

$$= \phi \cdot \left(\mathrm{ND}\left(F^*_{\Delta_1}\right) - \Delta_1\right) \tag{149}$$

$$= \left\{\mathbf{P}_\phi \mathbf{f} - \mathbf{P}_\phi \Delta_1 \mid \mathbf{f} \in \mathrm{ND}\left(F^*_{\Delta_1}\right)\right\} \tag{150}$$

$$\subseteq \left\{\mathbf{f} - \Delta_2 \mid \mathbf{f} \in F^*_{\Delta_2}\right\} \tag{151}$$

$$= \left\{\gamma \mathbf{f} \mid \mathbf{f} \in F_{\Delta_2}\right\}. \tag{152}$$

Equation (149) follows from lemma E.15 item 2. Since we assumed that $\phi \cdot \mathrm{ND}\left(F^*_{\Delta_1}\right) \subseteq F^*_{\Delta_2}$, $\phi \cdot \{\Delta_1\} = \phi \cdot \left(\mathrm{ND}\left(F^*_{\Delta_1}\right)(0)\right) \subseteq F^*_{\Delta_2}(0) = \{\Delta_2\}$. This implies that $\mathbf{P}_\phi \Delta_1 = \Delta_2$ and so eq. (151) follows.

Equation (152) shows that $\phi \cdot \left\{\gamma \mathbf{f} \mid \mathbf{f} \in \mathrm{ND}\left(F_{\Delta_1}\right)\right\} \subseteq \left\{\gamma \mathbf{f} \mid \mathbf{f} \in F_{\Delta_2}\right\}$. But we then have $\phi \cdot \left\{\gamma \mathbf{f} \mid \mathbf{f} \in \mathrm{ND}\left(F_{\Delta_1}\right)\right\} := \left\{\gamma \mathbf{P}_\phi \mathbf{f} \mid \mathbf{f} \in \mathrm{ND}\left(F_{\Delta_1}\right)\right\} = \left\{\gamma \mathbf{f} \mid \mathbf{f} \in \phi \cdot \mathrm{ND}\left(F_{\Delta_1}\right)\right\} \subseteq \left\{\gamma \mathbf{f} \mid \mathbf{f} \in F_{\Delta_2}\right\}$. Thus, $\phi \cdot \mathrm{ND}\left(F_{\Delta_1}\right) \subseteq F_{\Delta_2}$.

Suppose $\mathrm{ND}\left(F^*_{\Delta_2}\right) \setminus \phi \cdot \mathrm{ND}\left(F^*_{\Delta_1}\right)$ is non-empty, which implies that

$$\phi \cdot \left\{\gamma \mathbf{f} \mid \mathbf{f} \in \mathrm{ND}\left(F_{\Delta_1}\right)\right\} = \left\{\mathbf{P}_\phi \mathbf{f} - \mathbf{P}_\phi \Delta_1 \mid \mathbf{f} \in \mathrm{ND}\left(F^*_{\Delta_1}\right)\right\} \tag{153}$$

$$= \left\{\mathbf{f} - \mathbf{P}_\phi \Delta_1 \mid \mathbf{f} \in \phi \cdot \mathrm{ND}\left(F^*_{\Delta_1}\right)\right\} \tag{154}$$

$$\subsetneq \left\{\mathbf{f} - \Delta_2 \mid \mathbf{f} \in \mathrm{ND}\left(F^*_{\Delta_2}\right)\right\} \tag{155}$$

$$= \left\{\gamma \mathbf{f} \mid \mathbf{f} \in \mathrm{ND}\left(F_{\Delta_2}\right)\right\}. \tag{156}$$

Then $\mathrm{ND}\left(F_{\Delta_2}\right) \setminus \phi \cdot \mathrm{ND}\left(F_{\Delta_1}\right)$ must be non-empty. Therefore, if the preconditions of this result are met for $F^*_{\Delta_i}$, they are met for $F_{\Delta_i}$. $\qquad\square$

**Proposition 6.6** (States with "more options" have more POWER). *If $\mathcal{F}(s)$ contains a copy of $\mathcal{F}_{\mathrm{nd}}(s')$ via $\phi$, then $\forall \gamma \in [0, 1] : \mathrm{POWER}_{\mathcal{D}_{bound}}(s, \gamma) \geq_{\mathrm{most}} \mathrm{POWER}_{\mathcal{D}_{bound}}(s', \gamma)$. If $\mathcal{F}_{\mathrm{nd}}(s) \setminus \phi \cdot \mathcal{F}_{\mathrm{nd}}(s')$ is non-empty, then for all $\gamma \in (0, 1)$, the converse $\leq_{\mathrm{most}}$ statement does not hold.*

*Proof.* Let $F_{\mathrm{sub}} := \phi \cdot \mathcal{F}_{\mathrm{nd}}(s') \subseteq \mathcal{F}(s)$. Let $\Delta_1 := \mathbf{e}_{s'}, \Delta_2 := \mathbf{e}_s$, and define $F^*_{\Delta_i} := \left\{\mathbb{E}_{s_i \sim \Delta_i}\left[\mathbf{f}^{\pi, s_i}\right] \mid \pi \in \Pi\right\}$ for $i = 1, 2$. Then $\mathcal{F}_{\mathrm{nd}}(s') = \mathrm{ND}\left(F^*_{\Delta_1}\right)$ is similar to $F_{\mathrm{sub}} = F^*_{\mathrm{sub}} \subseteq F^*_{\Delta_2} = \mathcal{F}(s)$ via involution $\phi$. Apply lemma E.46 to conclude that $\forall \gamma \in [0, 1] : \mathrm{POWER}_{\mathcal{D}_{\mathrm{bound}}}\left(s', \gamma\right) \leq_{\text{most: } \mathfrak{D}_{\mathrm{bound}}} \mathrm{POWER}_{\mathcal{D}_{\mathrm{bound}}}\left(s, \gamma\right)$.

Furthermore, $\mathcal{F}_{\mathrm{nd}}(s) = \mathrm{ND}\left(F^*_{\Delta_2}\right)$, and $F_{\mathrm{sub}} = F^*_{\mathrm{sub}}$, and so if $\mathcal{F}_{\mathrm{nd}}(s) \setminus \phi \cdot \mathcal{F}_{\mathrm{nd}}(s') := \mathcal{F}_{\mathrm{nd}}(s) \setminus F_{\mathrm{sub}} = \mathrm{ND}\left(F^*_{\Delta_2}\right) \setminus F^*_{\mathrm{sub}}$ is non-empty, then lemma E.46 shows that for all $\gamma \in (0, 1)$, the inequality is strict for all $\mathcal{D}_{X\text{-IID}} \in \mathfrak{D}_{\text{C/B/IID}}$ and $\mathrm{POWER}_{\mathcal{D}_{\mathrm{bound}}}\left(s', \gamma\right) \not\geq_{\text{most: } \mathfrak{D}_{\mathrm{bound}}} \mathrm{POWER}_{\mathcal{D}_{\mathrm{bound}}}\left(s, \gamma\right)$. $\qquad\square$

**Lemma E.47** (Non-dominated visit distribution functions never agree with other visit distribution functions at that state)**.** *Let* $\mathbf{f} \in \mathcal{F}_{\mathrm{nd}}(s), \mathbf{f}' \in \mathcal{F}(s) \setminus \{\mathbf{f}\}$. $\forall \gamma \in (0,1) : \mathbf{f}(\gamma) \neq \mathbf{f}'(\gamma)$.

*Proof.* Let $\gamma \in (0,1)$. Since $\mathbf{f} \in \mathcal{F}_{\mathrm{nd}}(s)$, there exists a $\gamma^* \in (0,1)$ at which $\mathbf{f}$ is strictly optimal for some reward function. Then by proposition E.30, we can produce another reward function for which $\mathbf{f}$ is strictly optimal at discount rate $\gamma$; in particular, proposition E.30 guarantees that the policies which induce $\mathbf{f}'$ are not optimal at $\gamma$. So $\mathbf{f}(\gamma) \neq \mathbf{f}'(\gamma)$. $\qquad\square$

**Corollary E.48** (Cardinality of non-dominated visit distributions)**.** *Let* $F \subseteq \mathcal{F}(s)$. $\forall \gamma \in (0,1)$ : $\left| F \cap \mathcal{F}_{\mathrm{nd}}(s) \right| = \left| F(\gamma) \cap \mathcal{F}_{\mathrm{nd}}(s,\gamma) \right|$.

*Proof.* Let $\gamma \in (0,1)$. By applying lemma E.32 with $\Delta_d := \mathbf{e}_s$, $\mathbf{f} \in \mathcal{F}_{\mathrm{nd}}(s) = \mathrm{ND}\left(\mathcal{F}(s)\right)$ iff $\mathbf{f}(\gamma) \in \mathrm{ND}\left(\mathcal{F}(s,\gamma)\right)$. By lemma E.33, $\mathrm{ND}\left(\mathcal{F}(s,\gamma)\right) = \mathcal{F}_{\mathrm{nd}}(s,\gamma)$. So all $\mathbf{f} \in F \cap \mathcal{F}_{\mathrm{nd}}(s)$ induce $\mathbf{f}(\gamma) \in F(\gamma) \cap \mathcal{F}_{\mathrm{nd}}(s,\gamma)$, and $\left| F \cap \mathcal{F}_{\mathrm{nd}}(s) \right| \geq \left| F(\gamma) \cap \mathcal{F}_{\mathrm{nd}}(s,\gamma) \right|$.

Lemma E.47 implies that for all $\mathbf{f}, \mathbf{f}' \in \mathcal{F}_{\mathrm{nd}}(s)$, $\mathbf{f} = \mathbf{f}'$ iff $\mathbf{f}(\gamma) = \mathbf{f}'(\gamma)$. Therefore, $\left| F \cap \mathcal{F}_{\mathrm{nd}}(s) \right| \leq \left| F(\gamma) \cap \mathcal{F}_{\mathrm{nd}}(s,\gamma) \right|$. So $\left| F \cap \mathcal{F}_{\mathrm{nd}}(s) \right| = \left| F(\gamma) \cap \mathcal{F}_{\mathrm{nd}}(s,\gamma) \right|$. $\qquad\square$

**Lemma E.49** (Optimality probability and state bottlenecks)**.** *Suppose that $s$ can reach* $\mathrm{REACH}\left(s',a'\right) \cup \mathrm{REACH}\left(s',a\right)$, *but only by taking actions equivalent to $a'$ or $a$ at state $s'$.* $F_{nd,a'} := \mathcal{F}_{\mathrm{nd}}(s \mid \pi(s') = a'), F_a := \mathcal{F}(s \mid \pi(s') = a)$. *Suppose $F_a$ contains a copy of $F_{nd,a'}$ via $\phi$ which fixes all states not belonging to* $\mathrm{REACH}\left(s',a'\right) \cup \mathrm{REACH}\left(s',a\right)$. *Then* $\forall \gamma \in [0,1] : \mathbb{P}_{\mathcal{D}_{any}}\left(F_{nd,a'},\gamma\right) \leq_{\mathrm{most:}\ \mathfrak{D}_{any}} \mathbb{P}_{\mathcal{D}_{any}}\left(F_a,\gamma\right)$.

*If $\mathcal{F}_{\mathrm{nd}}(s) \cap \left(F_a \setminus \phi \cdot F_{nd,a'}\right)$ is non-empty, then for all $\gamma \in (0,1)$, the inequality is strict for all* $\mathcal{D}_{X\text{-IID}} \in \mathfrak{D}_{\mathrm{C/B/IID}}$, *and* $\mathbb{P}_{\mathcal{D}_{any}}\left(F_{nd,a'},\gamma\right) \not\geq_{\mathrm{most:}\ \mathfrak{D}_{any}} \mathbb{P}_{\mathcal{D}_{any}}\left(F_a,\gamma\right)$.

*Proof.* Let $F_{\mathrm{sub}} := \phi \cdot F_{nd,a'}$. Let $F^* := \bigcup_{\substack{a'' \in \mathcal{A}: \\ \left(a'' \not\equiv_{s'} a\right) \wedge \left(a'' \not\equiv_{s'} a'\right)}} \mathcal{F}(s \mid \pi(s') = a'') \cup F_{nd,a'} \cup F_{\mathrm{sub}}$.

$$\phi \cdot F^* := \phi \cdot \left( \bigcup_{\substack{a'' \in \mathcal{A}: \\ \left(a'' \not\equiv_{s'} a\right) \wedge \left(a'' \not\equiv_{s'} a'\right)}} \mathcal{F}(s \mid \pi(s') = a'') \cup F_{nd,a'} \cup F_{\mathrm{sub}} \right) \tag{157}$$

$$= \bigcup_{\substack{a'' \in \mathcal{A}: \\ \left(a'' \not\equiv_{s'} a\right) \wedge \left(a'' \not\equiv_{s'} a'\right)}} \phi \cdot \mathcal{F}(s \mid \pi(s') = a'') \cup \left(\phi \cdot F_{nd,a'}\right) \cup \left(\phi \cdot F_{\mathrm{sub}}\right) \tag{158}$$

$$= \bigcup_{\substack{a'' \in \mathcal{A}: \\ \left(a'' \not\equiv_{s'} a\right) \wedge \left(a'' \not\equiv_{s'} a'\right)}} \phi \cdot \mathcal{F}(s \mid \pi(s') = a'') \cup F_{\mathrm{sub}} \cup F_{nd,a'} \tag{159}$$

$$= \bigcup_{\substack{a'' \in \mathcal{A}: \\ \left(a'' \not\equiv_{s'} a\right) \wedge \left(a'' \not\equiv_{s'} a'\right)}} \mathcal{F}(s \mid \pi(s') = a'') \cup F_{\mathrm{sub}} \cup F_{nd,a'} \tag{160}$$

$$=: F^*. \tag{161}$$

Equation (159) follows because the involution $\phi$ ensures that $\phi \cdot F_{\mathrm{sub}} = F_{nd,a'}$. By assumption, $\phi$ fixes all $s' \notin \mathrm{REACH}\left(s',a'\right) \cup \mathrm{REACH}\left(s',a\right)$. Suppose $\mathbf{f} \in \mathcal{F}(s) \setminus \left(F_{nd,a'} \cup F_a\right)$. By the bottleneck assumption, $\mathbf{f}$ does not visit states in $\mathrm{REACH}\left(s',a'\right) \cup \mathrm{REACH}\left(s',a\right)$. Therefore, $\mathbf{P}_\phi \mathbf{f} = \mathbf{f}$, and so eq. (160) follows.

Let $F_Z := \left(\mathcal{F}(s) \setminus \left(\mathcal{F}(s \mid \pi(s) = a') \cup F_a\right)\right) \cup F_{nd,a'} \cup F_a$. By definition, $F_Z \subseteq \mathcal{F}(s)$. Furthermore, $\mathcal{F}_{\mathrm{nd}}(s) = \bigcup_{a'' \in \mathcal{A}} \mathcal{F}_{\mathrm{nd}}(s \mid \pi(s') = a'') \subseteq \left(\mathcal{F}(s) \setminus \left(\mathcal{F}(s \mid \pi(s) = a') \cup F_a\right)\right) \cup \mathcal{F}_{\mathrm{nd}}(s \mid \pi(s) = a') \cup F_a =: F_Z$, and so $\mathcal{F}_{\mathrm{nd}}(s) \subseteq F_Z$. Note that $F^* = F_Z \setminus \left(F_a \setminus F_{\mathrm{sub}}\right)$.

**Case:** $\gamma \in (0, 1)$.

$$\mathbb{P}_{\mathcal{D}_{\text{any}}} \left( F_{\text{nd},a'}, \gamma \right) = p_{\mathcal{D}_{\text{any}}} \left( F_{\text{nd},a'}(\gamma) \geq \mathcal{F}(s, \gamma) \right) \tag{162}$$

$$\leq_{\text{most: } \mathfrak{D}_{\text{any}}} p_{\mathcal{D}_{\text{any}}} \left( F_a(\gamma) \geq \mathcal{F}(s, \gamma) \right) \tag{163}$$

$$= \mathbb{P}_{\mathcal{D}_{\text{any}}} \left( F_{\text{nd},a'}, \gamma \right). \tag{164}$$

Equation (162) and eq. (164) follow from lemma E.37. Equation (163) follows by applying lemma E.28 with $A := F_{\text{nd},a'}(\gamma), B' := F_{\text{sub}}(\gamma), B := F_a(\gamma), C := \mathcal{F}(s, \gamma), Z := F_Z(\gamma)$ which satisfies $\text{ND}(C) = \mathcal{F}_{\text{nd}}(s, \gamma) \subseteq F_Z(\gamma) \subseteq \mathcal{F}(s, \gamma) = C$, and involution $\phi$ which satisfies $\phi \cdot F^*(\gamma) = \phi \cdot \left( Z \setminus (B \setminus B') \right) = Z \setminus (B \setminus B') = F^*(\gamma)$.

Suppose $\mathcal{F}_{\text{nd}}(s) \cap \left( F_a \setminus F_{\text{sub}} \right)$ is non-empty. $0 < \left| \mathcal{F}_{\text{nd}}(s) \cap \left( F_a \setminus F_{\text{sub}} \right) \right| = \left| \mathcal{F}_{\text{nd}}(s, \gamma) \cap \left( F_a(\gamma) \setminus F_{\text{sub}}(\gamma) \right) \right| =: \left| \text{ND}(C) \cap (B \setminus B') \right|$ (with the first equality holding by corollary E.48), and so $\text{ND}(C) \cap (B \setminus B')$ is non-empty. We also have $B := F_a(\gamma) \subseteq \mathcal{F}(s, \gamma) =: C$. Then reapplying lemma E.28, eq. (163) is strict for all $\mathcal{D}_{X\text{-IID}} \in \mathfrak{D}_{\text{C/B/IID}}$, and $\mathbb{P}_{\mathcal{D}_{\text{any}}} \left( F_{\text{nd},a'}, \gamma \right) \not\geq_{\text{most: } \mathfrak{D}_{\text{any}}} \mathbb{P}_{\mathcal{D}_{\text{any}}} \left( F_a, \gamma \right)$.

**Case:** $\gamma = 1$, $\gamma = 0$.

$$\mathbb{P}_{\mathcal{D}_{\text{any}}} \left( F_{\text{nd},a'}, 1 \right) = \lim_{\gamma^* \to 1} \mathbb{P}_{\mathcal{D}_{\text{any}}} \left( F_{\text{nd},a'}, \gamma^* \right) \tag{165}$$

$$= \lim_{\gamma^* \to 1} p_{\mathcal{D}_{\text{any}}} \left( F_{\text{nd},a'}(\gamma^*) \geq \mathcal{F}(s, \gamma^*) \right) \tag{166}$$

$$\leq_{\text{most: } \mathfrak{D}_{\text{any}}} \lim_{\gamma^* \to 1} p_{\mathcal{D}_{\text{any}}} \left( F_a(\gamma^*) \geq \mathcal{F}(s, \gamma^*) \right) \tag{167}$$

$$= \lim_{\gamma^* \to 1} \mathbb{P}_{\mathcal{D}_{\text{any}}} \left( F_a, \gamma^* \right) \tag{168}$$

$$= \mathbb{P}_{\mathcal{D}_{\text{any}}} \left( F_a, 1 \right). \tag{169}$$

Equation (165) and eq. (169) hold by proposition E.36. Equation (166) and eq. (168) follow by lemma E.37. Applying lemma E.29 with $\gamma := 1, I := (0, 1), F_A := F_{\text{nd},a'}, F_B := F_a, F_C := \mathcal{F}(s)$, $F_Z$ as defined above, and involution $\phi$ (for which $\phi \cdot \left( F_Z \setminus (F_B \setminus \phi \cdot F_A) \right) = F_Z \setminus (F_B \setminus \phi \cdot F_A)$), we conclude that eq. (167) follows.

The $\gamma = 0$ case proceeds similarly to $\gamma = 1$. $\qquad \square$

**Lemma E.50** (Action optimality probability is a special case of visit distribution optimality probability). $\mathbb{P}_{\mathcal{D}_{\text{any}}}(s, a, \gamma) = \mathbb{P}_{\mathcal{D}_{\text{any}}} \left( \mathcal{F}(s \mid \pi(s) = a), \gamma \right)$.

*Proof.* Let $F_a := \mathcal{F}(s \mid \pi(s) = a)$. For $\gamma \in (0, 1)$,

$$\mathbb{P}_{\mathcal{D}_{\text{any}}}(s, a, \gamma) := \mathbb{P}_{R \sim \mathcal{D}_{\text{any}}} \left( \exists \pi^* \in \Pi^*(R, \gamma) : \pi^*(s) = a \right) \tag{170}$$

$$= \mathbb{P}_{\mathbf{r} \sim \mathcal{D}_{\text{any}}} \left( \exists \mathbf{f}^{\pi^*, s} \in F_a : \mathbf{f}^{\pi^*, s}(\gamma)^\top \mathbf{r} = \max_{\mathbf{f} \in \mathcal{F}(s)} \mathbf{f}(\gamma)^\top \mathbf{r} \right) \tag{171}$$

$$= \mathbb{P}_{\mathcal{D}_{\text{any}}} \left( F_a, \gamma \right). \tag{172}$$

By lemma E.1, if $\exists \pi^* \in \Pi^*(R, \gamma) : \pi^*(s) = a$, then it induces some optimal $\mathbf{f}^{\pi^*, s} \in F_a$. Conversely, if $\mathbf{f}^{\pi^*, s} \in F_a$ is optimal at $\gamma \in (0, 1)$, then $\pi^*$ chooses optimal actions on the support of $\mathbf{f}^{\pi^*, s}(\gamma)$. Let $\pi'$ agree with $\pi^*$ on that support and let $\pi'$ take optimal actions at all other states. Then $\pi' \in \Pi^*(R, \gamma)$ and $\pi'(s) = a$. So eq. (171) follows.

Suppose $\gamma = 0$ or $\gamma = 1$. Consider any sequence $(\gamma_n)_{n=1}^\infty$ converging to $\gamma$, and let $\mathcal{D}_{\text{any}}$ induce probability measure $F$.

$$\mathbb{P}_{\mathcal{D}_{\text{any}}} \left( F_a, \gamma \right) := \lim_{\gamma^* \to \gamma} \mathbb{P}_{\mathcal{D}_{\text{any}}} \left( F_a, \gamma^* \right) \tag{173}$$

$$= \lim_{\gamma^* \to \gamma} \mathbb{P}_{R \sim \mathcal{D}_{\text{any}}} \left( \exists \pi^* \in \Pi^* \left( R, \gamma^* \right) : \pi^*(s) = a \right) \tag{174}$$

$$= \lim_{n \to \infty} \mathbb{P}_{R \sim \mathcal{D}_{\text{any}}} \left( \exists \pi^* \in \Pi^* \left( R, \gamma_n \right) : \pi^*(s) = a \right) \tag{175}$$

$$= \lim_{n \to \infty} \int_{\mathbb{R}^{\mathcal{S}}} \mathbb{1}_{\exists \pi^* \in \Pi^*(R,\gamma_n):\pi^*(s)=a} \, \mathrm{d}F(R) \tag{176}$$

$$= \int_{\mathbb{R}^{\mathcal{S}}} \lim_{n \to \infty} \mathbb{1}_{\exists \pi^* \in \Pi^*(R,\gamma_n):\pi^*(s)=a} \, \mathrm{d}F(R) \tag{177}$$

$$= \int_{\mathbb{R}^{\mathcal{S}}} \mathbb{1}_{\exists \pi^* \in \Pi^*(R,\gamma):\pi^*(s)=a} \, \mathrm{d}F(R) \tag{178}$$

$$=: \mathbb{P}_{\mathcal{D}_{\text{any}}} \left( s, a, \gamma \right). \tag{179}$$

Equation (174) follows by eq. (172). for $\gamma^* \in [0,1]$, let $f_{\gamma^*}(R) := \mathbb{1}_{\exists \pi^* \in \Pi^*(R,\gamma^*):\pi^*(s)=a}$. For each $R \in \mathbb{R}^{\mathcal{S}}$, lemma E.35 exists $\gamma_x \approx \gamma$ such that for all intermediate $\gamma'_x$ between $\gamma_x$ and $\gamma$, $\Pi^* \left( R, \gamma'_x \right) = \Pi^* \left( R, \gamma \right)$. Since $\gamma_n \to \gamma$, this means that $\left( f_{\gamma_n} \right)_{n=1}^{\infty}$ converges pointwise to $f_\gamma$. Furthermore, $\forall n \in \mathbb{N}, R \in \mathbb{R}^{\mathcal{S}} : \left| f_{\gamma_n}(R) \right| \le 1$ by definition. Therefore, eq. (177) follows by Lebesgue's dominated convergence theorem. $\square$

**Proposition 6.9** (Keeping options open tends to be POWER-seeking and tends to be optimal)**.**

*Suppose $F_a := \mathcal{F}(s \mid \pi(s) = a)$ contains a copy of $F_{a'} := \mathcal{F}(s \mid \pi(s) = a')$ via $\phi$.*

1. *If $s \notin \text{REACH} \left( s, a' \right)$, then $\forall \gamma \in [0,1] : \mathbb{E}_{s_a \sim T(s,a)} \left[ \text{POWER}_{\mathcal{D}_{bound}} \left( s_a, \gamma \right) \right] \ge_{\text{most: } \mathfrak{D}_{bound}} \mathbb{E}_{s_{a'} \sim T(s,a')} \left[ \text{POWER}_{\mathcal{D}_{bound}} \left( s_{a'}, \gamma \right) \right].$*

2. *If $s$ can only reach the states of $\text{REACH} \left( s, a' \right) \cup \text{REACH} \left( s, a \right)$ by taking actions equivalent to $a'$ or $a$ at state $s$, then $\forall \gamma \in [0,1] : \mathbb{P}_{\mathcal{D}_{any}} \left( s, a, \gamma \right) \ge_{\text{most: } \mathfrak{D}_{any}} \mathbb{P}_{\mathcal{D}_{any}} \left( s, a', \gamma \right).$*

*If $\mathcal{F}_{\text{nd}}(s) \cap \left( F_a \setminus \phi \cdot F_{a'} \right)$ is non-empty, then $\forall \gamma \in (0,1)$, the converse $\le_{\text{most}}$ statements do not hold.*

*Proof.* Note that by definition 3.3, $F_{a'}(0) = \{ \mathbf{e}_s \} = F_a(0)$. Since $\phi \cdot F_{a'} \subseteq F_a$, in particular we have $\phi \cdot F_{a'}(0) = \left\{ \mathbf{P}_\phi \mathbf{e}_s \right\} \subseteq \{ \mathbf{e}_s \} = F_a(0)$, and so $\phi(s) = s$.

**Item 1.** For state probability distribution $\Delta_s \in \Delta(\mathcal{S})$, let $F^*_{\Delta_s} := \left\{ \mathbb{E}_{s' \sim \Delta_s} \left[ \mathbf{f}^{\pi,s'} \right] \mid \pi \in \Pi \right\}$.

Unless otherwise stated, we treat $\gamma$ as a variable in this item; we apply element-wise vector addition, constant multiplication, and variable multiplication via the conventions outlined in definition E.14.

$$F_{a'} = \left\{ \mathbf{e}_s + \gamma \mathbb{E}_{s_{a'} \sim T(s,a')} \left[ \mathbf{f}^{\pi,s_{a'}} \right] \mid \pi \in \Pi : \pi(s) = a' \right\} \tag{180}$$

$$= \left\{ \mathbf{e}_s + \gamma \mathbb{E}_{s_{a'} \sim T(s,a')} \left[ \mathbf{f}^{\pi,s_{a'}} \right] \mid \pi \in \Pi \right\} \tag{181}$$

$$= \mathbf{e}_s + \gamma F^*_{T(s,a')}. \tag{182}$$

Equation (180) follows by definition 3.3, since each $\mathbf{f} \in \mathcal{F}(s)$ has an initial term of $\mathbf{e}_s$. Equation (181) follows because $s \notin \text{REACH} \left( s, a' \right)$, and so for all $s_{a'} \in \text{supp}(T(s,a'))$, $\mathbf{f}^{\pi,s_{a'}}$ is unaffected by the choice of action $\pi(s)$. Note that similar reasoning implies that $F_a \subseteq \mathbf{e}_s + \gamma F^*_{T(s,a)}$ (because eq. (181) is a containment relation in general).

Since $F_{a'} = \mathbf{e}_s + \gamma F^*_{T(s,a')}$, if $F_a$ contains a copy of $F_{a'}$ via $\phi$, then $F^*_{T(s,a)}$ contains a copy of $F^*_{T(s,a')}$ via $\phi$. Then $\phi \cdot \text{ND} \left( F^*_{T(s,a')} \right) \subseteq \phi \cdot F^*_{T(s,a')} \subseteq F^*_{T(s,a)}$, and so $F^*_{T(s,a)}$ contains a copy of $\text{ND} \left( F^*_{T(s,a')} \right)$. Then apply lemma E.46 with $\Delta_1 := T(s,a')$ and $\Delta_2 := T(s,a)$ to conclude that $\forall \gamma \in [0,1] : \mathbb{E}_{s_{a'} \sim T(s,a')} \left[ \text{POWER}_{\mathcal{D}_{bound}} \left( s_{a'}, \gamma \right) \right] \le_{\text{most: } \mathfrak{D}_{bound}} \mathbb{E}_{s_a \sim T(s,a)} \left[ \text{POWER}_{\mathcal{D}_{bound}} \left( s_a, \gamma \right) \right].$

Suppose $\mathcal{F}_{\mathrm{nd}}(s) \cap \left(F_a \setminus \phi \cdot F_{a'}\right)$ is non-empty. To apply the second condition of lemma E.46, we want to demonstrate that $\mathrm{ND}\left(F^*_{T(s,a)}\right) \setminus \phi \cdot \mathrm{ND}\left(F^*_{T(s,a')}\right)$ is also non-empty.

First consider $\mathbf{f} \in \mathcal{F}_{\mathrm{nd}}(s) \cap F_a$. Because $F_a \subseteq \mathbf{e}_s + \gamma F^*_{T(s,a)}$, we have that $\gamma^{-1}(\mathbf{f} - \mathbf{e}_s) \in F^*_{T(s,a)}$. Because $\mathbf{f} \in \mathcal{F}_{\mathrm{nd}}(s)$, by definition 3.6, $\exists \mathbf{r} \in \mathbb{R}^{|\mathcal{S}|}, \gamma_x \in (0,1)$ such that

$$\mathbf{f}(\gamma_x)^\top \mathbf{r} > \max_{\mathbf{f}' \in \mathcal{F}(s) \setminus \{\mathbf{f}\}} \mathbf{f}'(\gamma_x)^\top \mathbf{r}. \tag{183}$$

Then since $\gamma_x \in (0,1)$,

$$\gamma_x^{-1}(\mathbf{f}(\gamma_x) - \mathbf{e}_s)^\top \mathbf{r} > \max_{\mathbf{f}' \in \mathcal{F}(s) \setminus \{\mathbf{f}\}} \gamma_x^{-1}(\mathbf{f}'(\gamma_x) - \mathbf{e}_s)^\top \mathbf{r} \tag{184}$$

$$= \max_{\mathbf{f}' \in \gamma_x^{-1}\left((\mathcal{F}(s) \setminus \{\mathbf{f}\}) - \mathbf{e}_s\right)} \mathbf{f}'(\gamma_x)^\top \mathbf{r} \tag{185}$$

$$\geq \max_{\mathbf{f}' \in \gamma_x^{-1}\left((F_a \setminus \{\mathbf{f}\}) - \mathbf{e}_s\right)} \mathbf{f}'(\gamma_x)^\top \mathbf{r} \tag{186}$$

$$= \max_{\mathbf{f}' \in F^*_{T(s,a)} \setminus \left\{\gamma_x^{-1}(\mathbf{f} - \mathbf{e}_s)\right\}} \mathbf{f}'(\gamma_x)^\top \mathbf{r}. \tag{187}$$

Equation (186) holds because $F_a \subseteq \mathcal{F}(s)$. By assumption, action $a$ is optimal for $\mathbf{r}$ at state $s$ and at discount rate $\gamma_x$. Equation (181) shows that $F^*_{T(s,a)}$ potentially allows the agent a non-stationary policy choice at $s$, but non-stationary policies cannot increase optimal value [Puterman, 2014]. Therefore, eq. (187) holds.

We assumed that $\gamma^{-1}(\mathbf{f} - \mathbf{e}_s) \in \gamma^{-1}(\mathcal{F}_{\mathrm{nd}}(s) - \mathbf{e}_s)$. Furthermore, since we just showed that $\gamma^{-1}(\mathbf{f} - \mathbf{e}_s) \in F^*_{T(s,a)}$ is strictly optimal over the other elements of $F^*_{T(s,a)}$ for reward function $\mathbf{r}$ at discount rate $\gamma_x \in (0,1)$, we conclude that it is an element of $\mathrm{ND}\left(F^*_{T(s,a)}\right)$ by definition E.13. Then we conclude that $\gamma^{-1}(\mathcal{F}_{\mathrm{nd}}(s) - \mathbf{e}_s) \cap F^*_{T(s,a)} \subseteq \mathrm{ND}\left(F^*_{T(s,a)}\right)$.

We now show that $\mathrm{ND}\left(F^*_{T(s,a)}\right) \setminus \phi \cdot \mathrm{ND}\left(F^*_{T(s,a')}\right)$ is non-empty.

$$0 < \left| \mathcal{F}_{\mathrm{nd}}(s) \cap \left(F_a \setminus \phi \cdot F_{a'}\right) \right| \tag{188}$$

$$= \left| \gamma^{-1}\left(\mathcal{F}_{\mathrm{nd}}(s) \cap \left(F_a \setminus \phi \cdot F_{a'}\right) - \mathbf{e}_s\right) \right| \tag{189}$$

$$\leq \left| \gamma^{-1}\left(\mathcal{F}_{\mathrm{nd}}(s) - \mathbf{e}_s\right) \cap \left(F^*_{T(s,a)} \setminus \phi \cdot F^*_{T(s,a')}\right) \right| \tag{190}$$

$$= \left| \left(\gamma^{-1}\left(\mathcal{F}_{\mathrm{nd}}(s) - \mathbf{e}_s\right) \cap F^*_{T(s,a)}\right) \setminus \phi \cdot F^*_{T(s,a')} \right| \tag{191}$$

$$\leq \left| \mathrm{ND}\left(F^*_{T(s,a)}\right) \setminus \phi \cdot F^*_{T(s,a')} \right| \tag{192}$$

$$\leq \left| \mathrm{ND}\left(F^*_{T(s,a)}\right) \setminus \phi \cdot \mathrm{ND}\left(F^*_{T(s,a')}\right) \right|. \tag{193}$$

Equation (188) follows by the assumption that $\mathcal{F}_{\mathrm{nd}}(s) \cap \left(F_a \setminus \phi \cdot F_{a'}\right)$ is non-empty. Let $\mathbf{f}, \mathbf{f}' \in \mathcal{F}_{\mathrm{nd}}(s) \cap \left(F_a \setminus \phi \cdot F_{a'}\right)$ be distinct. Then we must have that for some $\gamma_x \in (0,1), \mathbf{f}(\gamma_x) \neq \mathbf{f}'(\gamma_x)$. This holds iff $\gamma_x^{-1}(\mathbf{f}(\gamma_x) - \mathbf{e}_s) \neq \gamma_x^{-1}(\mathbf{f}'(\gamma_x) - \mathbf{e}_s)$, and so eq. (189) holds.

Equation (190) holds because $F_a \subseteq \mathbf{e}_s + \gamma F^*_{T(s,a)}$ and $F'_a = \mathbf{e}_s + \gamma F^*_{T(s,a')}$ by eq. (182). Equation (192) holds because we showed above that $\gamma^{-1}(\mathcal{F}_{\mathrm{nd}}(s) - \mathbf{e}_s) \cap F^*_{T(s,a)} \subseteq \mathrm{ND}\left(F^*_{T(s,a)}\right)$. Equation (193) holds because $\mathrm{ND}\left(F^*_{T(s,a')}\right) \subseteq F^*_{T(s,a')}$ by definition E.13.

Therefore, $\mathrm{ND}\left(F^*_{T(s,a)}\right) \setminus \phi \cdot \mathrm{ND}\left(F^*_{T(s,a')}\right)$ is non-empty, and so apply the second condition of lemma E.46 to conclude that for all $\mathcal{D}_{X\text{-IID}} \in \mathfrak{D}_{\text{C/B/IID}}, \forall \gamma \in (0,1) :$

$\mathbb{E}_{s_{a'} \sim T(s,a')} \left[ \text{POWER}_{\mathcal{D}_{X\text{-IID}}} (s_{a'}, \gamma) \right] < \mathbb{E}_{s_a \sim T(s,a)} \left[ \text{POWER}_{\mathcal{D}_{X\text{-IID}}} (s_a, \gamma) \right]$, and that $\forall \gamma \in (0,1)$ : $\mathbb{E}_{s_{a'} \sim T(s,a')} \left[ \text{POWER}_{\mathcal{D}_{\text{bound}}} (s_{a'}, \gamma) \right] \not\geq_{\text{most: } \mathfrak{D}_{\text{bound}}} \mathbb{E}_{s_a \sim T(s,a)} \left[ \text{POWER}_{\mathcal{D}_{\text{bound}}} (s_a, \gamma) \right]$.

**Item 2.** Let $\phi'(s_x) := \phi(s_x)$ when $s_x \in \text{REACH}(s,a') \cup \text{REACH}(s,a)$, and equal $s_x$ otherwise. Since $\phi$ is an involution, so is $\phi'$.

$$\phi' \cdot F_{a'} := \left\{ \mathbf{P}_{\phi'} \left( \mathbf{e}_s + \gamma \mathop{\mathbb{E}}_{s_{a'} \sim T(s,a')} \left[ \mathbf{f}^{\pi, s_{a'}} \right] \right) \mid \pi \in \Pi, \pi(s) = a' \right\} \tag{194}$$

$$= \left\{ \mathbf{e}_s + \gamma \mathop{\mathbb{E}}_{s_{a'} \sim T(s,a')} \left[ \mathbf{P}_{\phi'} \mathbf{f}^{\pi, s_{a'}} \right] \mid \pi \in \Pi, \pi(s) = a' \right\} \tag{195}$$

$$= \left\{ \mathbf{P}_\phi \mathbf{e}_s + \gamma \mathop{\mathbb{E}}_{s_{a'} \sim T(s,a')} \left[ \mathbf{P}_\phi \mathbf{f}^{\pi, s_{a'}} \right] \mid \pi \in \Pi, \pi(s) = a' \right\} \tag{196}$$

$$=: \phi \cdot F_{a'} \tag{197}$$

$$\subseteq F_a. \tag{198}$$

Equation (195) follows because if $s \in \text{REACH}(s,a') \cup \text{REACH}(s,a)$, then we already showed that $\phi$ fixes $s$. Otherwise, $\phi'(s) = s$ by definition. Equation (196) follows by the definition of $\phi'$ on $\text{REACH}(s,a') \cup \text{REACH}(s,a)$ and because $\mathbf{e}_s = \mathbf{P}_\phi \mathbf{e}_s$. Next, we assumed that $\phi \cdot F_{a'} \subseteq F_a$, and so eq. (198) holds.

Therefore, $F_a$ contains a copy of $F_{a'}$ via $\phi'$ fixing all $s_x \notin \text{REACH}(s,a') \cup \text{REACH}(s,a)$. Therefore, $F_a$ contains a copy of $F_{\text{nd},a'} := \mathcal{F}_{\text{nd}}(s) \cap F_{a'}$ via the same $\phi'$. Then apply lemma E.49 with $s' := s$ to conclude that $\forall \gamma \in [0,1] : \mathbb{P}_{\mathcal{D}_{\text{any}}} (F_{a'}, \gamma) \leq_{\text{most: } \mathfrak{D}_{\text{any}}} \mathbb{P}_{\mathcal{D}_{\text{any}}} (F_a, \gamma)$. By lemma E.50, $\mathbb{P}_{\mathcal{D}_{\text{any}}} (s, a', \gamma) = \mathbb{P}_{\mathcal{D}_{\text{any}}} (F_{a'}, \gamma)$ and $\mathbb{P}_{\mathcal{D}_{\text{any}}} (s, a, \gamma) = \mathbb{P}_{\mathcal{D}_{\text{any}}} (F_a, \gamma)$. Therefore, $\forall \gamma \in [0,1] : \mathbb{P}_{\mathcal{D}_{\text{any}}} (s, a', \gamma) \leq_{\text{most: } \mathfrak{D}_{\text{any}}} \mathbb{P}_{\mathcal{D}_{\text{any}}} (s, a, \gamma)$.

If $\mathcal{F}_{\text{nd}}(s) \cap (F_a \setminus \phi \cdot F_{a'})$ is non-empty, then apply the second condition of lemma E.49 to conclude that for all $\gamma \in (0,1)$, the inequality is strict for all $\mathcal{D}_{X\text{-IID}} \in \mathfrak{D}_{\text{C/B/IID}}$, and $\mathbb{P}_{\mathcal{D}_{\text{any}}} (s, a', \gamma) \not\geq_{\text{most: } \mathfrak{D}_{\text{any}}} \mathbb{P}_{\mathcal{D}_{\text{any}}} (s, a, \gamma)$. $\qquad \square$

### E.4.2 When $\gamma = 1$, optimal policies tend to navigate towards "larger" sets of cycles

**Lemma E.51** (POWER identity when $\gamma = 1$).

$$\text{POWER}_{\mathcal{D}_{bound}} (s, 1) = \mathop{\mathbb{E}}_{\mathbf{r} \sim \mathcal{D}_{bound}} \left[ \max_{\mathbf{d} \in \text{RSD}(s)} \mathbf{d}^\top \mathbf{r} \right] = \mathop{\mathbb{E}}_{\mathbf{r} \sim \mathcal{D}_{bound}} \left[ \max_{\mathbf{d} \in \text{RSD}_{\text{nd}}(s)} \mathbf{d}^\top \mathbf{r} \right]. \tag{199}$$

*Proof.*

$$\text{POWER}_{\mathcal{D}_{bound}} (s, 1) = \mathop{\mathbb{E}}_{\mathbf{r} \sim \mathcal{D}_{bound}} \left[ \max_{\mathbf{f}^{\pi, s} \in \mathcal{F}(s)} \lim_{\gamma \to 1} \frac{1 - \gamma}{\gamma} \left( \mathbf{f}^{\pi, s}(\gamma) - \mathbf{e}_s \right)^\top \mathbf{r} \right] \tag{200}$$

$$= \mathop{\mathbb{E}}_{\mathbf{r} \sim \mathcal{D}_{bound}} \left[ \max_{\mathbf{d} \in \text{RSD}(s)} \mathbf{d}^\top \mathbf{r} \right] \tag{201}$$

$$= \mathop{\mathbb{E}}_{\mathbf{r} \sim \mathcal{D}_{bound}} \left[ \max_{\mathbf{d} \in \text{RSD}_{\text{nd}}(s)} \mathbf{d}^\top \mathbf{r} \right]. \tag{202}$$

Equation (200) follows by lemma E.45. Equation (201) follows by the definition of $\text{RSD}(s)$ (definition 6.10). Equation (202) follows because for all $\mathbf{r} \in \mathbb{R}^{|\mathcal{S}|}$, corollary E.11 shows that $\max_{\mathbf{d} \in \text{RSD}(s)} \mathbf{d}^\top \mathbf{r} = \max_{\mathbf{d} \in \text{ND}(\text{RSD}(s))} \mathbf{d}^\top \mathbf{r} =: \max_{\mathbf{d} \in \text{RSD}_{\text{nd}}(s)} \mathbf{d}^\top \mathbf{r}$. $\qquad \square$

**Proposition 6.12** (When $\gamma = 1$, RSDs control POWER). *If $\text{RSD}(s)$ contains a copy of $\text{RSD}_{\text{nd}}(s')$ via $\phi$, then $\text{POWER}_{\mathcal{D}_{bound}} (s, 1) \geq_{\text{most}} \text{POWER}_{\mathcal{D}_{bound}} (s', 1)$. If $\text{RSD}_{\text{nd}}(s) \setminus \phi \cdot \text{RSD}_{\text{nd}}(s')$ is non-empty, then the converse $\leq_{\text{most}}$ statement does not hold.*

*Proof.* Suppose $\text{RSD}_{\text{nd}}\left(s'\right)$ is similar to $D \subseteq \text{RSD}\left(s\right)$ via involution $\phi$.

$$\text{POWER}_{\mathcal{D}_{\text{bound}}}\left(s', 1\right) = \underset{\mathbf{r} \sim \mathcal{D}_{\text{bound}}}{\mathbb{E}}\left[\max_{\mathbf{d} \in \text{RSD}_{\text{nd}}(s')} \mathbf{d}^\top \mathbf{r}\right] \tag{203}$$

$$\leq_{\text{most: } \mathfrak{D}_{\text{bound}}} \underset{\mathbf{r} \sim \mathcal{D}_{\text{bound}}}{\mathbb{E}}\left[\max_{\mathbf{d} \in \text{RSD}_{\text{nd}}(s)} \mathbf{d}^\top \mathbf{r}\right] \tag{204}$$

$$= \text{POWER}_{\mathcal{D}_{\text{bound}}}\left(s, 1\right) \tag{205}$$

Equation (203) and eq. (205) follow from lemma E.51. By applying lemma E.24 with $A := \text{RSD}\left(s'\right), B' := D, B := \text{RSD}\left(s\right)$ and $g$ the identity function, eq. (204) follows.

Suppose $\text{RSD}_{\text{nd}}\left(s\right) \setminus D$ is non-empty. By the same result, eq. (204) is a strict inequality for all $\mathcal{D}_{X\text{-IID}} \in \mathfrak{D}_{\text{C/B/IID}}$, and we conclude that $\text{POWER}_{\mathcal{D}_{\text{bound}}}\left(s', 1\right) \not\geq_{\text{most: } \mathfrak{D}_{\text{bound}}} \text{POWER}_{\mathcal{D}_{\text{bound}}}\left(s, 1\right)$. $\qquad\square$

**Theorem 6.13** (Average-optimal policies tend to end up in "larger" sets of RSDs). *Let $D, D' \subseteq \text{RSD}\left(s\right)$. Suppose that $D$ contains a copy of $D'$ via $\phi$, and that the sets $D \cup D'$ and $\text{RSD}_{\text{nd}}\left(s\right) \setminus \left(D' \cup D\right)$ have pairwise orthogonal vector elements (i.e. pairwise disjoint vector support). Then $\mathbb{P}_{\mathcal{D}_{\text{any}}}\left(D, \text{average}\right) \geq_{\text{most}} \mathbb{P}_{\mathcal{D}_{\text{any}}}\left(D', \text{average}\right)$. If $\text{RSD}_{\text{nd}}\left(s\right) \cap \left(D \setminus \phi \cdot D'\right)$ is non-empty, the converse $\leq_{\text{most}}$ statement does not hold.*

*Proof.* Let $D_{\text{sub}} := \phi \cdot D'$, where $D_{\text{sub}} \subseteq D$ by assumption. Let $X := \left\{s_i \in \mathcal{S} \mid \max_{\mathbf{d} \in D' \cup D} \mathbf{d}^\top \mathbf{e}_{s_i} > 0\right\}$. Define

$$\phi'(s_i) := \begin{cases} \phi(s_i) & \text{if } s_i \in X \\ s_i & \text{else.} \end{cases} \tag{206}$$

Since $\phi$ is an involution, $\phi'$ is also an involution. Furthermore, by the definition of $X$, $\phi' \cdot D' = D_{\text{sub}}$ and $\phi' \cdot D_{\text{sub}} = D'$ (because we assumed that both equalities hold for $\phi$).

Let $D^* := D' \cup D_{\text{sub}} \cup \left(\text{RSD}_{\text{nd}}\left(s\right) \setminus \left(D' \cup D\right)\right)$.

$$\phi' \cdot D^* := \phi' \cdot \left(D' \cup D_{\text{sub}} \cup \left(\text{RSD}_{\text{nd}}\left(s\right) \setminus \left(D' \cup D\right)\right)\right) \tag{207}$$

$$= \left(\phi' \cdot D'\right) \cup \left(\phi' \cdot D_{\text{sub}}\right) \cup \phi' \cdot \left(\text{RSD}_{\text{nd}}\left(s\right) \setminus \left(D' \cup D\right)\right) \tag{208}$$

$$= D_{\text{sub}} \cup D' \cup \left(\text{RSD}_{\text{nd}}\left(s\right) \setminus \left(D' \cup D\right)\right) \tag{209}$$

$$=: D^*. \tag{210}$$

In eq. (209), we know that $\phi' \cdot D' = D_{\text{sub}}$ and $\phi' \cdot D_{\text{sub}} = D'$. We just need to show that $\phi' \cdot \left(\text{RSD}_{\text{nd}}\left(s\right) \setminus \left(D' \cup D\right)\right) = \text{RSD}_{\text{nd}}\left(s\right) \setminus \left(D' \cup D\right)$.

Suppose $\exists s_i \in X, \mathbf{d}' \in \text{RSD}_{\text{nd}}\left(s\right) \setminus \left(D' \cup D\right) : \mathbf{d'}^\top \mathbf{e}_{s_i} > 0$. By the definition of $X$, $\exists \mathbf{d} \in D' \cup D : \mathbf{d}^\top \mathbf{e}_{s_i} > 0$. Then

$$\mathbf{d}^\top \mathbf{d}' = \sum_{j=1}^{|\mathcal{S}|} \mathbf{d}^\top \left(\mathbf{d}' \odot \mathbf{e}_{s_j}\right) \tag{211}$$

$$\geq \mathbf{d}^\top \left(\mathbf{d}' \odot \mathbf{e}_{s_i}\right) \tag{212}$$

$$= \mathbf{d}^\top \left(\left(\mathbf{d'}^\top \mathbf{e}_{s_i}\right)\mathbf{e}_{s_i}\right) \tag{213}$$

$$= \left(\mathbf{d'}^\top \mathbf{e}_{s_i}\right) \cdot \left(\mathbf{d}^\top \mathbf{e}_{s_i}\right) \tag{214}$$

$$> 0. \tag{215}$$

Equation (211) follows from the definitions of the dot and Hadamard products. Equation (212) follows because $\mathbf{d}$ and $\mathbf{d}'$ have non-negative entries. Equation (215) follows because $\mathbf{d}^\top \mathbf{e}_{s_i}$ and $\mathbf{d'}^\top \mathbf{e}_{s_i}$ are both positive. But eq. (215) shows that $\mathbf{d}^\top \mathbf{d}' > 0$, contradicting our assumption that $\mathbf{d}$ and $\mathbf{d}'$ are orthogonal.

Therefore, such an $s_i$ cannot exist, and $X' := \left\{ s_i' \in \mathcal{S} \mid \max_{\mathbf{d}' \in \mathrm{RSD}_{\mathrm{nd}}(s) \setminus (D' \cup D)} \mathbf{d}'^\top \mathbf{e}_{s_i} > 0 \right\} \subseteq (\mathcal{S} \setminus X)$. By eq. (206), $\forall s_i' \in X' : \phi'(s_i') = s_i'$. Thus, $\phi' \cdot \left( \mathrm{RSD}_{\mathrm{nd}}(s) \setminus (D' \cup D) \right) = \mathrm{RSD}_{\mathrm{nd}}(s) \setminus (D' \cup D)$, and eq. (209) follows. We conclude that $\phi' \cdot D^* = D^*$.

Consider $Z := \left( \mathrm{RSD}_{\mathrm{nd}}(s) \setminus (D' \cup D) \right) \cup D \cup D'$. First, $Z \subseteq \mathrm{RSD}(s)$ by definition. Second, $\mathrm{RSD}_{\mathrm{nd}}(s) = \mathrm{RSD}_{\mathrm{nd}}(s) \setminus (D' \cup D) \cup (\mathrm{RSD}_{\mathrm{nd}}(s) \cap D') \cup (\mathrm{RSD}_{\mathrm{nd}}(s) \cap D) \subseteq Z$. Note that $D^* = Z \setminus (D \setminus D_{\mathrm{sub}})$.

$$\mathbb{P}_{\mathcal{D}_{\mathrm{any}}} \left( D', \mathrm{average} \right) = p_{\mathcal{D}_{\mathrm{any}}} \left( D' \geq \mathrm{RSD}(s) \right) \tag{216}$$

$$\leq_{\mathrm{most}: \mathfrak{D}_{\mathrm{any}}} p_{\mathcal{D}_{\mathrm{any}}} \left( D \geq \mathrm{RSD}(s) \right) \tag{217}$$

$$= \mathbb{P}_{\mathcal{D}_{\mathrm{any}}} \left( D, \mathrm{average} \right). \tag{218}$$

Since $\phi \cdot D' \subseteq D$ and $\mathrm{ND}(D') \subseteq D'$, $\phi \cdot \mathrm{ND}(D') \subseteq D$. Then eq. (217) holds by applying lemma E.28 with $A := D', B' := D_{\mathrm{sub}}, B := D, C := \mathrm{RSD}(s)$, and the previously defined $Z$ which we showed satisfies $\mathrm{ND}(C) \subseteq Z \subseteq C$. Furthermore, involution $\phi'$ satisfies $\phi' \cdot B^* = \phi' \cdot \left( Z \setminus (B \setminus B') \right) = Z \setminus (B \setminus B') = B^*$ by eq. (210).

When $\mathrm{RSD}_{\mathrm{nd}}(s) \cap \left( D \setminus D_{\mathrm{sub}} \right)$ is non-empty, since $B' \subseteq C$ by assumption, lemma E.28 also shows that eq. (217) is strict for all $\mathcal{D}_{X\text{-IID}} \in \mathfrak{D}_{\mathrm{C/B/IID}}$, and that $\mathbb{P}_{\mathcal{D}_{\mathrm{any}}} \left( D', \mathrm{average} \right) \not\leq_{\mathrm{most}: \mathfrak{D}_{\mathrm{any}}} \mathbb{P}_{\mathcal{D}_{\mathrm{any}}} \left( D, \mathrm{average} \right)$. $\square$

**Proposition E.52** (RSD properties). *Let* $\mathbf{d} \in \mathrm{RSD}(s)$. $\mathbf{d}$ *is element-wise non-negative and* $\|\mathbf{d}\|_1 = 1$.

*Proof.* $\mathbf{d}$ has non-negative elements because it equals the limit of $\lim_{\gamma \to 1}(1 - \gamma)\mathbf{f}(\gamma)$, whose elements are non-negative by proposition E.3 item 1.

$$\|\mathbf{d}\|_1 = \left\| \lim_{\gamma \to 1}(1 - \gamma)\mathbf{f}(\gamma) \right\|_1 \tag{219}$$

$$= \lim_{\gamma \to 1}(1 - \gamma) \left\| \mathbf{f}(\gamma) \right\|_1 \tag{220}$$

$$= 1. \tag{221}$$

Equation (219) follows because the definition of RSDs (definition 6.10) ensures that $\exists \mathbf{f} \in \mathcal{F}(s) : \lim_{\gamma \to 1}(1 - \gamma)\mathbf{f}(\gamma) = \mathbf{d}$. Equation (220) follows because $\|\cdot\|_1$ is a continuous function. Equation (221) follows because $\left\| \mathbf{f}(\gamma) \right\|_1 = \frac{1}{1 - \gamma}$ by proposition E.3 item 2. $\square$

**Lemma E.53** (When reachable with probability 1, 1-cycles induce non-dominated RSDs). *If* $\mathbf{e}_{s'} \in \mathrm{RSD}(s)$, *then* $\mathbf{e}_{s'} \in \mathrm{RSD}_{\mathrm{nd}}(s)$.

*Proof.* If $\mathbf{d} \in \mathrm{RSD}(s)$ is distinct from $\mathbf{e}_{s'}$, then $\|\mathbf{d}\|_1 = 1$ and $\mathbf{d}$ has non-negative entries by proposition E.52. Since $\mathbf{d}$ is distinct from $\mathbf{e}_{s'}$, then its entry for index $s'$ must be strictly less than 1: $\mathbf{d}^\top \mathbf{e}_{s'} < 1 = \mathbf{e}_{s'}^\top \mathbf{e}_{s'}$. Therefore, $\mathbf{e}_{s'} \in \mathrm{RSD}(s)$ is strictly optimal for the *reward function* $\mathbf{r} := \mathbf{e}_{s'}$, and so $\mathbf{e}_{s'} \in \mathrm{RSD}_{\mathrm{nd}}(s)$. $\square$

**Corollary 6.14** (Average-optimal policies tend not to end up in any given 1-cycle). *Suppose* $\mathbf{e}_{s_x}, \mathbf{e}_{s'} \in \mathrm{RSD}(s)$ *are distinct. Then* $\mathbb{P}_{\mathcal{D}_{\mathrm{any}}} \left( \mathrm{RSD}(s) \setminus \{\mathbf{e}_{s_x}\}, \mathrm{average} \right) \geq_{\mathrm{most}} \mathbb{P}_{\mathcal{D}_{\mathrm{any}}} \left( \{\mathbf{e}_{s_x}\}, \mathrm{average} \right)$. *If there is a third* $\mathbf{e}_{s''} \in \mathrm{RSD}(s)$, *the converse* $\leq_{\mathrm{most}}$ *statement does not hold.*

*Proof.* Suppose $\mathbf{e}_{s_x}, \mathbf{e}_{s'} \in \mathrm{RSD}(s)$ are distinct. Let $\phi := (s_x \ s'), D' := \{\mathbf{e}_{s_x}\}, D := \mathrm{RSD}(s) \setminus \{\mathbf{e}_{s_x}\}$. $\phi \cdot D' = \{\mathbf{e}_{s'}\} \subseteq \mathrm{RSD}(s) \setminus \{\mathbf{e}_{s_x}\} =: D$ since $s_x \neq s'$. $D' \cup D = \mathrm{RSD}(s)$ and $\mathrm{RSD}_{\mathrm{nd}}(s) \setminus (D' \cup D) = \mathrm{RSD}_{\mathrm{nd}}(s) \setminus \mathrm{RSD}(s) = \emptyset$ trivially have pairwise orthogonal vector elements. Then apply theorem 6.13 to conclude that $\mathbb{P}_{\mathcal{D}_{\mathrm{any}}} \left( \{\mathbf{e}_{s_x}\}, \mathrm{average} \right) \leq_{\mathrm{most}: \mathfrak{D}_{\mathrm{any}}} \mathbb{P}_{\mathcal{D}_{\mathrm{any}}} \left( \mathrm{RSD}(s) \setminus \{\mathbf{e}_{s_x}\}, \mathrm{average} \right)$.

Suppose there exists another $\mathbf{e}_{s''} \in \mathrm{RSD}(s)$. By lemma E.53, $\mathbf{e}_{s''} \in \mathrm{RSD}_{\mathrm{nd}}(s)$. Furthermore, since $s'' \notin \{s', s_x\}$, $\mathbf{e}_{s''} \in \left( \mathrm{RSD}(s) \setminus \{\mathbf{e}_{s_x}\} \right) \setminus \{\mathbf{e}_{s'}\} = D \setminus \phi \cdot D'$. Therefore,

$\mathbf{e}_{s''} \in \mathrm{RSD}_{\mathrm{nd}}(s) \cap \left(D \setminus \phi \cdot D'\right)$. Then apply the second condition of theorem 6.13 to conclude that $\mathbb{P}_{\mathcal{D}_{\mathrm{any}}}\left(\{\mathbf{e}_{s_x}\}, \mathrm{average}\right) \not\succeq_{\mathrm{most:}\ \mathfrak{D}_{\mathrm{bound}}} \mathbb{P}_{\mathcal{D}_{\mathrm{any}}}\left(\mathrm{RSD}(s) \setminus \{\mathbf{e}_{s_x}\}, \mathrm{average}\right)$. $\qquad\square$