# OpenReview forum: "Optimal Policies Tend To Seek Power"
_NeurIPS.cc/2021/Conference — NeurIPS 2021 Spotlight_

### Official Review · Reviewer_U4CB · 2021-06-28

**Rating:** 7
**Confidence:** 4

**Summary:**

The paper formalises the notion of an agent's power in a state $s$ as the average optimal value of that state, which the authors claim can be measured using a related power function. The paper posits that most optimal policies tend to seek states that achieve a higher value of power according to the power function under the set of bounded reward distributions. This is explained by optimal policies keeping options open, which naturally leads to a greater probability of optimal value for at least one option under the set of bounded reward distributions, and hence greater power. The paper then gives conditions in terms of symmetries in the MDP where such behaviour can be proved.

**Limitations And Societal Impact:**

On a philosophical level, I think that the authors' formalisation that power is being in a state that can lead to many rewarding goals being achieved in terms of average optimal value is limited and doesn't fully account for power under the working definition given in the paper, i.e. that 'power is the ability to achieve a range of goals [Sattarov, 2019]'. For example, the agent might not be able to learn the optimal value to act optimally, they might be uncertain about the MDP or even have incorrect knowledge of the state they are in - in the context of an AI agent avoiding being shut down, the agent is clearly in a lesser state of power if we can withold information about it ever being able to be shut down. Even the MDP the agent is in must be an essential part of understanding power. All of these other aspects are necessary for an agent to have power in terms of being able to actually achieve a goal. However these apsects are ignored in this work. As a real-life example, imagine that somebody has inherited a car. Unfortunately, they have no idea how to drive it. Having a car clearly increases the agent's power under the authors' definition of average optimal value as they are in a state where their actions have the potential to acheive a wider ranger of goals. Indeed, it would be easy to show that their average optimal value in this state is higher than a state without a car.  Crucially though, they don't have access to the optimal value function to be able to actually drive it. This means that they may actually be in state of less power (now being burdened with road tax , storage fees, nobody wanting to buy it etc.). One could posit a way of resolving this by making knowledge part of the agent's state, however this would mean extending state to include the agent's knowledge of the value function, which clearly creates issues in an MDP as any optimal value function would depend on knowledge of itself!

I believe this issue can be resolved by acknowledging that there are many necessary and sufficient aspects of being in a position of power, with one sufficient aspect being that states where more options are available can be more powerful. I think taking this perspective, discussing (or better formalising) other necessary aspects of power as well as fixing the pathologies identified above would significantly improve this paper.

**Main Review:**

Taking a dispositional view of power 'as the ability to achieve a wide variety of goals' is certainly valid, however I have concerns about the limitations of its formalisation here.

1. PATHOLOGIES IN THE FORMALISATION OF POWER

The paper posits that 'in an MDP, optimal value functions capture the agent’s ability to “achieve the goal” R. So average optimal value captures the agent’s ability to achieve a range of goals'. In the current definition of the MDP, I'm not sure that this is true as this does not account for preference over goals in terms of their optimal value. It would seem that using the average optimal reward would be vunerable to certain pathologies. As a counter example, consider the following MDP:

Consider a starting state $s_0$ and two actions that lead the agent deterministically to $s_l$ and $s_r$. In $s_l$, the agent recieves a reward of $R_l$ and then transitions to a terminal state $s_t$ with reward of 0. In $s_r$, the agent has a choice of 9 actions, each deterministically transitioning to a state $s_{r_i}$, $i\in\{1:9\}$. The reward for each state is $R_{r_i}$. The agent then transitions to the terminal state $s_t$.

If $R_l >> \max\_{i\in\{1:9\}} R_{r_i}$, then the average optimal value for being in $s_l$ is much higher than $s_r$. However being in state $s_l$ only allows the achievement of one goal, not a wide variety of goals. One could avoid such situations by introducing a restriction on rewards to normalise optimal value functions, however I believe this would severely limit the scope of the work as it seems natural that some goals should always have more value to agents than other: As an example of where this sort of MDP could be encountered, consider the option of somebody training for a very specific but very powerful job represented by entering in state $s_l$ vs obtaining a more general qualification that would allow that person to do a wide variety of jobs, but none of which would have as much power (or earn as much money) as the job in $s_l$. It seems natural that we would want to account for these situations if we were to formalise power. Perhaps a better solution would be to re-define or at least clarify what is meant by power to include a notion that powerful states can also lead agents to goals that a achieve a greater affect on the world in terms of optimal value at the expense of being able to achieve fewer goals overall.

2. DIFFERENCE BETWEEN $\textrm{Power}$ FUNCTION AND AVERAGE OPTIMAL VALUE

Ignoring this pathology, there is also a disconnect between the measure of power in terms of average optimal value and the $\textrm{Power}$ function that isn't explained. This is revealed by manipulating its definition:
$$
\begin{aligned}\textrm{Power}_\mathcal{D}(s,\gamma):=&\frac{1-\gamma}{\gamma} \mathbb{E}_\mathcal{D}\left[V_R^*(s)-R(s)\right],\\\\
=&\frac{1-\gamma}{\gamma} \mathbb{E}_\mathcal{D}\left[ \mathbb{E}\_{P\^\star(\tau\vert s)}\left[R(s) +\sum\_{i=1}^\infty\gamma^i R(s_i)\right]- R(s)\right],\\\
=&(1-\gamma)\mathbb{E}_\mathcal{D}\left[ \mathbb{E}\_{P\^\star(\tau\vert s)}\left[ \sum\_{i=1}^\infty\gamma\^{i-1} R(s_i)\right]\right],\\\\
=&(1-\gamma) \mathbb{E}_\mathcal{D}\left[\mathbb{E}\_{P\^\star(s_1\vert s)}\left[V^*_R(s_1)\right]\right],
\end{aligned}
$$
which is the average optimal value of the next state given state $s$. This leads to a different definition of power that ignores the reward of the current state (notice how it cancels between lines 2 and 3 above). It is easy to construct counterexample MDPs that would lead to opposing notions of which state is more powerful under the average optimal value definition and the $\textrm{Power}$ function in the same vein as the pathological example above. In addition, the $\textrm{Power}$ function is not motivated as a replacement for average optimal value beyond it not diverging in the limit $\gamma\rightarrow 1$. If this is a critical property, why not just use the optimal average value scaled by $(1-\gamma)$? As we see in the above derivation, this is essentially what you are using but averaged over the transition dynamics for the next state, and so by using optimal average value scaled by $(1-\gamma)$ you will obtain the same desired properties without changing its meaning.

3. UNCLEAR SIGNIFICANCE OF CONTRIBUTION

To me, it seems like the paper proves an obvious observation that can be elucidated by considering the MDP in 1 again. It seems somewhat trivial that for most bounded reward distributions over the reward function, being in state $s_r$ would lead to greater average optimal value as $9$ out of $10$ times the reward function would give the highest reward to one of $s_{r_i}$. Hence having more options would clearly lead to greater power under your definition. Although the paper seems to formalise and generalise this observation, which I think may be a valuable contribution as an abstract peice of theory, I'm not sure that it makes sense in the context of trying to formalise power. As discussed in 1, characterising power in this way is very limited, precisely because in many situations it doesn't make sense to consider all bounded distributions of rewards: we want to preserve some notion that highly specific states can lead to more power because they tend to give greater reward, even if we are uncertain about precisely what that reward is.

4. VALID CONTRIBUTION IN TERMS OF AI SAFETLY

I do, however, believe there is a valid contribution in the context of analysing the behaviour of a agent trying to be shut down in an MDP. If this was the sole focus of the work, rather than attempting to formalise a notion of power, I think it would improve its clarity and contribution significantly.

EDIT:

Following discussion with authors, most of my concerns have been addressed.

**Time Spent Reviewing:**

8

---

> ### Author Response · Authors · 2021-08-09
> **Reply to reviewer U4CB**
>
> Thank you for your review.
>
> While we will discuss and answer your particular concerns, we would like to point out that the main contribution of our paper is providing the first theory of the statistical tendencies of optimal policies in reinforcement learning. The main contribution is *not* the POWER formalism. We anticipated that reasonable people might disagree about how to formalize the intuitive notion of power, and so we made no claim to perfection.
>
> 1. In the caption of fig. 1, we wrote: “policies which go right [instead of] left… seek power – both intuitively, and in a reasonable formal sense.”
>
>     We formalized a notion of power from the philosophical literature, demonstrated POWER’s intuitive superiority over empowerment (a well-known and useful proxy for power, despite its demonstrated counter-intuitive edge-case performance), and showed that POWER has desirable qualitative properties (results 5.3 – 5.5). We believe we have met the claimed standard of ‘reasonableness.’
>
> 2. In the conclusion: “we proved sufficient conditions under which optimal policies tend to seek power, both formally (by taking Power-seeking actions) and intuitively (by taking actions which keep the agent’s options open).” Throughout the paper, we make our power-seeking arguments with both formal (POWER) and intuitive, “you know it when you see it” grounds.
>
> That said, we stand behind our POWER formalism. We think it is theoretically interesting, practically useful, and philosophically appropriate. While reasonable people may disagree about whether it can be further improved, we believe that we have convincingly argued that POWER is at least a reasonable proxy for power.
>
> We now reply to your specific concerns.
>
> ### Pathologies in POWER?
> We aren't sure we fully understand your concern. Here's our best guess – please correct us if your concern is different:
>
> You point out that $s_l$ can have strictly less median optimal value than $s_r$ while also having strictly greater average optimal value than $s_r$. We agree that normalization could be undesirable, since "it seems natural that some goals should always have more value to agents than others." We will implement your proposed solution of simply clarifying in the paper that average optimal value can, under some distributions, prioritize states where relatively few high-value goals can be achieved.
>
> ### POWER vs average optimal value
> We are familiar with your manipulation; indeed, it forms the proof of lemma D.35 on page 27 in the supplementary material. POWER is defined in order to delete a decision-irrelevant constant term (expected current-state reward). We meant to explain this choice with lines 165-166: “All $\mathbf{f}\in \mathcal{F}(s)$ count the agent’s presence at the initial state $s$.” However, as both you and reviewer TsSV implicitly point out, this explanation is insufficient, and we will clarify in the camera-ready.
>
> We want POWER to capture the agent’s control over the future, but in the state-based reward setting, the agent has no control over the fact that it’s already at a given state. So why should POWER include a constant based on that state’s reward? Furthermore, by removing current-state reward and dividing by $\gamma$, POWER$(s,0)$ becomes the expected greedy next-state reward at state $s$, which we find appealing.
>
> You’re right in that none of this really matters for this paper’s results – it’s more a matter of taste. Our results hold whether or not we subtract this constant, so we do not see it as a material objection to the significance of our contributions.
>
> ### On knowing the optimal value
> > the agent might not be able to learn the optimal value to act optimally
>
> This is a great point (although we do point out that we flagged the assumption of optimality in the introduction (l. 32-33), the discussion (l. 362-364), and the conclusion (l. 393-394)).
>
> We do believe that despite not having optimal policies in practice, it is still important to understand the statistical tendencies of optimal policies. Echoing reviewer xxyk, the explicitly stated objective of the field of reinforcement learning is to find optimal policies; it seems clearly significant to understand what the field as a whole is aiming for.
>
> Regarding your observation – in a previous version of the paper, we wrote:
>
> > In certain situations, POWER returns intuitively surprising verdicts. [To make a loose analogy,] there exists a policy under which the reader chooses a winning lottery ticket, but it seems wrong to say that the reader has the power to win the lottery with high probability. For various reasons, humans and other bounded agents are generally incapable of computing optimal policies for arbitrary objectives.
>
> We then defined a relaxed version of POWER which handles these issues nicely. We will include it as an appendix for the camera-ready.
>
> There is only room for so much content in the paper. While this philosophical clarification is intriguing, it isn’t central to our paper’s main contribution. We originally decided to cut this appendix in order to minimize the length of an already-long submission. In the end, we cut the length down from 54 pages to a “slender” 42! We are reviewers ourselves, and we appreciate the burden imposed by longer works.
>
> ### Obviousness of contributions?
> > To me, it seems like the paper proves an obvious observation… It seems somewhat trivial that for most bounded reward distributions over the reward function, being in state $s_r$ would lead to greater average optimal value as 9 out of 10 times the reward function would give the highest reward to one of the [9 states to the right].
>
> We are glad that this idea makes sense after reading the paper. However, calling this observation ‘obvious’ strains credulity:
>
> 1. The relevant quantities were unimagined before this work – let alone formally defined.
> 2. The observation escaped us for months after we had defined POWER.
> 3. In mathematics, ‘obvious’ formal observations can be untrustworthy. The value comes from the proofs (which are non-trivial).
> 4. The technical conditions themselves are non-obvious.
>     1. For example, the involution requirement is a requirement; permutation-enforced similarities are not enough in general.
>     2. For example, it seems measure-theoretically impossible to construct a uniform probability measure over bounded probability measures on $\mathbb{R}^{|\mathcal{S}|}$, and so it’s *a priori* unclear whether “most bounded distributions” is even a coherent claim.
>
> The observation is obvious – in hindsight.
>
> ### Significance
> Reviewer TsSV: “I find the arguments here compelling and novel and expect them to have relevance to the RL community as well as more widely (especially the final few comments relating to autonomous agents acting in society)”, remarking that we provide “neat” and “novel” theoretical formalizations and abstractions.
>
> Reviewer xxyk states “The result is perhaps not surprising to everyone but nonetheless important because it contributes to this ongoing debate. Not only the results but also the formalizations will be useful for future research and discussion. Taken together, the paper is likely to be among the most high-impact ones at Neurips.”
>
> Above all, we believe that the paper speaks for itself. Potential quirks or imperfections in POWER are irrelevant to our core contribution, which is: the first theory of the statistical tendencies of optimal policies.

---

> > ### Comment · Reviewer_U4CB · 2021-08-19
> > **Updated score**
> >
> > Thanks for clarifying my concerns, I'm happy to revise my score in light of this provided that the paper is updated to address these issues.

---

> > > ### Author Response · Authors · 2021-08-19
> > > **Re: updated score**
> > >
> > > Great — we will be sure to update the paper accordingly.

---

> > > > ### Comment · Reviewer_U4CB · 2021-08-20
> > > > **semantics**
> > > >
> > > > what a strange comment

---

> ### Comment · Reviewer_xxyk · 2021-08-11
> **Re pathologies in POWER**
>
> Concern 1. constructs an MDP where the agent chooses between the option to enter many states with low reward and one state with high reward. The real world may contains such choices, e.g. choosing between learning a generic skill useful for a range of low-income (or low power) jobs versus a specific skill for a high-income job. The example assumes a fixed reward function: high reward for the high-income job and low reward for low-income jobs. If I understand the concern (I might not), it is that POWER would be higher for preserving option value, instead of reaping the high reward of the high-income job.
>
> I agree that there exist specific states in the real world like the high-income job that tend to lead to high reward across different reward functions: for example, different items one may like to buy. However, the example MDP is incomplete as it lacks the terminal states where the different items are bought. For example, if the high-income job allows one choose from 1000 items whereas the lower-income jobs only allows one to buy one of 100 items, the high-income job has more POWER. Thus it works as desired.
>
> The paper could clarify this by highlighting that intuitive notions of power (such as money) are not typically rewarded, and rather are a means to obtain future rewards (such as buying items). Thus the example actually seems to argue FOR the main point of the paper by saying that there exist specific states that confer high reward across different reward functions.

---

> ### Comment · Reviewer_xxyk · 2021-08-11
> **Re limitation about suboptimal policies**
>
> The final criticism is that the notion of power is unsatisfactory for suboptimal policies. This does seems true. To me it is acceptable in a paper about *optimal* policies. It can and should surely be extended in future work.

---

### Official Review · Reviewer_xxyk · 2021-07-12

**Rating:** 8
**Confidence:** 4

**Summary:**

The paper formalizes a notion of power-seeking in MDPs and shows that many reward functions lead to optimal policies that achieve powerful states.

**Limitations And Societal Impact:**

Limitations and societal impact are adequately addressed in the discussion.


**Main Review:**

This is a significant step towards settling a long-standing debate that most AI researchers will have considered or even participated in but only in an informal context. It is also an important debate as it affects the field’s priorities. The result is perhaps not surprising to everyone but nonetheless important because it contributes to this ongoing debate. Not only the results but also the formalizations will be useful for future research and discussion. Taken together, the paper is likely to be among the most high-impact ones at Neurips.

Although the community has expected that results like the ones in this paper can be proven, my impression is that, it has been difficult to do so with any generality and therefore nothing is published yet. It is good to see results with some generality now.

Perhaps as a side effect of trying to achieve more general results, the derivations with substantial amounts of notation are a bit tedious to follow at times. Although the prose is well-written, mathematical presentation could be further improved.

-------


Minor comments:

- Can we use these insights to engineer a system that avoids power seeking?

- First paragraph of discussion: not clear what is the point of this paragraph

- Can you discuss more clearly why and to what extent your results apply to many MDPs?

- It’s obvious but maybe mention why an analysis of optimal policies is useful. I.e. because our community is constantly improving agents to produce closer to optimal policies

-------

Comments on presentation/writing:

- The paper could benefit from an illustrative example. The MDP graphs help but they are not self-explanatory. One way to address this would be to cast the MDP in terms of concrete objects and explain the intuitions in the caption. Additionally, section 6 could also benefit from an intuitive characterization both in section 6 itself and in other parts of the paper that refer to it. Otherwise I fear you’ve lost many readers by this point. This should also explain why many MDPs should have the proposed symmetries.

- The paper could state more clearly throughout if the goal is to show that agents seek influence (e.g. by earning money), or seek survival, or both. The last sentence of the discussion highlights this issue - can you discuss the evidence that your results can be extrapolated to accumulating resources? The concept of ‘keeping options open’ sounds more like postponing commitment than e.g. accumulating resources or influence. A more appropriate term might be ‘maximizing open options’?

- The notation is overall a cluttered which can make the derivations tedious to follow.

- “into computational resources” - while this is related to power seeking it seems more of a central example for side effects

- “Section 5 defines “power” as the ability to achieve a wide range of goals: after all, “money is power”, and money is instrumentally useful for many goals.”: the first part of the sentence (“power” as the ability to achieve a wide range of goals) does not follow from the second.

- I’d recommend using less scare quotes

- “Optimal policies for e.g. Pac-Man do not seek power in the real world”: I don’t follow, why would someone deploy a pacman policy in the real world?



**Time Spent Reviewing:**

3

---

> ### Author Response · Authors · 2021-08-09
> **Reply to reviewer xxyk**
>
> Thank you for your well-considered remarks and suggestions. We are honored by your appraisal of this work’s impact, and of the utility of its formalizations and results. We look forward to further refining the paper through discussion with the reviewers.
>
> > Can we use these insights to engineer a system that avoids power seeking?
>
> We do hope to gain that level of understanding by more precisely detailing the causes of power-seeking incentives (which need not always be bad, as noted in lines 369-370). That said, we do not currently see a way to avoid power-seeking with high confidence via just these insights.
>
> > Can you discuss more clearly why and to what extent your results apply to many MDPs?
>
> As you know, in section 6.3 (page 8) we walk through several reasonable environments which our results apply to. These environments were easy to come by because optimal policies tend to end up in larger sets of RSDs (theorem 6.13), with ‘larger’ only requiring similarity-via-permutation.
>
> Why should these environmental symmetries be present in many practically interesting MDPs? One answer is that these MDPs tend to exhibit spatial regularity and tend to factorize along several dimensions, and so different sets of RSDs will be similar, requiring only modification of several factor values. For example, if an embodied agent can navigate a set of three similar rooms (spatial regularity), then these environment states factorize via {room number} $\times$ {room state}. And so the RSDs can be divided into three similar subsets, depending on the agent’s room number.
>
> And in any environment in which the agent must choose between two disjoint sets of RSDs (a “fork in the road”), our results imply that when $\gamma=1$, optimal policies will tend to navigate towards the larger set. Such “forks” seem reasonably common in environments with irreversible actions.
>
> > state more clearly throughout if the goal is to show that agents seek influence (e.g. by earning money), or seek survival, or both.
>
> We believe that POWER captures an important part of what it means to have influence. That said, we focus on survival-seeking because it can be determined graphically, while money-seeking requires semantic interpretation of world states. We will clarify the paper’s goal in the camera-ready.
>
> > can you discuss the evidence that your results can be extrapolated to accumulating resources?
>
> Roughly, at $\gamma\approx 1$, POWER sometimes captures “ability to gain resources”, instead of the agent’s actual level of resources (as you note). However, since POWER nicely captures the agent’s influence over the future (see Appendix A for how POWER succeeds where empowerment does not), it seems natural to expect that some version of our results will hold for an intuitive version of resources.
>
> (Note that “extrapolating from our results” [359-361] was meant to indicate that we were sharing an intuition based on our results, not that our present theorems will literally extrapolate to abstract resources like money.)

---

> > ### Comment · Reviewer_xxyk · 2021-08-11
> > **Survival vs power seeking**
> >
> > > We believe that POWER captures an important part of what it means to have influence. That said, we focus on survival-seeking because it can be determined graphically, while money-seeking requires semantic interpretation of world states. We will clarify the paper’s goal in the camera-ready.
> >
> > To clarify my concern, the paper seems to equivocate between
> > 1) Keeping *some* options open. This is what I understand survival seeking to mean.
> > 2) Maximizing open options. This could be reasonably called power-seeking.

---

> > > ### Author Response · Authors · 2021-08-11
> > > **Re: survival vs power seeking**
> > >
> > > Thank you for the clarification. You have correctly understood what we meant by ‘survival-seeking.’
> > >
> > > We define POWER-seeking (definition 5.6 on page 5) to be relative: one action seeks *more* POWER than another. We think of intuitive power-seeking in a similar way. Therefore, we consider “keep some options open” to be power-seeking compared to “die and keep no options open.” Similarly, maximizing open options is power-seeking compared to not maximizing open options.
> > >
> > > In this sense, we did not consider ourselves to be equivocating between two kinds of arguments. However, there may certainly be passages where we communicated unclearly (as you noted for our conjecture about resource-seeking).
> > >
> > > Does this address the issue, or do you think we still seem to be equivocating between different notions?

---

> > > > ### Comment · Reviewer_xxyk · 2021-08-19
> > > > **Clarified**
> > > >
> > > > Thank you, saying that power seeking is relative does does clarify the confusion.

---

> > ### Comment · Reviewer_xxyk · 2021-08-11
> > **Presentation**
> >
> > Since this wasn't addressed in the response, I'll also reiterate that I think the paper's impact could be significantly improved by presenting its mathematical arguments more intuitively if possible (see review). Currently, readers will have to go through understanding the cumbersome introduced notation. Ideally, the figures with captions would be self-explanatory.

---

> > > ### Author Response · Authors · 2021-08-11
> > > **Re: Presentation**
> > >
> > > We agree that this is important. We are passionate about ensuring this paper is as accessible and widely understood as possible.
> > >
> > > The key challenge is that to the best of our knowledge, English does not admit short informal phrases which losslessly explain the preconditions and conclusions of our results. That said, by accepting a bit of ambiguity in the informal explanations, we may be able to further improve the presentation (in the submission, we used scare quotes to mark lossy informal compression, although we will cut back on these quotes re: your feedback).
> > >
> > > Concretely, here are our currently planned improvements for each figure:
> > > 1. already contains an intuitive explanation.
> > > 2. prepend to caption: "Some trajectories cannot be strictly optimal for any reward function, and so our results are able to ignore them."
> > > 3. prepend to caption: "Intuitively, state $r_{\searrow}$ affords the agent more power than state $\varnothing$. Our POWER formalism captures that intuition by computing a function of the agent's average optimal value across a range of reward functions."
> > > 4. prepend to caption: "Intuitively, the agent can do more starting from $r_{\searrow}$ than from $\ell_{\swarrow}$. We make this precise via an embedding."
> > > 5. prepend to caption: "A permutation of a reward function swaps which states get which rewards. We will show that in certain situations, for any reward function $R$, power-seeking is optimal for most of the permutations of $R$. The orbit of a reward function is the set of its permutations. We can also consider the orbit of a distribution over reward functions. This figure shows the [probability density...]"
> > > 6. prepend to caption: "At all discount rates, it's optimal to go $\texttt{right}$ for most reward functions $R$. This is true because whenever $R$ makes $\texttt{left}$ strictly optimal over $\texttt{right}$, its permutation $R\circ \phi$ makes $\texttt{right}$ strictly optimal over $\texttt{left}$ by switching which states get which rewards."
> > > 7. append to caption: "When $\gamma=1$, it will be optimal for most reward functions to avoid $\varnothing$, because $\varnothing$ is only a single inescapable terminal state. Other parts of the state space offer more 1-cycles (*e.g.* $\ell_{\swarrow},r_{\nwarrow},r_{\searrow}$)."
> > > 8. already contains reasonably intuitive text, although we will modify the last sentence to be a statement instead of a question: "Fixing the dynamics and $\gamma=1$, as the reward function varies, $\texttt{right}$ tends to be optimal over $\texttt{left}$. Roughly, this is because the agent can do more by staying alive."
> > >
> > > In addition to improving the captions, we will incorporate your other suggestions regarding clarity and accessibility in the rest of the paper.

---

> > > > ### Comment · Reviewer_xxyk · 2021-08-19
> > > > **Thanks**
> > > >
> > > > Thank you, these will be helpful improvements.

---

### Official Review · Reviewer_TsSV · 2021-07-14

**Rating:** 8
**Confidence:** 3

**Summary:**

The paper presents a new way to think about the power of an MDP state with a view to understanding when and why autonomous agents seek power in a given environment. Similar to other treatments "power" captures something about the freedom to act towards diverse objectives, and the authors relate this appropriately to that literature, e.g. empowerment.

The authors begin with motivation to develop a notion of power that is free from anthropomorphic a priori assumption. However, this is a mostly theoretical/mathematical contribution with a series of definitions, e.g. the power-function of a state, and theoretical results, e.g. smoothness properties of their power-function and gives a general perspective of why autonomous agents might act to seek power (i.e. to navigate towards states of high power). The later results exploit ways to think about symmetries in MDPs based on permutations of states, and allows for quite general results pertaining to agents navigating to states of high power under weak assumptions.

The paper ends with a discussion of how the concrete mathematical arguments can be applied to more general domains including those of a general agent acting in an open environment. One conclusion is a warning about how weak assumptions about how a an autonomous learner is motivates can lead to impulsions to "seek power" and explain that this may have relevance for the safe development of autonomous agents operating in society.

**Limitations And Societal Impact:**

There is a clear route to societal impact outlined by the authors, and while it isn't conclusive, the authors are asking for a debate and I think that is valid.

My concerns are relatively minor and tend to relate to the justifications for a implications of the theoretical work rather than the work itself. These are:

* A comment that different results make different distributional assumptions over rewards appears on line 129 in section 4, but it would be good to have this stated very early on e.g. abstract/introduction.
* That at times (towards the end of the paper), I am not sure whether the descriptive language appropriately captures the mathematical concepts being developed. For instance, the use of the words "tends", "usually" and "most". While there are distributions applied to reward functions, the orbits are sets of distributions and there is no measure defined over this set. The use of adverbs of degree to describe the high frequency outcomes in a subset of distributions should be made a little clearer I think.
* In fact, it could be clearer what the motivation is for defining distributions of plausible reward functions in the way that is done. It may be the case that certain topological properties of sub-parts of a naturally arising decision problem correlate with reward function over that sub-part. In fact, it is extremely uncommon in games/human activities for task success to be associated with hard to reach states. That isn't to say these states aren't of high power, just that the assumptions about IID rewards and orbits based on permutations of rewards, may not always be applicable.
* On lines 320-323, you state that your result applies to all degenerate distributions. However, we should remember that the orbit of a degnenerate distributions is any permutation of a degenerate distribution. Your earlier use of "tends" and "most" etc , seems to suggest that you are treating this orbit in some kind of measurable way, e.g. as a distribution of its own, and so the message here isn't as clear as it could be.

These issues could be addressed with a couple of comments relating to the validity/scope of assumptions and how to interpret particular choices of words.

**Main Review:**

The authors develop their mathematical ideas for an MDP with infinite horizon (i.e. with absorbing/terminal states represented as those with only reflexive outgoing edges of the graph) and then define states in terms of their set of available "visit distribution functions" and various restrictions of this notion. This is a very neat way to abstract away from many of the complexities arising from such things as multiple actions having the same outcome, and lays the foundations for simliar neat abstractions on top of this, such as the non-dominated set of visit functions.

Section 5 gives a result that states something along the lines of "this state has a higher expected optimal value" (as an expectation over reward functions drawn from a known distribution) and consequently defines the power of a state as a related function with various goodness properties. I had some concerns reading the opening sections that there were going to be (strong) implicit assumptions about the distribution of tasks/reward functions, and these concerned were alleviated here, as the assumptions are explicitly stated and in many cases only weak constraining (e.g. that the functions/distributions over functions are bounded).

Section 6 defines the orbit of a distribution as a set of distributions resulting from permutations of the original distribution's dimensions, and another novel (to me) idea that similarity up to permutation between a one states visit distribution (functions) and a subset of another's visit distribution (functions). This gives them a way to explore how fairly general symmetries in the environment can be used to score states and actions within states with very weak assumptions over reward functions. Finally, the authors give results relating to Blackwell optimal policies (essentially average reward optimisers) again reasoning in a general way about the states that "tend" to be navigated towards by "most" policies. The descriptive language here lets them down a little, as I think they could be clearer in explaining what they mean by such things as "tend", "usually" and "most" but the mathematical definitions are clear.


I find the arguments here compelling and novel and expect them to have relevance to the RL community as well as more widely (especially the final few comments relating to autonomous agents acting in society). The arguments are clearly made with formal mathematical definitions. I spotted a few minor things that were either unclear or ambiguious, but on the whole it was exceptionally clear mathematically and well described in natural language.

I have a few specific points, e.g. typos etc, these are:

* [line 39 onwards] The definition of robustly instrumental is making some assumption about distribution of tasks. That could be clearer.
* [line 64] The comment about computer vision systems reliably learning is less obviously tied to the line of reasoning than it could be.
* [line 111] The set of "non-dominated" visit distribution functions seems to comprise of visit functions that strictly dominate for at least one reward function. Would a better name be "dominating" visit distribution functions? It's a minor point but one that confused me for a while.
* [lines 129] For $D_{bound}$ are you assuming a distribution whose support is bounded, or a distribution with bounded variance. I assume the former, but this could be clearer, and the former would exclude a gaussian distribution.
* [lines 144-148] This was a bit difficult for me to parse.
* [line 166] By "count the agent's initial presence" I think you mean "reward" the agent for its presence in state s at the beginning of any trajectory from s.
* [Figure 3 caption] The last sentence should end with something like "...when greater reward is assigned to $l_{\nwarrow}$.
* [line 194] drop the word "instead"
* [lines 201-206] I get a bit lost here in the example.
* Definition 6.7 uses a very similar notation for a subset of possible states as it does for the full set of states (only the font style changes).
* [lines 333-334] It isn't really clear what you mean by randomly generated MDPs nor why they would be "unlikely" to have any particular property. Nor is it entirely clear what sufficient conditions you are referring to.


**Time Spent Reviewing:**

5

---

> ### Author Response · Authors · 2021-08-09
> **Reply to reviewer TsSV**
>
> Thank you for your careful review – we will incorporate your feedback into the camera-ready. We, too, are interested in the theoretical significance of state visit distribution functions. These functions make it easy to understand MDP structure, and they constitute a theoretical framing which seems under-appreciated in the literature. We outline further, incidental MDP theory contributions in the appendix (section C.1 on page 16).
>
> For $\mathcal{D}_\text{bound}$, we assume bounded support – not bounded variance. However:
> * Our optimality probability results hold for the orbits of all probability distributions.
> * We conjecture that the POWER results hold for more distributions – including the multivariate normal family. Bounded support of $\mathcal{D}$ is just a sufficient condition for the expectation of POWER$\_{\mathcal{D}’}(s,\gamma)$ being well-defined for all $s\in \mathcal{S}$, $\gamma\in(0,1)$, and all orbit elements $\mathcal{D}’\in S_{|\mathcal{S}|}\cdot \mathcal{D}$.
>
> > use of "tends" and "most" etc , seems to suggest that you are treating this orbit in some kind of measurable way
>
> We agree: the descriptive text for definition 6.4 ($\geq_\text{most}$) implicitly invokes the counting measure over orbit elements. However, as we note in lines 376-380, we need not consider all orbit elements equally plausible, or equally likely to be specified by RL designers. We will better clarify this point in the camera-ready.

---

> > ### Comment · Reviewer_TsSV · 2021-08-16
> > **Reply to reply**
> >
> > I've read the other reviews now and your responses to them. There are a few concerns among Reviewer U4CB, but of those that I understand I think your response deals appropriately with those. Partly the concern is, I think, one of semantics, by which I mean the meaning of the word "power".  This may be related to one point, that I wasn't sure I made very clearly was in the second to last bullet (I wrote uncommon whereas I believe I intended to write common). This would then make my text read:
> >
> > > It may be the case that certain topological properties of sub-parts of a naturally arising decision problem correlate with reward function over that sub-part. In fact, it is extremely **common** in games/human activities for task success to be associated with hard to reach states.
> >
> > This might be partly linked to the comments made by Reviewer U4CB in relation to pathologies in the definition of power.  Nonetheless, I feel that the scope of your results are made clear in the paper, and are well defended in your rebuttal and I am satisfied with that. This may (and I apologise for the repeated use of may/could/might) be related to the difference between a domain which is designed with a single objective in mind (e.g. by an adversarial designer), versus a more naturally arising one. In the former, states of high value are designed to be hard-to-reach and may be associated with reaching a goal (a terminating state). In a sense, I would like this paper to be published so that the discussion can be had. It is not at all obvious to me whether my point is a valid one.
> >
> > I note that Reviewer xxyk remarks that they believe your " paper is likely to be among the most high-impact ones at Neurips". I would not be surprised if it was.
> >
> > To conclude, I stand by my original assessment that this is a very high quality paper and I expect it to be of interest to RL researchers as well as more widely.

---

> > > ### Author Response · Authors · 2021-08-17
> > > **Clarification re:topological properties?**
> > >
> > > > In fact, it could be clearer what the motivation is for defining distributions of plausible reward functions in the way that is done. It may be the case that certain topological properties of sub-parts of a naturally arising decision problem correlate with reward function over that sub-part. In fact, it is extremely common in games/human activities for task success to be associated with hard to reach states. That isn't to say these states aren't of high power, just that the assumptions about IID rewards and orbits based on permutations of rewards, may not always be applicable.
> > >
> > > So that we can be sure to add proper clarification in the camera-ready, let us attempt to restate your point:
> > >
> > > **Summary:** Because hard-to-reach states can be more likely to correspond to task success, they are more likely to have high reward. For example, considering the orbit of a state indicator reward function $\mathbf{e}_s$, orbit elements placing their reward on hard-to-reach states are _a priori_ more empirically plausible than orbit elements placing their reward on easy-to-reach states. This is one example of a situation in which the counting measure on orbit elements may lead one to inappropriate inferences about what optimal policies tend to do for empirically likely reward functions.
> > >
> > > Is this a reasonable restatement of your point?

---

> > > > ### Comment · Reviewer_TsSV · 2021-08-18
> > > > **Confirmation on restatement**
> > > >
> > > > Yes. Your restatement is much clearer than my original.
> > > >
> > > > Making it explicit that there is a measure (or at least an implied one) on the orbits (the counting measure) would go a long way to satisfying me here. Discussing the implications of this with respect to the reachability of states would help readers to judge the appropriateness of your theory to any particular domain.
> > > >
> > > > Making appropriate changes to the camera ready paper would improve the paper (in my opinion) but my recommendation of accept is not predicated on the assumption that they will be made.

---

### Official Review · Reviewer_un2T · 2021-07-16

**Rating:** 7
**Confidence:** 3

**Summary:**

The paper proposes a formal definition for a measure of power in reinforcement learning. It is expressed as a scaled expectation of the difference of the optimal value function in a state and the immediate reward in that state, where the expectation is taken over possible reward functions and it is scaled by (1-\gamma)/\gamma. Using this measure, the authors show that any state or action that leaves more options open has more power.

**Limitations And Societal Impact:**

The authors discuss the societal impact of their work in a satisfying way.

**Main Review:**

The paper is generally very well-written and propose an intriguing framework to discuss about whether intelligent agent may seek more power. I also appreciate the small example of Markov decision process, which is provided to illustrate all the results.

However, I am not completely convinced that the proposed measure of power helps answer the initially-raised question of whether an intelligent agent would seek power, a question that I understand as focusing on one given agent, which would in the course of solving a given task seek power. It seems to me that the authors rather provide an answer to the question of whether given many intelligent agents with potentially different reward functions would they on average favor some states or actions for their optimal policies. For this latter question, the obtained results can be understood quite intuitively since a state or an action that leaves option open can potentially satisfy more various reward functions.
Although the authors mention that their results apply to the case with a degenerate distribution, it is not clear to me how the proposed measure is useful to answer the former question about one given agent. I think more discussion on that point would be warranted.

Besides, could the introduced notion of power be extended when the reward function depends on the action and/or future state?

POST-REBUTTAL
The authors' responses clarify the first point I raised and address my concern.

**Time Spent Reviewing:**

4

---

> ### Author Response · Authors · 2021-08-09
> **Reply to reviewer un2T**
>
> Thank you for your review; we’re glad that you find the paper well-written and intriguing.
>
> We are not certain we fully understand your concern about POWER and its ability to support our paper’s titular claim: that optimal agents tend to seek power. We will summarize and respond to our current understanding of your concern. Please let us know if your concern is not as described.
>
> Summarized concern: “POWER captures the agent’s expected optimal value at a state and discount rate, taking the expectation with respect to some distribution over reward functions. While your results indeed show that some states tend to have greater POWER$_{\mathcal{D}_\text{bound}}$ than others, we will want to understand whether an agent optimizing a fixed reward function will seek power – not whether mixtures of agents tend to do so.”
>
> Response to summarized concern: Suppose we plan on deploying an agent with reward function $R$, into an environment with the appropriate environmental symmetries (for example, the example rewardless MDP introduced in fig. 1). We represent $R$ as a degenerate probability distribution $\mathcal{D}_{R}$, which puts probability 1 on $R$ and 0 everywhere else. At e.g. $\gamma=\frac{1}{2}$, $R$ may make $\texttt{right}$ strictly suboptimal to $\texttt{left}$. However, our results show that for most permutations of the fixed reward function $R$, $\texttt{right}$ will be at least as good as $\texttt{left}$. Most permutations of $R$ will give greater value to keeping options open to the $\texttt{right}$, compared to closing them off by heading $\texttt{left}$.
>
> We therefore prove that for most ways we could specify a single reward function for a single agent, it will be optimal for the agent to take actions which keep its options open, and – at $\gamma=1$ – head towards larger sets of RSDs. In other words, we show that when the right symmetries exist, optimal policies tend to seek power. It is possible that in any given instance, we know enough about the structure of the fixed reward $R$ to know that its optimal policy will not take power-seeking actions, but our results demonstrate that you must know about such structure in the reward in order to be confident that none of its optimal policies will seek power.
>
> > Besides, could the introduced notion of power be extended when the reward function depends on the action and/or future state?
>
> POWER can be easily extended to account for state-action(-state’) reward functions, by simply considering a distribution over SA(S) reward functions and taking the expected optimal value over the drawn SA(S) reward functions. In this setting, POWER would not subtract current-state reward or divide by $\gamma$ (since dividing by $\gamma$ without subtracting current-state reward would make POWER tend to diverge as $\gamma\to 0$). See our response to reviewer U4CB for more on this choice.
>
> As for our main results on optimal policy tendencies, future work can extend them to the SA(S) case. Any SAS MDP can be converted into a SA MDP, and an SA MDP can be converted into a state-based reward MDP. At this point, this paper’s results will apply.

---

> > ### Comment · Reviewer_un2T · 2021-08-20
> > **About my first comment**
> >
> > Thank you for your answers. I think you reformulated correctly my first point, which is also the main criticism I have about your paper. I believe that among the literature that is cited in the introduction (given my superficial understanding of this previous literature), most work discusses the issue of an intelligent agent seeking power (please correct me if I’m wrong). However, in the formalization that is provided in your paper, the discussion shifts to the issue of looking at a population of intelligent agents. In my opinion, the introduction and related work (and maybe the title as well) don’t clarify enough between those two assertions:
> > “an optimal policy tends to seek power”
> > and
> > “over a set of possible reward functions, optimal policies statistically tend to seek POWER”
> > For a final version of this paper, I would suggest the authors to emphasize earlier that they focus on this second assertion.
> >
> > Regarding the second assertion, the results are not very surprising in my opinion, since when all possible reward functions are possible, what becomes important is the structure of the Markov process. Having said that, I appreciate the formalization effort. My evaluation was mainly based on the following impression I had: after my first reading of the introduction, I expected a discussion on the first assertion, which lead to some disappointment when I understood that the topic is about the second one.

---

> > > ### Author Response · Authors · 2021-08-21
> > > **Re: clarifying assertions**
> > >
> > > > I think you reformulated correctly my first point, which is also the main criticism I have about your paper.
> > >
> > > We do not quite understand what it would mean to discuss the first assertion: “an optimal policy tends to seek power”, or why that would be desirable. When considering the question of power-seeking incentives, three approaches spring to mind:
> > > 1. Showing that all optimal policies seek power.
> > > 2. Considering a specific agent and arguing that it seeks power.
> > > 3. Showing that most reward functions incentivize power-seeking.
> > >
> > >
> > > (1) is impossible: it is trivial to design an agent which assigns 1 reward to being dead, and 0 otherwise.
> > >
> > >
> > > (2) is uninformative: even if we could somehow guess what would be optimal for a particular agent, and argue that the agent would seek power, what would the point be? If we realized that a particular agent would seek power in an undesirable way, we would simply not build that agent. All we would learn is that one particular objective produces power-seeking incentives.
> > >
> > >
> > > (3) is our approach: we show that for _most_ ways to motivate an optimal agent — for most reward functions we could provide — power-seeking will be optimal in certain situations. This doesn’t let us know for sure whether a fixed reward function incentivizes power-seeking, but it gives us reason to suspect that that is true by default.
> > >
> > > > I believe that among the literature that is cited in the introduction (given my superficial understanding of this previous literature), most work discusses the issue of an intelligent agent seeking power (please correct me if I’m wrong). However, in the formalization that is provided in your paper, the discussion shifts to the issue of looking at a population of intelligent agents.
> > >
> > > No, this is not correct: we discuss the same issues as previous work. Previous work considers the possibility that, for _most ways_ we could design intelligent agents, the agents would have incentives to seek power, gain resources, and stay alive.
> > >
> > > In lines 42-46, we quote [Bostrom, 2014] (emphasis added):
> > >
> > > > Several instrumental values can be identified which are convergent in the sense that their attainment would increase the chances of the agent's goal being realized _for a wide range of final goals_ and a wide range of situations, implying that these instrumental values are _likely to be pursued by a broad spectrum_ of situated intelligent agents.
> > >
> > > On line 16, we cited [Omohundro, 2008]’s _Basic AI Drives_, which contains passages such as (emphasis added):
> > >
> > > > For _most_ utility functions, utility will not accrue if the system is turned off or destroyed
> > >
> > > And
> > >
> > > > _Almost any_ goal can be better accomplished by having more… resources
> > >
> > > On lines 48-49, we wrote: “Many AI alignment researchers hypothesize that _most_ advanced AI agents will have concerning instrumental incentives”, which reflects the literature’s focus on incentives which may be common to intelligent agents with a range of different goals. This is exactly what our work covers.

---

> > > > ### Comment · Reviewer_un2T · 2021-08-25
> > > > **Reply**
> > > >
> > > > Thank you for the clarification! That addresses my concern.

---

> ### Comment · Reviewer_xxyk · 2021-08-11
> **Quick comment**
>
> If it helps clarify things: using Definition 6.4, an optimal policy for a random reward function is robustly more likely to seek POWER than not to seek POWER. Though this doesn't allow us to be certain that a given optimal policy seeks POWER, it tells us to be careful when deploying a given optimal policy if we haven't ascertained whether it seeks POWER. And at the least it shows that power-seeking optimal policies are common. To me these seem like beneficial contributions to the ongoing debate cited in the introduction.

---

### Decision · Program_Chairs · 2021-09-27

**Decision:**

Accept (Spotlight)

**Comment:**

The discussion with the authors helped clear all the reviewer concerns.  The scores were all updated to clear accept.